# A Neural Network Approach for Efficiently Answering Most Probable Explanation Queries in Probabilistic Models

**Shivvrat Arya**
Department of Computer Science
The University of Texas at Dallas
shivvrat.arya@utdallas.edu

**Tahrima Rahman**
Department of Computer Science
The University of Texas at Dallas
tahrima.rahman@utdallas.edu

**Vibhav Gogate**
Department of Computer Science
The University of Texas at Dallas
vibhav.gogate@utdallas.edu

## Abstract

We propose a novel neural networks based approach to efficiently answer arbitrary Most Probable Explanation (MPE) queries—a well-known NP-hard task—in large probabilistic models such as Bayesian and Markov networks, probabilistic circuits, and neural auto-regressive models. By arbitrary MPE queries, we mean that there is no predefined partition of variables into evidence and non-evidence variables. The key idea is to distill all MPE queries over a given probabilistic model into a neural network and then use the latter for answering queries, eliminating the need for time-consuming inference algorithms that operate directly on the probabilistic model. We improve upon this idea by incorporating inference-time optimization with self-supervised loss to iteratively improve the solutions and employ a teacher-student framework that provides a better initial network, which in turn, helps reduce the number of inference-time optimization steps. The teacher network utilizes a self-supervised loss function optimized for getting the exact MPE solution, while the student network learns from the teacher's near-optimal outputs through supervised loss. We demonstrate the efficacy and scalability of our approach on various datasets and a broad class of probabilistic models, showcasing its practical effectiveness.

## 1 Introduction

Probabilistic representations such as Probabilistic Circuits (PCs) [8], graphical models [26] such as Bayesian Networks (BNs) and Markov Networks (MNs), and Neural Autoregressive Models (NAMs) [50] are widely used to model large, multi-dimensional probability distributions. However, they face a significant challenge: as the complexity of these distributions increases, solving practically relevant NP-hard inference tasks such as finding the Most Probable Explanation (MPE) via exact inference techniques [39, 40] becomes increasingly difficult and time-consuming. In particular, although various exact and approximate solvers exist for the MPE task in PCs, BNs and MNs, exact solvers are often too slow for practical use, and approximate solvers tend to lack the necessary accuracy, particularly in autoregressive models that currently rely on slow hill-climbing/beam search methods.

38th Conference on Neural Information Processing Systems (NeurIPS 2024).

In recent work, Arya et al. [4] proposed a method to overcome the limitations of existing approximate methods by using neural networks (NNs) to solve the MPE task in PCs.[1] Their method draws inspiration from the learning to optimize literature [12, 15, 29, 42, 55]. Given a PC and a *predefined partition of variables into query and evidence sets*, the core idea is to train a NN that takes an assignment to the evidence variables as input and outputs the most likely assignment to the query variables w.r.t. the distribution defined by the PC. Arya et al. suggest using either supervised or self-supervised learning techniques to train the NN; the former requires access to exact inference schemes, while the latter does not and is therefore more practical.

In this paper, we address a more general and complex version of the MPE task than the one considered by Arya et al. Specifically, we assume that there is *no predefined partition of the variables into evidence and query sets*, which we refer to as the **any-MPE** task. The complexity of the any-MPE task arises from the exponential increase in the number of input configurations, compounded by the exponential number of possible divisions of variables into evidence and query sets. Furthermore, our method applies to a broad class of probabilistic models, including BNs, MNs and NAMs, whereas Arya et al.'s method is limited to PCs. In addition, Arya et al.'s method does not fully exploit the capabilities of self-supervision, and the benefits of combining supervised and self-supervised loss functions.

This paper presents a novel approach that uses a NN for solving the any-MPE task in a broad class of probabilistic models (PMs) and achieves technical advancements in three key aspects:

**1. Efficient MPE Inference via Encoding Scheme and Loss Function:** We introduce a new encoding scheme that tailors the NN architecture to the specific structure of the input PM. This scheme not only delineates the input and output nodes for the NN but also establishes a methodology for setting input values and extracting the MPE solution from the NN's outputs. Furthermore, we propose a tractable, and differentiable self-supervised loss function, enabling efficient training.

**2. Inference Time Optimization with ITSELF:** We introduce a novel inference technique called Inference Time Self Supervised Training (ITSELF). This technique iteratively refines the MPE solution during the inference process itself. It utilizes gradient descent (*back-propagation*) to update the NN's parameters using our proposed self-supervised loss, leading to continual (anytime) improvement towards near-optimal solutions. ITSELF fully utilizes the power of our self-supervised loss, as it does not require labeled data or an external MPE solver.

**3. Two-Phase Pre-training with Teacher-Student Architecture:** To address challenges associated with self-supervised learning and ITSELF, we propose a two-phase pre-training strategy that leverages a teacher-student architecture. Self-supervised learning can suffer from overfitting and requires careful regularization. Additionally, ITSELF, especially with random initializations, might necessitate a substantial number of gradient updates to converge on optimal solutions. Our approach addresses these issues using the following methodology: (i) The teacher network first overfits the training data using ITSELF and (ii) The student network is then trained using supervised loss functions (e.g., binary cross-entropy) by treating the teacher network's output as pseudo-labels. This supervised training phase improves and regularizes the parameter learning process of the student network. It also provides a robust starting point for ITSELF, significantly reducing the required optimization steps and leading to substantial performance gains.

Finally, we conduct a detailed experimental comparison of our method with existing approaches on several types of PMs such as PCs, PGMs and NAMs. Our results demonstrate that our method surpasses state-of-the-art approximate inference techniques in terms of both accuracy and speed.

## 2   Background and Motivation

Without loss of generality, we use binary variables which take values from the set $\{0, 1\}$. We denote a random variable by an uppercase letter (e.g., $X$), and a value assigned to it by the corresponding lowercase letter (e.g., $x$). We denote a set of random variables by a bold uppercase letter (e.g., $\mathbf{X}$) and an assignment of values to all variables in the set by the corresponding bold lowercase letter (e.g., $\mathbf{x}$).

---

[1]Arya et al. [4] developed a NN-based method for solving the *marginal maximum-a-posteriori* (MMAP) task in PCs. In this paper, we focus on the MPE task, also sometimes referred to as the full MAP task, which is a special case of MMAP. Our method can be easily extended for solving the MMAP problem in PCs and tractable graphical models. For simplicity of exposition, we concentrate on the MPE task in this paper.

Throughout the paper when we use the term probabilistic models (PMs), we are referring to a broad class of probabilistic models in which computing the likelihood[2] of an assignment to all variables in the model can be done in polynomial (preferably linear) time in the size of the model. This class includes, among others, Bayesian and Markov networks collectively called Probabilistic Graphical Models (PGMs) [26], smooth and decomposable Probabilistic Circuits (PCs) [8], and Neural Autoregressive Models (NAMs) such as NADE [50] and MADE [17].

We are interested in solving the most probable explanation (MPE) task in PMs, namely the task of finding the most likely assignment to all unobserved (non-evidence) variables given observations (evidence). Formally, let $\mathcal{M}$ denote a probabilistic model defined over a set of variables $\mathbf{X}$ that represents the distribution $p_{\mathcal{M}}(\mathbf{x})$. We categorize the variables $\mathbf{X}$ into evidence $\mathbf{E} \subseteq \mathbf{X}$ and query $\mathbf{Q} \subseteq \mathbf{X}$ groups, ensuring that $\mathbf{E} \cap \mathbf{Q} = \emptyset$ and $\mathbf{E} \cup \mathbf{Q} = \mathbf{X}$. Then, given an assignment $\mathbf{e}$ to the set of evidence variables $\mathbf{E}$, the MPE task can be formulated as:

$$\mathrm{MPE}(\mathbf{Q}, \mathbf{e}) = \underset{\mathbf{q}}{\mathrm{argmax}}\, p_{\mathcal{M}}(\mathbf{q}|\mathbf{e}) = \underset{\mathbf{q}}{\mathrm{argmax}}\, \{\log p_{\mathcal{M}}(\mathbf{q}, \mathbf{e})\} \tag{1}$$

It is known that the MPE task is NP-hard in general and even hard to approximate [9, 11, 36, 41, 44].

**Motivation:** The goal of this paper is to develop a method that trains a NN for a given PM and, at test time, serves as an approximate MPE solver for any-MPE query posed over the PM. By any-MPE, we mean that the NN can take an assignment to an arbitrary subset of variables (evidence) as input and output the most likely assignments to the remaining (query) variables. Recently, Arya et al. [4] proposed a NN-based solution for solving the MPE task in PCs under the constraint that the partition of the variables into evidence and query sets *is known before training the NN*. This constraint is highly restrictive because, for generative models, it is unlikely that such a partition of variables is known in advance. In such cases, one would typically train a discriminative model rather than a generative one. Unlike Arya et al.'s method, our approach yields an any-MPE solver. Additionally, Arya et al.'s approach has several limitations in that it does not fully exploit the benefits of self-supervision during inference time and requires the use of relatively large NNs to achieve good performance in practice. Our proposed approach, described next, addresses these limitations.

## 3 A Self-Supervised Neural Approximator for any-MPE

In this section, we develop a neural network (NN) based approach for solving the *any-MPE* task. Specifically, given a PM, we develop an input encoding (see Section 3.1) that determines the number of input nodes of the NN and sets their values for the given MPE query. Additionally, we develop an output encoding scheme that specifies the number of NN output nodes required for the given PM and enables the recovery of the MPE solution from the outputs. For training the NN, we introduce a tractable and differentiable self-supervised loss function (see Section 3.2), whose global minima aligns with the MPE solutions to efficiently learn the parameters of the NN given *unlabeled data*.

### 3.1 An Encoding For any-MPE Instances

Since NNs require fixed-sized inputs and outputs, we introduce input and output encodings that generate fixed-length input and output vectors for each PM from a given MPE problem instance $\mathrm{MPE}(\mathbf{Q}, \mathbf{e})$. To encode the input, for each variable $X_i \in \mathbf{X}$, we associate two input nodes in the NN, denoted by $\hat{X}_i$ and $\bar{X}_i$. Thus for a PM having $n$ (namely, $|\mathbf{X}| = n$) variables, the corresponding NN has $2n$ input nodes. Given a query $\mathrm{MPE}(\mathbf{Q}, \mathbf{e})$, we set the values of the input nodes as follows: (1) If $X_i \in \mathbf{E}$ and $X_i = 0$ is in $\mathbf{e}$, then we set $\hat{X}_i = 0$ and $\bar{X}_i = 1$; (2) If $X_i \in \mathbf{E}$ and $X_i = 1$ is in $\mathbf{e}$, then we set $\hat{X}_i = 1$ and $\bar{X}_i = 0$; and (3) If $X_i \in \mathbf{Q}$ then we set $\hat{X}_i = 0$ and $\bar{X}_i = 0$. (The assignment $\hat{X}_i = 1$ and $\bar{X}_i = 1$ is not used.) It is easy to see that the input encoding described above yields an *injective* mapping between the set of all possible MPE queries over the given PM and the set $\{0, 1\}^{2n}$. This means that each unique MPE query $(\mathbf{Q}, \mathbf{e})$ will yield a unique 0-1 input vector of size $2n$.

The output of the neural network comprises of $n$ nodes with sigmoid activation, where each output node is associated with a variable $X_i \in \mathbf{X}$. We ignore the outputs corresponding to the evidence variables and define a loss function over the outputs corresponding to the query variables in the set $\mathbf{Q}$. The MPE solution can be reconstructed from the output nodes of the NN by thresholding the

---

[2]or a value proportional to it such as the unnormalized probability in Markov networks.

output nodes corresponding to the query variables appropriately (e.g., if the value of the output node is greater than 0.5, then the query variable is assigned the value 1; otherwise it is assigned to 0).

## 3.2 A Self-Supervised Loss Function for any-MPE

Since the output nodes of our proposed NN use sigmoid activation, each output is continuous and lies in the range $[0, 1]$. Given an MPE query $\text{MPE}(\mathbf{Q}, \mathbf{e})$, let $\mathbf{q}^c \in [0, 1]^{|\mathbf{Q}|}$ denote the (continuous) *Most Probable Explanation* (MPE) assignment predicted by the NN. In MPE inference, given $\mathbf{e}$, we want to find an assignment $\mathbf{q}$ such that $\log p_{\mathcal{M}}(\mathbf{q}, \mathbf{e})$ is maximized, namely, $-\log p_{\mathcal{M}}(\mathbf{q}, \mathbf{e})$ is minimized. Thus, a natural loss function that we can use is $-\log p_{\mathcal{M}}(\mathbf{q}, \mathbf{e})$. Unfortunately, the NN outputs a continuous vector $\mathbf{q}^c$ and as a result $p_{\mathcal{M}}(\mathbf{q}^c, \mathbf{e})$ is not defined.

Next, we describe how to solve the above problem by leveraging the following property of the class of PMs that we consider in this paper—specifically BNs, MNs, PCs and NAMs. In these PMs, the function $\ell(\mathbf{q}, \mathbf{e}) = -\log p_{\mathcal{M}}(\mathbf{e}, \mathbf{q})$, which is a function from $\{0, 1\}^n \to \mathbb{R}$ is either a multi-linear polynomial or a neural network, and can be computed in linear time in the size of the PM. To facilitate the use of continuous outputs, we define a loss function $\ell^c(\mathbf{q}^c, \mathbf{e}) : [0, 1]^n \to \mathbb{R}$ such that $\ell^c$ coincides with $\ell$ on $\{0, 1\}^n$. For PGMs and PCs, $\ell$ is a multi-linear function and $\ell^c$ is obtained by substituting each occurrence of a discrete variable $q_i \in \mathbf{q}$ with the corresponding continuous variable $q_i^c \in \mathbf{q}^c$ where $q_i^c \in [0, 1]$. In NAMs, $\ell$ is a NN and we can perform a similar substitution—we substitute each binary input $q_i$ in the NN with a continuous variable $q_i^c \in [0, 1]$. This substitution transforms the discrete NN into a continuous function while preserving its functional form.

An important property of $\ell^c$ is that it can be evaluated and differentiated in polynomial time. Moreover, when $\ell$ is defined by either a neural network (in NAMs) or a multilinear function (in BNs, MNs and PCs), the minimum value of $\ell^c$ over the domain $[0, 1]^n$ is less than or equal to the minimum value of the original function $\ell$ over the discrete domain $\{0, 1\}^n$. Formally,

**Proposition 1.** *Let $l(\mathbf{q}, \mathbf{e}) : \{0, 1\}^n \to \mathbb{R}$ be either a neural network or a multilinear function, and let $l^c(\mathbf{q}^c, \mathbf{e}) : [0, 1]^n \to \mathbb{R}$ be its continuous extension obtained by substituting each binary input $q_i$ with a continuous variable $q_i^c \in [0, 1]$. Then,*

$$\min_{\mathbf{q}^c \in [0,1]^n} \ell^c(\mathbf{q}^c, \mathbf{e}) \leq \min_{\mathbf{q} \in \{0,1\}^n} \ell(\mathbf{q}, \mathbf{e})$$

Following Arya et al. [4], we propose to improve the quality of the loss function by tightening the lower bound given in proposition 1 with an entropy-based penalty ($\ell_E$), governed by $\alpha > 0$.

$$\ell_E(\mathbf{q}^c, \alpha) = -\alpha \sum_{j=1}^{|\mathbf{Q}|} \left[ q_j^c \log(q_j^c) + (1 - q_j^c) \log(1 - q_j^c) \right] \tag{2}$$

This penalty encourages discrete solutions by preferring $q_j^c$ values close to 0 or 1, where $\alpha$ modulates the trade-off. Setting $\alpha$ to 0 yields the continuous approximation; conversely, an $\alpha$ value of $\infty$ results exclusively in discrete outcomes. From proposition 1 and by using the theory of Lagrange multipliers, we can show that for any $\alpha > 0$, the use of the entropy penalty yields a tighter lower bound:

**Proposition 2.**

$$\min_{\mathbf{q}^c \in [0,1]^n} \ell^c(\mathbf{q}^c, \mathbf{e}) \leq \min_{\mathbf{q}^c \in [0,1]^n} \ell^c(\mathbf{q}^c, \mathbf{e}) + \ell_E(\mathbf{q}^c, \alpha) \leq \min_{\mathbf{q} \in \{0,1\}^n} \ell(\mathbf{q}, \mathbf{e})$$

**How to use the Loss Function:** Given a PM defined over $n$ variables, we can use the self-supervised loss function $\ell^c(\mathbf{q}^c, \mathbf{e}) + \ell_E(\mathbf{q}^c, \alpha)$ (treating $\alpha$ as a hyper-parameter) to train any neural network (NN) architecture that has $2n$ input nodes and $n$ output nodes. This trained NN can then be used to answer any arbitrary MPE query posed over the PM. The training data for the neural network consists of assignments (evidence $\mathbf{e}$) to a subset of the variables. Each training example can be generated using the following three-step process. We first sample a full assignment $\mathbf{x}$ to all variables in the PM using techniques like Gibbs sampling or perfect sampling for tractable distributions such as PCs and BNs. Second, we choose an integer $k$ uniformly at random from the range $\{1, \ldots, n\}$ and designate $k$ randomly selected variables as evidence variables $\mathbf{E}$, and the remaining $n - k$ as query variables $\mathbf{Q}$. Finally, we project the full assignment $\mathbf{x}$ on $\mathbf{E}$. The primary advantage of using the self-supervised loss function is that it eliminates the need for access to a dedicated MPE solver to provide supervision during training; gradient-based training of the neural network provides the necessary supervision.

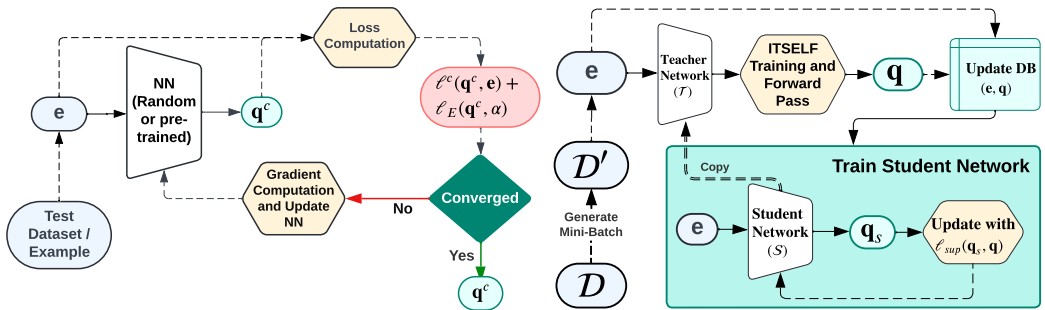

Figure 1: ITSELF Training Procedure    Figure 2: One Training Epoch for $\mathcal{GUIDE}$

### 3.3 Inference-Time Neural Optimization using Self-Supervised Loss

At a high level, assuming that the NN is over-parameterized, if we use the self-supervised loss and repeatedly run (stochastic) gradient updates over the NN for a given dataset, theoretical results [2, 13] as well as prior experimental work [46, 56] suggest that the parameters of the NN will converge to a point near the global minimum of the self-supervised loss function. This means that through gradient updates, the network will find a near-optimal MPE assignment for each training example. This strategy of performing gradient updates over the NN can also be used *during inference (test) time to iteratively improve the MPE solution*, thereby maximizing the benefits of self-supervision.

Specifically, at test time, given a test dataset (or example), we initialize the NN either randomly or using a pre-trained model and then run gradient-based updates over the NN iteratively until convergence. The gradient is computed w.r.t. the self-supervised loss function $\ell^c(\mathbf{q}^c, \mathbf{e}) + \ell_E(\mathbf{q}^c, \alpha)$. We call the resulting algorithm ITSELF (Inference Time Optimization using SELF-Supervised Loss), as detailed in Figure 1. The performance of ITSELF typically improves with each iteration until the loss converges.

Our proposed method, ITSELF, is closely related to test-time training approaches which are widely used to solve problems in deep learning [1, 10, 19, 30–32, 38, 49, 51, 57]. Our method differs from these previous approaches in that the global minima of our proposed self-supervised loss correspond to the MPE solutions, provided that the penalty $\alpha$ is sufficiently large.

## 4 Supervised Knowledge Transfer from ITSELF

A drawback of our self-supervised loss function is that, unlike supervised loss functions such as binary cross entropy, it is a non-convex function of the NN outputs[3]. As a result, it has a significantly larger number of local minima compared to the supervised loss function, but also a potentially exponential number of global minima, because an MPE problem can have multiple optimal solutions [35], all of which have the same loss function value. Thus, optimizing and regularizing using the self-supervised loss is difficult compared to a supervised loss, especially when the number of training examples is large.

Moreover, our experiments show that large datasets necessitate large, over-parameterized neural networks (NNs) to achieve near-optimal MPE solutions for all examples. However, when the training data is limited and the NN is sufficiently over-parameterized, our preliminary findings, along with theoretical and empirical results from prior studies [3, 6, 23, 27, 28], suggest that the NN is more likely to approach the global optima. Specifically, with a reasonably sized NN and a small dataset, the algorithm ITSELF tends to yield near-optimal MPE solutions. A further challenge with ITSELF is that even for small datasets, achieving convergence from a random initialization requires numerous iterations of gradient descent, rendering the training process inefficient and slow.

---

[3]Note that we are referring to convexity with respect to the outputs, not the parameters of the NN.

**Algorithm 1** GUided Iterative Dual LEarning with Self-supervised Teacher ($\mathcal{GUIDE}$)

---

1: **Input:** Training data $\mathcal{D}$, teacher $\mathcal{T}$ and student $\mathcal{S}$ having the same structure
2: **Output:** Trained student network $\mathcal{S}$
3:          ▷ Database $DB$ stores the best MPE assignment and loss value for each example in $\mathcal{D}$
4: **Initialize:** Randomly initialize $\mathcal{T}$, $\mathcal{S}$, and $DB$
5: **for** each epoch **do**
6:    Sample a mini-batch $\mathcal{D}'$ from $\mathcal{D}$
7:    Update the parameters of $\mathcal{T}$ using the algorithm ITSELF (self-supervised loss) with Dataset $\mathcal{D}'$
8:    **for** each example $\mathbf{e}_i$ in $\mathcal{D}'$ **do**
9:      Make a forward-pass over $\mathcal{T}$ to get an MPE assignment $\mathbf{q}_i$ for $\mathbf{e}_i$
10:      Update the entry in $DB$ for $\mathbf{e}_i$ with $\mathbf{q}_i$ if it has a lower loss value than the current entry
11:    **end for**
12:    Update the parameters of $\mathcal{S}$ using the mini-batch $\mathcal{D}'$ and labels from $DB$ and a supervised loss
13:    $\mathcal{T} \leftarrow \mathcal{S}$                            ▷ Initialize $\mathcal{T}$ with $\mathcal{S}$ for the next epoch
14: **end for**

---

## 4.1 Teacher-Student Strategy

To address these challenges (using small datasets with ITSELF; designing better initialization for it; and using non-convex loss functions for training), we propose a two-network teacher-student strategy [7, 16, 20–22, 24, 37, 47, 52–54], where we have two networks with the same structure that are trained via mini-batch gradient updates. The teacher network is overfitted to the mini-batch using our self-supervised loss via the ITSELF algorithm, and the student network is subsequently trained with a supervised loss function such as binary cross entropy. By overfitting the teacher network via ITSELF on the mini-batch, we ensure that it finds near-optimal MPE assignments for all (unlabeled) examples in the mini-batch and eventually over the whole training dataset.

The student network then learns from the teacher's outputs, using them as soft labels in a supervised learning framework. This transfer of knowledge mitigates the optimization difficulties associated with the non-convex self-supervised loss, allowing the student network to achieve faster convergence and better generalization with a more manageable model size. Additionally, this strategy reduces the need for severe over-parameterization and extensive training iterations for the teacher network because it is operating on a smaller dataset. It also helps achieve better initialization for ITSELF.

## 4.2 Training Procedure

Our proposed training procedure, which we call $\mathcal{GUIDE}$, is detailed in Algorithm 1. The algorithm trains a two-network system comprising a teacher network ($\mathcal{T}$) and a student network ($\mathcal{S}$) with the same structure. The goal is to train the student network using a combination of self-supervised and supervised learning strategies. The algorithm takes as input the training data $\mathcal{D}$, along with the teacher and student networks, $\mathcal{T}$ and $\mathcal{S}$, respectively and outputs a trained network $\mathcal{S}$. A database ($DB$) is utilized to store the best MPE assignment and corresponding loss value for each example in $\mathcal{D}$. The parameters of $\mathcal{T}$ and $\mathcal{S}$, and the entries in $DB$, are randomly initialized at the start.

In each epoch, a mini-batch $\mathcal{D}'$ is sampled from the training data $\mathcal{D}$. The parameters of the teacher network $\mathcal{T}$ are then updated using the ITSELF algorithm (which uses a self-supervised loss), applied to the mini-batch $\mathcal{D}'$ (the mini-batch helps address large data issues associated with ITSELF). For each example $\mathbf{e}_i$ in $\mathcal{D}'$, we perform a forward-pass over $\mathcal{T}$ to obtain an MPE assignment $\mathbf{q}_i$. The database $DB$ is subsequently updated with $\mathbf{q}_i$ if it has a lower loss value than the current entry for $\mathbf{e}_i$.

Following this, the parameters of the student network $\mathcal{S}$ are updated using the mini-batch $\mathcal{D}'$, the labels from $DB$, and a supervised loss function ($\ell_{sup}$) such as Binary Cross Entropy or $L2$ loss. Finally, the parameters of the teacher network $\mathcal{T}$ are reinitialized with the updated parameters of the student network $\mathcal{S}$ to prepare for the next epoch (addressing the initialization issue associated with ITSELF). Figure 2 illustrates a single training epoch of GUIDE.

Thus, at a high level, Algorithm 1 leverages the strengths of both self-supervised and supervised learning to improve training efficiency and reduce the model complexity, yielding a student network $\mathcal{S}$. Moreover, at test time, the student network can serve as an initialization for ITSELF.

# 5 Experiments

This section evaluates the ITSELF method (see section 3.3), the $\mathcal{GUIDE}$ teacher-student training method (see section 4) and the method that uses only self-supervised training, which we call SSMP (see section 3.2). We benchmark these against various baselines, including neural network-based and traditional polynomial-time algorithms that directly operate on the probabilistic model. We begin by detailing our experimental framework, including competing methods, evaluation metrics, neural network architectures, and datasets.

## 5.1 Datasets and Graphical Models

We used twenty binary datasets extensively used in tractable probabilistic models literature [5, 18, 34, 50]—referred to as TPM datasets—for evaluating PCs and NAMs. For the purpose of evaluating PGMs, we utilized high treewidth models from previous UAI inference competitions [14].

To train Sum Product Networks (SPNs), our choice of PCs, we employed the DeeProb-kit library [33], with SPN sizes ranging from 46 to 9666 nodes. For NAMs, we trained Masked Autoencoder for Distribution Estimation (MADE) models using PyTorch, following the approach in Germain et al. [17]. For Markov Networks (MNs), a specific type of PGM, we applied Gibbs sampling to generate 8,000, 1,000, and 1,000 samples for the training, testing, and validation sets, respectively. The query ratio ($qr$), defined as the fraction of variables in the query set, was varied across the set $\{0.1, 0.3, 0.5, 0.7, 0.8, 0.9\}$ for each probabilistic model (PM).

## 5.2 Baseline Methods and Evaluation Criteria

**PC**s - We used three polynomial-time baseline methods from the probabilistic circuits and probabilistic graphical models literature as benchmarks [41, 45].

- MAX Approximation (MAX) [45] transforms sum nodes into max nodes. During the upward pass, max nodes output the highest weighted value from their children. The downward pass, starting from the root, selects the child with the highest value at each max node and includes all children of product nodes.
- Maximum Likelihood Approximation (ML) [41] computes the marginal distribution $p_{\mathcal{M}}(Q_i|\mathbf{e})$ for each variable $Q_i \in \mathbf{Q}$, setting $Q_i$ to its most likely value.
- Sequential Approximation (Seq) [41] iteratively assigns query variables according to an order $o$. At each step $j$, it selects the $j$-th query variable $Q_j$ in $o$ and assigns to it a value $q_j$ such that $p_{\mathcal{M}}(q_j|\mathbf{e}, \mathbf{y})$ is maximized, where $\mathbf{y}$ is an assignment of values to all query variables from 1 to $j - 1$.

We further evaluated the impact of initializing stochastic hill climbing searches using solutions from all baseline approaches and our proposed methods for MPE inference, conducting 60-second searches for each MPE problem in our experiments, as detailed in Park and Darwiche [41].

**NAM**s - As a baseline, we used the stochastic hill-climbing search (HC) algorithm. Following a procedure similar to that used for PCs, we conducted a 60-second hill-climbing search for each test example, with query variables initialized randomly and setting evidence variables according to the values in the given example.

**PGM**s - We employed the distributed AND/OR Branch and Bound (AOBB) method [39] as a baseline, using the implementation outlined in Otten [40]. Since AOBB is an anytime algorithm, we set a 60-second time limit for inference per test example.

**Neural Baselines** - Arya et al. [4] introduced Self-Supervised learning based MMAP solver for PCs (SSMP), training a neural network to handle queries on a fixed variable partition within PCs. We extend this approach to address the any-MPE task in PMs (see Section 3.2), using a single network to answer any-MPE queries as an additional neural baseline.

**Evaluation Criteria** - We evaluated competing approaches based on log-likelihood (LL) scores, calculated as $\ln p_{\mathcal{M}}(\mathbf{e}, \mathbf{q})$, and inference times for given evidence $\mathbf{e}$ and query output $\mathbf{q}$. Higher log-likelihood scores indicate better performance, while shorter inference times are preferable.

## 5.3 Neural Network-Based Approaches

We implemented two neural network training protocols for each PM and query ratio: SSMP and $\mathcal{GUIDE}$. Each model was trained for 20 epochs following the training procedure outlined by Arya et al. [4] for SSMP. Both protocols employed two distinct inference strategies, thus forming four neural-based variants. In the first strategy, we performed a single forward pass through the network to estimate the values of query variable, as specified by Arya et al. [4]. The second strategy utilized our novel test-time optimization-based ITSELF approach for inference. The ITSELF optimization terminates after 100 iterations or upon loss convergence for both PCs and PGMs. For NAMs, we increase the limit to 1,000 iterations while keeping the convergence criterion.

We standardized network architectures for PMs across all experiments. For PCs, we used fully connected Neural Networks (NN) with three hidden layers (128, 256, 512 nodes). For NAMs and PGMs, a single hidden layer of 512 nodes was employed. All hidden layers featured ReLU activation, while the output layers used sigmoid functions with dropout for regularization [48]. We optimized all models using Adam [25] and implemented them in PyTorch [43] on an NVIDIA A40 GPU.

**Results for PCs**: We compare methods—including three polynomial-time baselines, neural network-based SSMP, and our ITSELF and $\mathcal{GUIDE}$ methods—on 20 TPM datasets as shown in the contingency table in figure 3a (detailed results in the supplementary materials). We generated 120 test datasets for the MPE task using 20 PCs across 6 query ratios ($qr$). Each cell $(i, j)$ in the table represents how often (out of 120) the method in row $i$ outperformed the method in column $j$ based on average log-likelihood scores. Any difference between 120 and the combined frequencies of cells $(i, j)$ and $(j, i)$ indicates cases where the compared methods achieved similar scores. We present similar contingency tables for Hill Climbing Search over PCs (Fig. 3b), NAMs (Fig. 3c), and PGMs (Fig. 3d) to benchmark the proposed methods against the baselines.

The contingency table for PC (Fig. 3a) shows that methods incorporating ITSELF consistently outperform both polynomial-time and traditional neural baselines, as indicated by the dark blue cells in the corresponding rows. Notably, $\mathcal{GUIDE}$ + ITSELF is superior to all the other methods in almost two-thirds of the 120 cases, while SSMP + ITSELF is better than both SSMP and $\mathcal{GUIDE}$. In contrast, the polynomial-time baseline MAX is better than both SSMP and $\mathcal{GUIDE}$ (as used in Arya et al. [4]), highlighting ITSELF's significant role in boosting model performance for the complex *any-MPE* task.

We compare MAX and $\mathcal{GUIDE}$ + ITSELF using a heatmap in Figure 4a. The y-axis presents datasets by variable count and the x-axis represents query ratio. Each cell displays the percentage difference in mean LL scores between the methods, calculated as %Diff. $= 100 \times (ll_{nn} - ll_{max})/|ll_{max}|$. The heatmap shows that $\mathcal{GUIDE}$ + ITSELF achieves performance comparable to MAX for small query sets. As the problem complexity increases with an increase in query set size, our method consistently outperforms MAX across all datasets, except for NLTCS and Tretail, as highlighted by the green cells. In the 12 cases where $\mathcal{GUIDE}$ + ITSELF underperforms, the performance gap remains minimal, as indicated by the limited number of red cells in the heatmap.

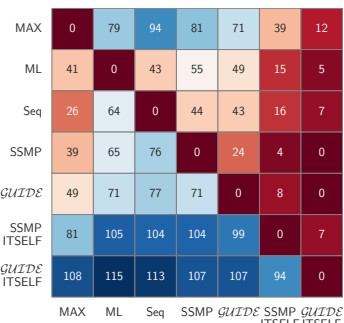

(a) PCs: Initial Solutions

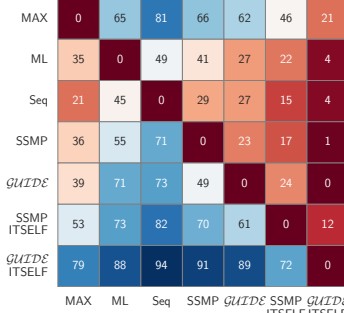

(b) PCs: Hill-Climbing

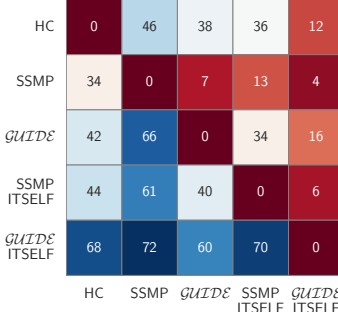

(c) NAMs

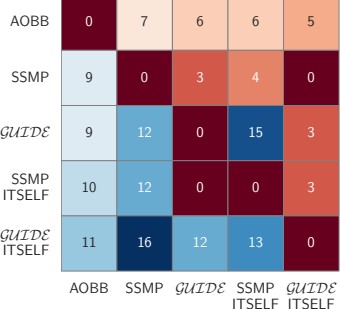

(d) PGMs

Figure 3: MPE method comparison across PMs. Blue shows row superiority, red shows column superiority; darker shades indicate larger values.

Figure 3b further analyzes the performance of our proposed methods against various baselines as initialization strategies for Hill Climbing Search. This comparison evaluates the effectiveness of ITSELF and $\mathcal{GUIDE}$ in enhancing *anytime methods* compared to conventional heuristic initialization approaches. Notably, methods incorporating ITSELF provide superior initialization for local search-based algorithms.

**Results for NAMs**: The contingency table in Figure 3c presents our evaluation of several methods for NAMs, including HC and two neural network approaches, SSMP and $\mathcal{GUIDE}$, each tested with two inference schemes. We evaluated these methods on 20 TPM datasets, creating 80 test sets for the MPE task using 20 MADEs across four query ratios ($qr$).

The $\mathcal{GUIDE}$ + ITSELF approach demonstrates superior performance compared to both baseline methods and other neural inference schemes, aligning with observations from PC. While HC outperforms SSMP, both $\mathcal{GUIDE}$ and the combination of SSMP-based training with ITSELF-based inference surpass HC, highlighting their advantages over the baseline.

The heatmaps in Figure 4b further highlight the superior performance of $\mathcal{GUIDE}$ + ITSELF for NAMs, particularly in larger datasets where it outperforms the HC baseline by over 50% in most cases, as indicated by the dark green cells. The combination of $\mathcal{GUIDE}$-based learning with ITSELF-based inference consistently outperforms the baseline across most datasets, with exceptions only in the Mushrooms, Connect 4, and Retail. Overall, the $\mathcal{GUIDE}$ + ITSELF approach significantly enhances the quality of the MPE solutions in NAM models.

**Results for PGMs**: The contingency table in 3d compares the performance of AOBB and four neural-network-based methods on PGMs across four high-treewidth networks. For this evaluation, we generated 16 test datasets for the MPE task using four PGMs across four query ratios ($qr$).

Consistent with results from previous PMs, methods using ITSELF for inference consistently outperform the baseline methods AOBB and SSMP across most scenarios. Both $\mathcal{GUIDE}$ and SSMP outperform AOBB in at least 50 percent of the tests. The supplementary material presents comparisons against exact solutions, conducted on less complex probabilistic models where ground truth computation remains tractable.

**Does a teacher-student-based network outperform a single network trained with the self-supervised loss? ($\mathcal{GUIDE}$ vs. SSMP):**

This analysis aims to evaluate the performance of $\mathcal{GUIDE}$ against traditional neural network training methods used in SSMP across different PMs and inference schemes. Using traditional inference scheme (i.e., one forward pass through the network), $\mathcal{GUIDE}$ consistently outperforms SSMP, demonstrating its superiority in 60% of scenarios for PCs, more than 80% for NAM models, and 75% for PGM models. When employing ITSELF-based inference, $\mathcal{GUIDE}$ maintains this advantage, achieving higher quality solutions in more than 75%, 85%, and 80% of cases for PCs, NAMs, and PGMs, respectively. Therefore, models trained using $\mathcal{GUIDE}$ are consistently superior to those trained with SSMP for the *any-MPE* task.

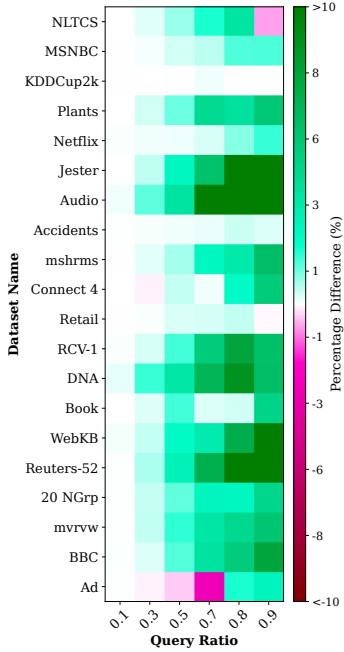

(a) PC: $\mathcal{GUIDE}$ + ITSELF vs. MAX

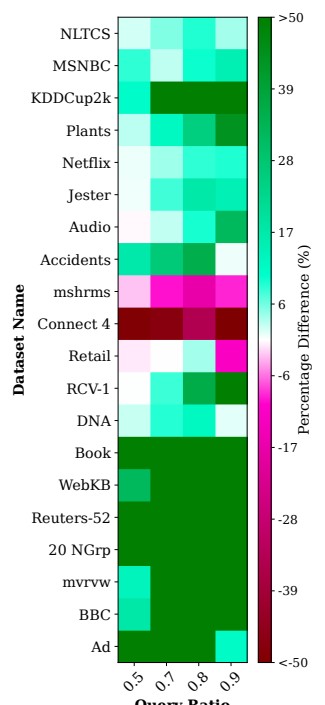

(b) NAM: $\mathcal{GUIDE}$ + ITSELF vs. HC

Figure 4: Heatmaps showing LL % Differences. Top: PC; Bottom: NAM. Green cells: our method is better. Darker shades indicate larger values.

**Does inference time optimization improve performance? (One-Pass vs. Multi-Pass):**

In this analysis, we compare the performance of the single-pass inference method to that of the proposed multi-pass inference method (ITSELF). ITSELF combined with SSMP training outperforms the other methods in over 85% cases for PC, and more than 75% for NAM and PGM models. When used on models trained with $\mathcal{GUIDE}$, ITSELF demonstrates even better results, achieving superior performance in nearly 90% of PC cases and 75% for both NAMs and PGMs. Overall, $\mathcal{GUIDE}$ with ITSELF inference emerges as the most effective method across all experiments. Empirical evidence consistently demonstrates ITSELF's superiority over single-pass inference across PMs.

The inference time analysis, detailed in the supplementary material, compares computational efficiency across methods using the natural logarithm of execution time in microseconds. Neural network-based approaches with traditional inference demonstrate the fastest performance across all PMs, as they only require a single forward pass to compute query variable values. For MADE, models trained with $\mathcal{GUIDE}$ and ITSELF are the next most efficient. In PGMs, $\mathcal{GUIDE}$ + ITSELF ranks third, followed by SSMP + ITSELF. For PCs, MAX is marginally faster than both $\mathcal{GUIDE}$ + ITSELF and SSMP + ITSELF, while ML and Seq have the longest computational times. In general, models trained with $\mathcal{GUIDE}$ achieve shorter inference times than those trained with the self-supervised loss (SSMP), as they require fewer ITSELF iterations due to more effective initial training.

**Summary:** Our experiments demonstrate that $\mathcal{GUIDE}$ + ITSELF outperforms both polynomial-time and neural-based baselines across various PMs, as evidenced by higher log-likelihood scores. Notably, ITSELF demonstrates significant advantages over traditional single-pass inference in addressing the complex *any-MPE* query task within probabilistic models, emphasizing the importance of Inference Time Optimization. Furthermore, the superior performance of models trained with $\mathcal{GUIDE}$ compared to SSMP highlights the effectiveness of the dual network approach, which improves initial model quality and establishes an optimal starting point for ITSELF.

## 6 Conclusion and Future Work

We introduced novel methods for answering Most Probable Explanation (MPE) queries in probabilistic models. Our approach employs self-supervised loss functions to represent MPE objectives, enabling tractable loss and gradient computations during neural network training. We also proposed a new inference time optimization technique, ITSELF, which iteratively improves the solution to the MPE problem via gradient updates. Additionally, we introduced a dual-network-based strategy that combines supervised and unsupervised training which we call $\mathcal{GUIDE}$ to provide better initialization for ITSELF and addressing various challenges associated with self-supervised training. Our method was tested on various benchmarks, including probabilistic circuits, neural autoregressive models, and probabilistic graphical models, using 20 binary datasets and high tree-width networks. It outperformed polytime baselines and other neural methods, substantially in some cases. Additionally, it improved the effectiveness of stochastic hill climbing (local) search strategies.

Future work includes solving complex queries in probabilistic models with constraints; training neural networks with losses from multiple probabilistic models to embed their inference mechanisms; boosting performance by developing advanced encoding strategies for similar tasks; implementing sophisticated neural architectures tailored to probabilistic models; etc.

## Acknowledgements

This work was supported in part by the DARPA Perceptually-Enabled Task Guidance (PTG) Program under contract number HR00112220005, the DARPA Assured Neuro Symbolic Learning and Reasoning (ANSR) Program under contract number HR001122S0039, the National Science Foundation grant IIS-1652835, and the AFOSR award FA9550-23-1-0239.

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

## A  Experimental Setup

### A.1  Datasets and Models

Table 1 summarizes the datasets and the probabilistic circuits trained on them. We use the same datasets for both PCs [8] and NAMs [17, 50]. The selection includes both smaller datasets, such as NLTCS and MSNBC, and larger datasets with over 1000 nodes.

For Markov networks, we utilize high treewidth grid networks, specifically grid40x40.f2.wrap, grid40x40.f5.wrap, grid40x40.f10.wrap, and grid40x40.f15.wrap. Each model contains 4800 variables and 1600 factors.

Table 1: Summary of datasets used with their respective numbers of variables and nodes in probabilistic circuits.

| Dataset | Number of Variables | Number of Nodes in PC |
|---|---|---|
| NLTCS | 16 | 125 |
| MSNBC | 17 | 46 |
| KDDCup2k | 64 | 274 |
| Plants | 69 | 3737 |
| Audio | 100 | 348 |
| Jester | 100 | 274 |
| Netflix | 100 | 400 |
| Accidents | 111 | 1178 |
| Mushrooms | 112 | 902 |
| Connect 4 | 126 | 2128 |
| Retail | 135 | 359 |
| RCV-1 | 150 | 519 |
| DNA | 180 | 1855 |
| Book | 500 | 1628 |
| WebKB | 839 | 3154 |
| Reuters-52 | 889 | 7348 |
| 20 NewsGroup | 910 | 2467 |
| Movie reviews | 1001 | 2567 |
| BBC | 1058 | 3399 |
| Ad | 1556 | 9666 |

### A.2  Hyperparameters Details

Our experimental framework was designed to ensure consistency and efficiency across all conducted experiments. For NAM's, we used MADE, training the model with two hidden layers of 512 and 1024 units, respectively, using the hyperparameters from Germain et al. [17].

For neural network-based solvers, the mini-batch size was set to 512 samples, and a learning rate decay strategy, reducing the rate by 0.9 upon loss plateauing, was implemented to improve training efficiency. Optimal hyperparameters were identified via extensive 5-fold cross-validation.

In discrete loss scenarios, the hyperparameter $\alpha$ played a pivotal role. We systematically explored the optimal $\alpha$ value across the range $0.001, 0.01, 0.1, 1, 10, 100, 1000$ for neural-based models, including ITSELF and $\mathcal{GUIDE}$. Notably, higher $\alpha$ values better constrain outputs to binary, thereby facilitating near-optimal results.

## B  Extending the Current Approach to Other Data Types and Inference Tasks

The current approach can be extended to support both multi-valued discrete and continuous variables, broadening its utility in diverse scenarios.

For multi-valued discrete variables, the method can be adapted by implementing a multi-class, multi-output classification head. Each query variable is represented by a softmax output node, which provides soft evidence by generating probabilistic distributions across multiple discrete values.

To incorporate continuous variables, we introduce a linear activation function in the output layer. The loss function, specifically the multi-linear representation of the PM, is modified to accommodate continuous neural network outputs. For example, in Probabilistic Circuits that use Gaussian distributions, continuous values can be directly integrated into the loss function, facilitating gradient-based backpropagation.

These extensions primarily involve adjusting the network's output layer and refining the self-supervised loss function represented by the PM. Notably, other elements of our approach, including the ITSELF and GUIDE procedures, remain unchanged.

Our approach further extends to additional inference tasks over probabilistic models, including marginal MAP and constrained most probable explanation (CMPE) tasks. However, the scalability of this approach depends on the computational efficiency of evaluating the loss function for each inference task. When this evaluation becomes computationally infeasible, the proposed method—training a neural network to answer queries over probabilistic models—may itself become infeasible. For example, performing marginal MAP inference over NAMs and PGMs requires repeated evaluations of the loss function associated with the marginal MAP task and its gradient during training. This iterative process, essential for updating the neural network's parameters, can become prohibitively resource-intensive due to the high computational demands of evaluating the marginal MAP loss over these probabilistic models.

## C    A Comparative Analysis of Performance of ITSELF for Different Pre-Training Methods

This section evaluates the performance of models initialized through various techniques—random initialization, SSMP, and $\mathcal{GUIDE}$. Each plot represents the loss for a distinct test example, with the x-axis denoting the number of ITSELF iterations and the y-axis showing the Negative Log Likelihood (NLL) scores. Lower NLL values signify better solutions. Through this empirical assessment, we compare the impact of different pre-training methods on model performance.

Figures 5 to 28 present the plots for NAMs. The plots for PCs are shown in Figures 29 to 67. Figures 68 to 78 illustrate the plots for PGMs. We selected the following datasets for PCs and NAMs: DNA, RCV-1, Reuters-52, Netflix, WebKB, Audio, Moviereview, and Jester. For PGMs, we used all the datasets presented in the main paper. Each plot consists of two sections. The left section presents the Negative Log-Likelihood Loss for 1000 iterations for all methods. The right section contains two sub-plots: the top sub-plot displays the zoomed-in losses for the first 200 iterations, while the bottom sub-plot shows the zoomed-in losses for the last 200 iterations.

We randomly initialize the parameters for the random model and perform 1000 iterations of ITSELF for inference. For the two pre-trained models (SSMP and $\mathcal{GUIDE}$), we update the top $N$ layers, where $N$ is the number of layers corresponding to that loss curve, and fix the remaining bottom layers. We extract features by passing the input through these fixed layers and then train the parameters of the top $N$ layers. We again perform 1000 iterations of ITSELF for inference. For NAMs and PGMs, we use neural networks with up to one hidden layer, while for PCs, we employ models with up to three hidden layers.

From the plots for the three Probabilistic Modelss (PMs), we observe that models pre-trained using the proposed $\mathcal{GUIDE}$ training scheme generally have a better starting point for ITSELF, indicated by a lower loss, compared to all other models. Across a wide array of datasets, PGMs, and query percentages, the $\mathcal{GUIDE}$ method consistently converges to a lower or equivalent loss compared to other models. Remarkably, it sometimes achieves a loss value that is less than half of the nearest competing model. Furthermore, the losses for $\mathcal{GUIDE}$ are typically more stable than those of other initialization. In some scenarios, all models achieve a similar final loss, although models initialized with SSMP and those randomly initialized may experience oscillations in their loss values.

Models pre-trained using the traditional self-supervised loss (SSMP) typically have better or similar starting points than randomly initialized models. However, models pre-trained using the SSMP method might converge to a worse loss than their $\mathcal{GUIDE}$ pre-trained counterparts.

In most cases, convergence is rapid, even with a reduced learning rate of $10^{-4}$ compared to the experiments shown in the main paper. Most methods converge within 200 to 300 iterations, although some may still oscillate during the later iterations of ITSELF.

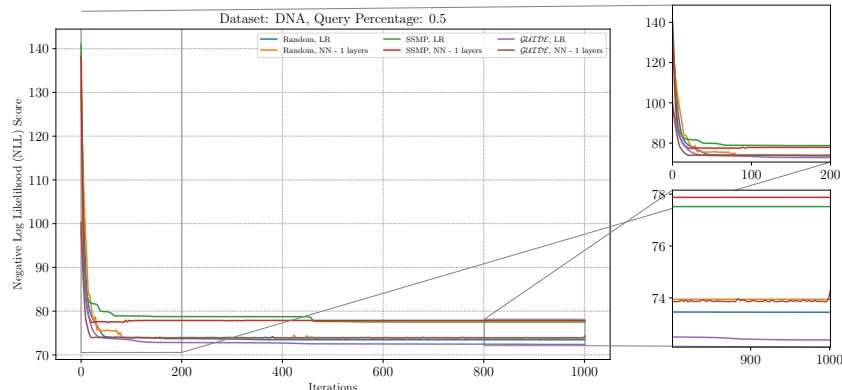

Figure 5: Analysis of ITSELF Loss Across Various Pre-Trained Models for NAMs on the DNA Dataset at a Query Ratio of 0.5.

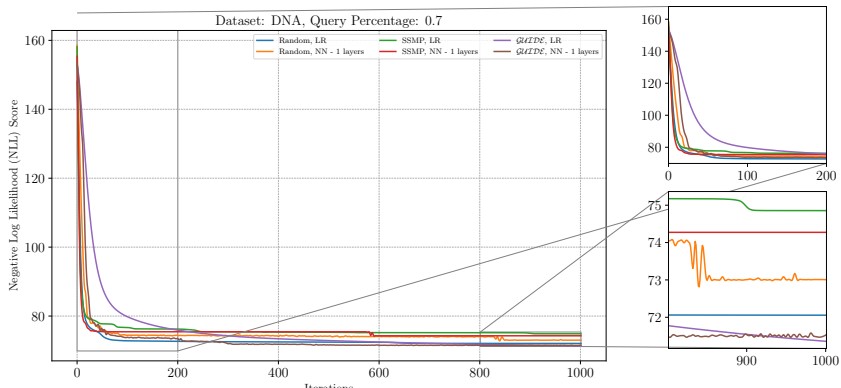

Figure 6: Analysis of ITSELF Loss Across Various Pre-Trained Models for NAMs on the DNA Dataset at a Query Ratio of 0.7.

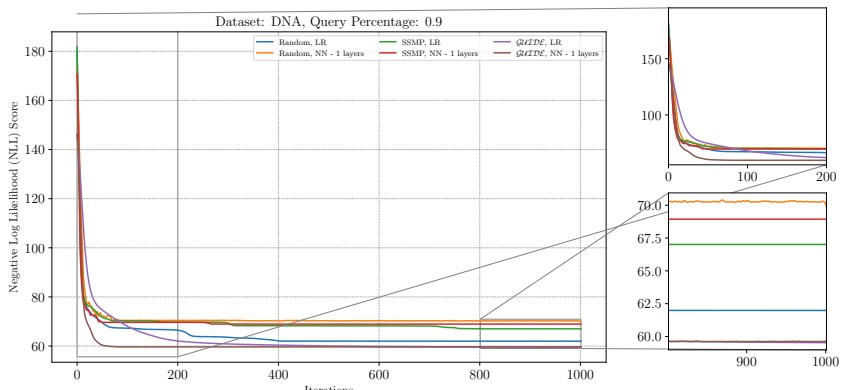

Figure 7: Analysis of ITSELF Loss Across Various Pre-Trained Models for NAMs on the DNA Dataset at a Query Ratio of 0.9.

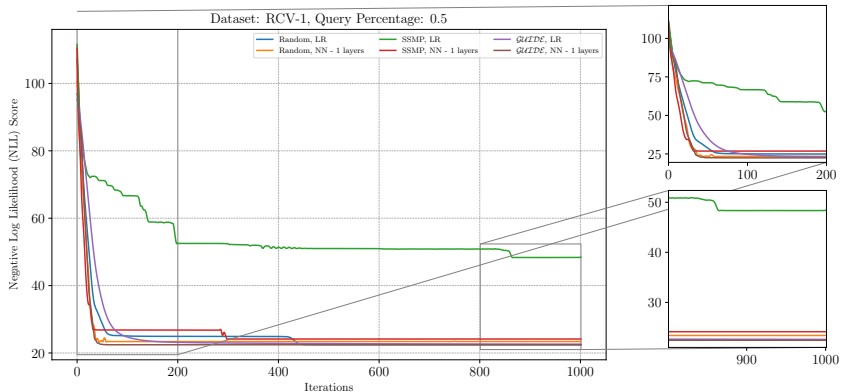

Figure 8: Analysis of ITSELF Loss Across Various Pre-Trained Models for NAMs on the RCV-1 Dataset at a Query Ratio of 0.5.

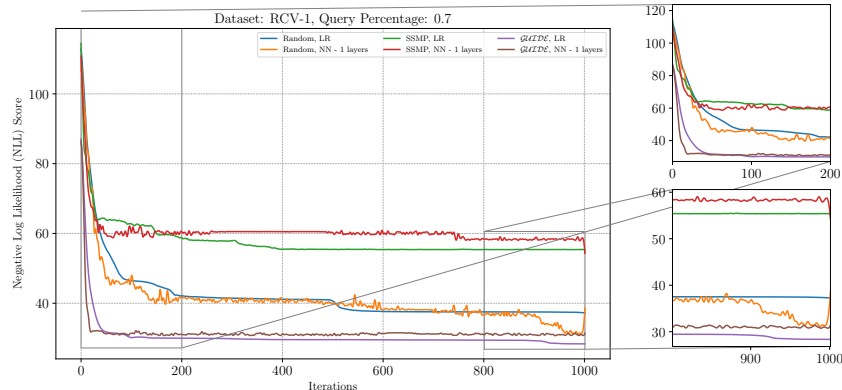

Figure 9: Analysis of ITSELF Loss Across Various Pre-Trained Models for NAMs on the RCV-1 Dataset at a Query Ratio of 0.7.

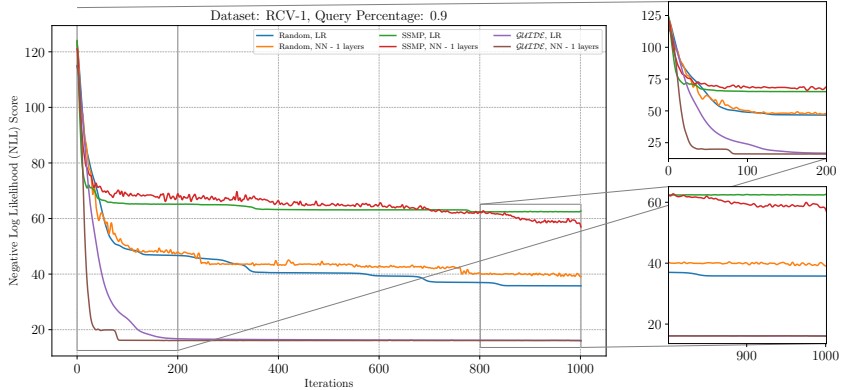

Figure 10: Analysis of ITSELF Loss Across Various Pre-Trained Models for NAMs on the RCV-1 Dataset at a Query Ratio of 0.9.

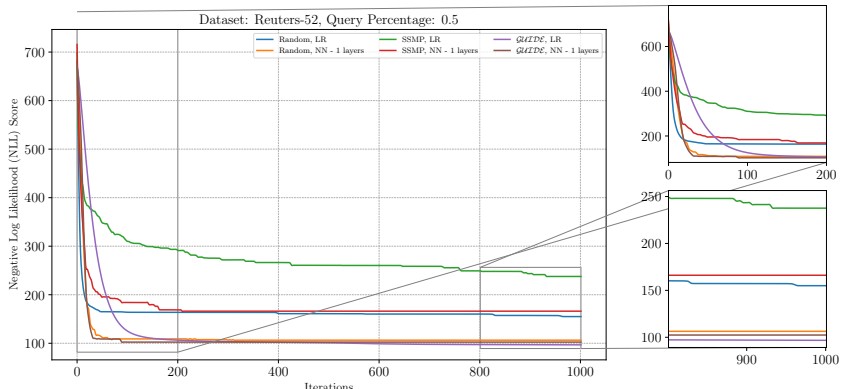

Figure 11: Analysis of ITSELF Loss Across Various Pre-Trained Models for NAMs on the Reuters-52 Dataset at a Query Ratio of 0.5.

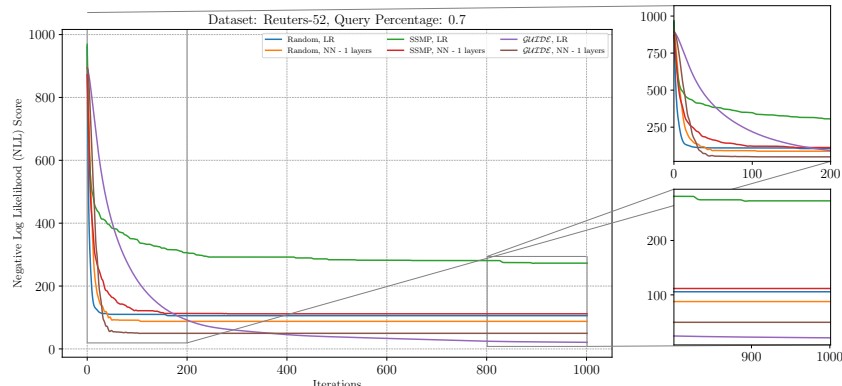

Figure 12: Analysis of ITSELF Loss Across Various Pre-Trained Models for NAMs on the Reuters-52 Dataset at a Query Ratio of 0.7.

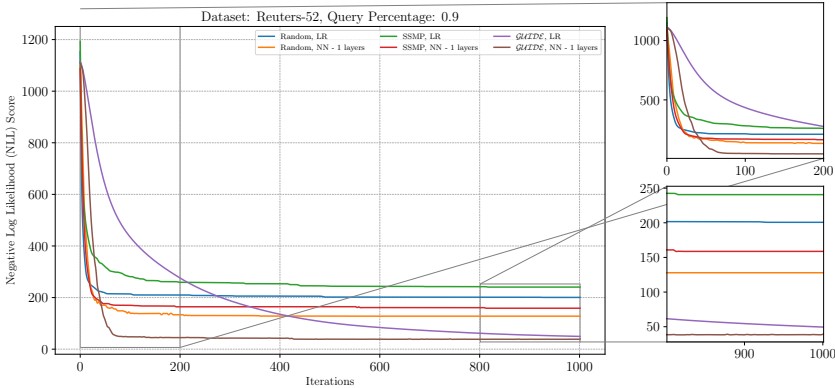

Figure 13: Analysis of ITSELF Loss Across Various Pre-Trained Models for NAMs on the Reuters-52 Dataset at a Query Ratio of 0.9.

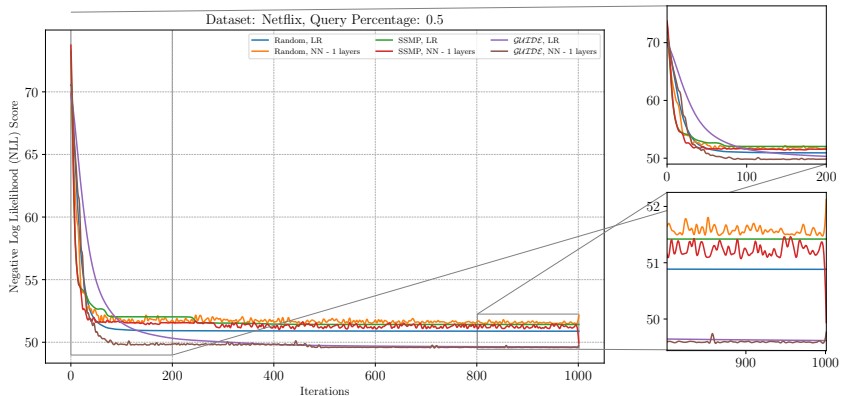

Figure 14: Analysis of ITSELF Loss Across Various Pre-Trained Models for NAMs on the Netflix Dataset at a Query Ratio of 0.5.

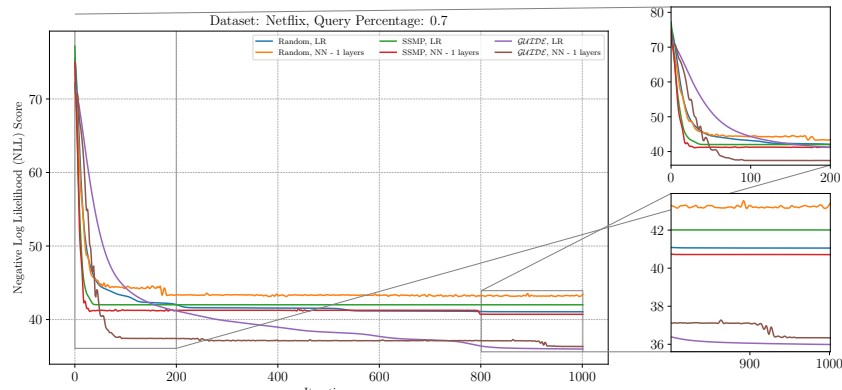

Figure 15: Analysis of ITSELF Loss Across Various Pre-Trained Models for NAMs on the Netflix Dataset at a Query Ratio of 0.7.

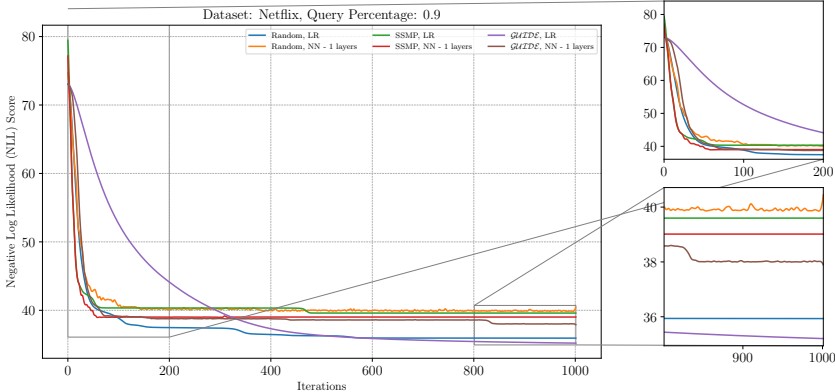

Figure 16: Analysis of ITSELF Loss Across Various Pre-Trained Models for NAMs on the Netflix Dataset at a Query Ratio of 0.9.

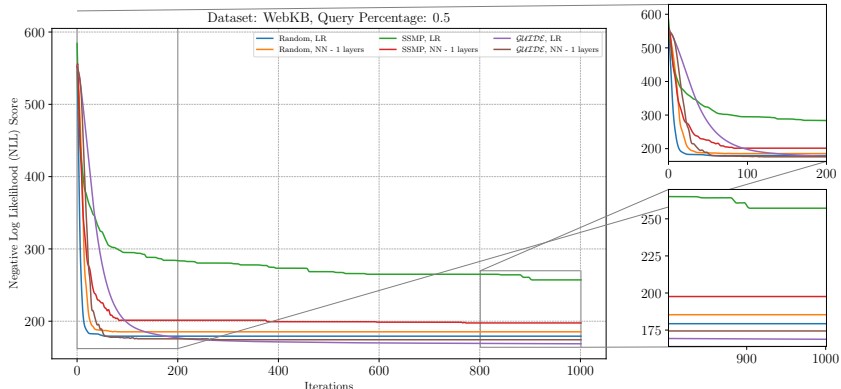

Figure 17: Analysis of ITSELF Loss Across Various Pre-Trained Models for NAMs on the WebKB Dataset at a Query Ratio of 0.5.

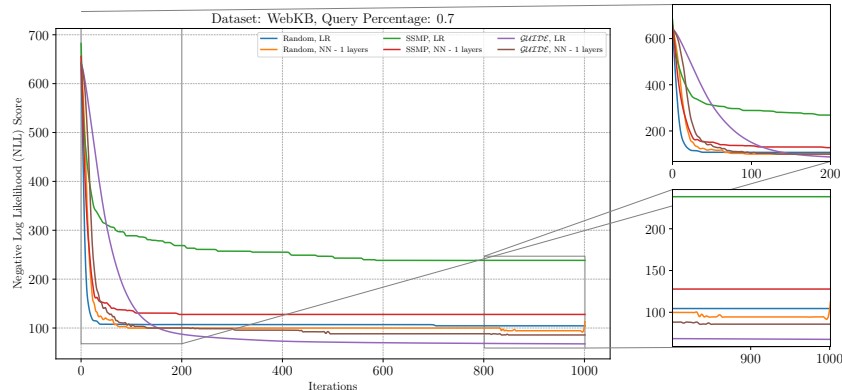

Figure 18: Analysis of ITSELF Loss Across Various Pre-Trained Models for NAMs on the WebKB Dataset at a Query Ratio of 0.7.

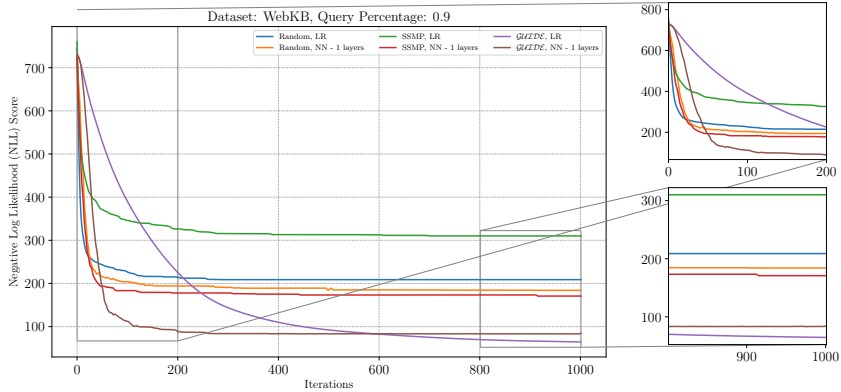

Figure 19: Analysis of ITSELF Loss Across Various Pre-Trained Models for NAMs on the WebKB Dataset at a Query Ratio of 0.9.

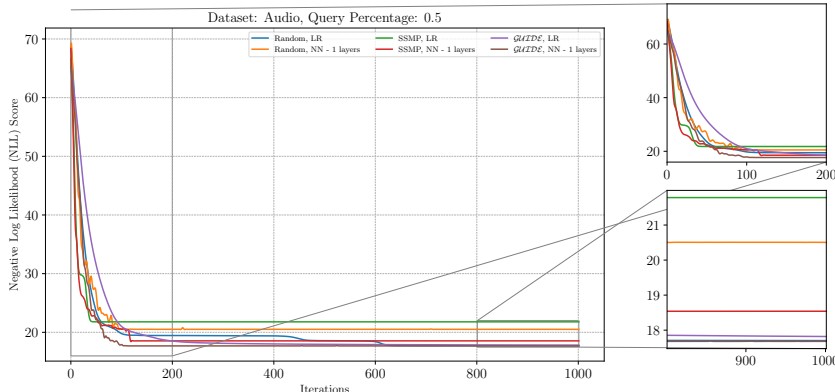

Figure 20: Analysis of ITSELF Loss Across Various Pre-Trained Models for NAMs on the Audio Dataset at a Query Ratio of 0.5.

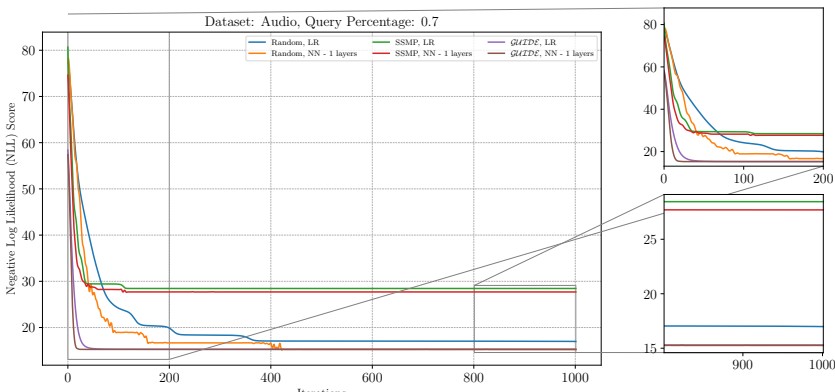

Figure 21: Analysis of ITSELF Loss Across Various Pre-Trained Models for NAMs on the Audio Dataset at a Query Ratio of 0.7.

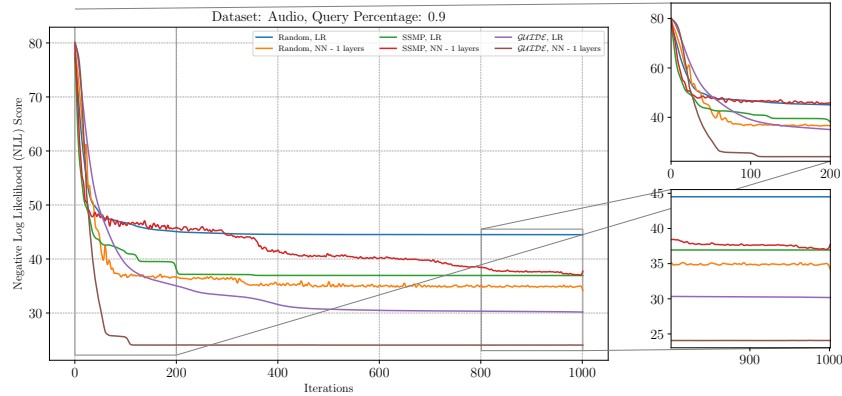

Figure 22: Analysis of ITSELF Loss Across Various Pre-Trained Models for NAMs on the Audio Dataset at a Query Ratio of 0.9.

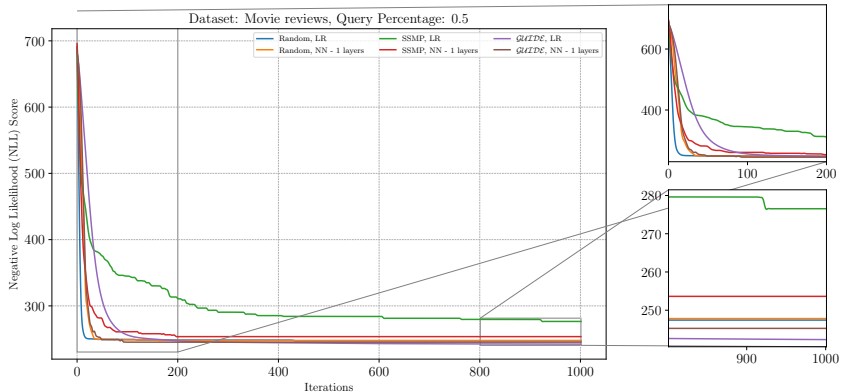

Figure 23: Analysis of ITSELF Loss Across Various Pre-Trained Models for NAMs on the Movie reviews Dataset at a Query Ratio of 0.5.

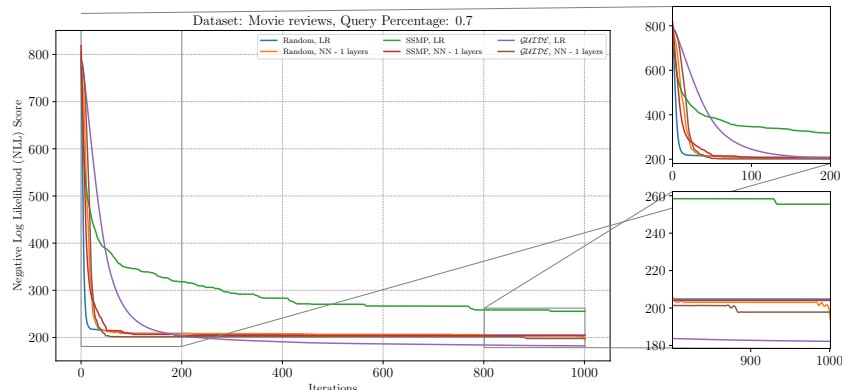

Figure 24: Analysis of ITSELF Loss Across Various Pre-Trained Models for NAMs on the Movie reviews Dataset at a Query Ratio of 0.7.

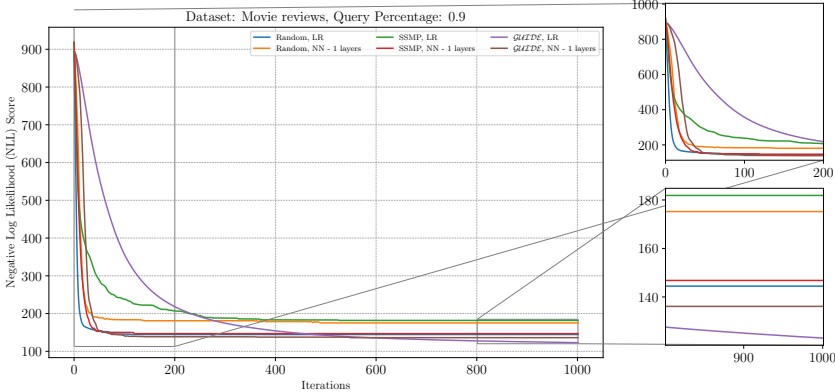

Figure 25: Analysis of ITSELF Loss Across Various Pre-Trained Models for NAMs on the Movie reviews Dataset at a Query Ratio of 0.9.

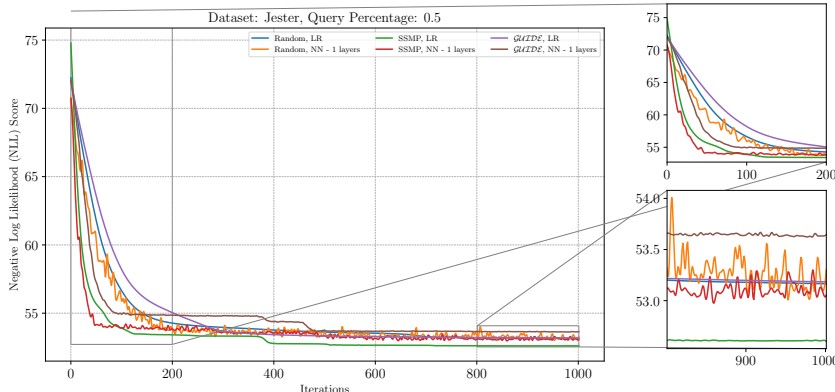

Figure 26: Analysis of ITSELF Loss Across Various Pre-Trained Models for NAMs on the Jester Dataset at a Query Ratio of 0.5.

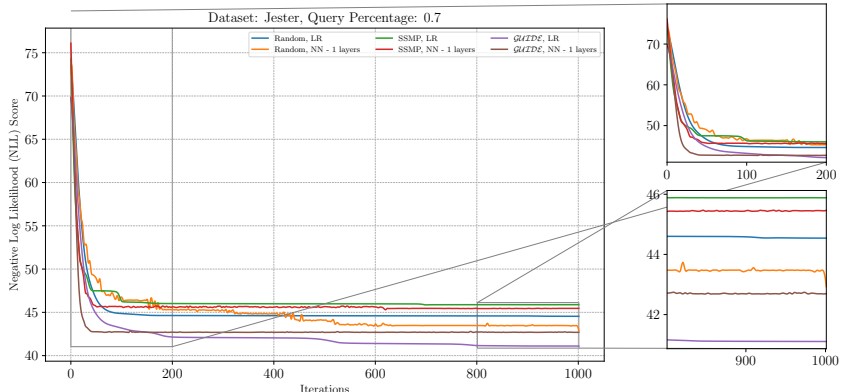

Figure 27: Analysis of ITSELF Loss Across Various Pre-Trained Models for NAMs on the Jester Dataset at a Query Ratio of 0.7.

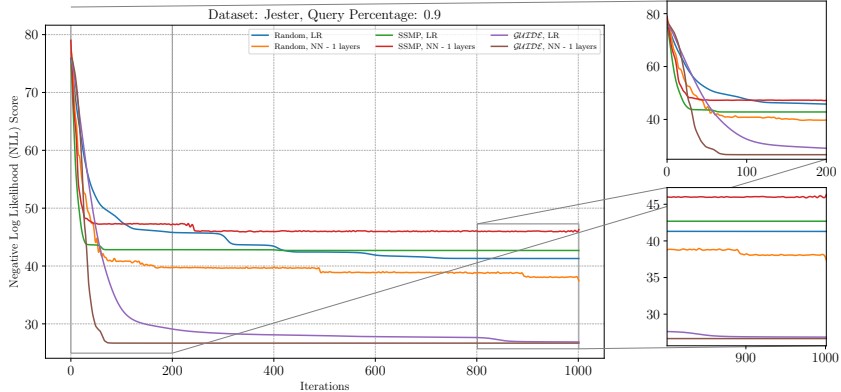

Figure 28: Analysis of ITSELF Loss Across Various Pre-Trained Models for NAMs on the Jester Dataset at a Query Ratio of 0.9.

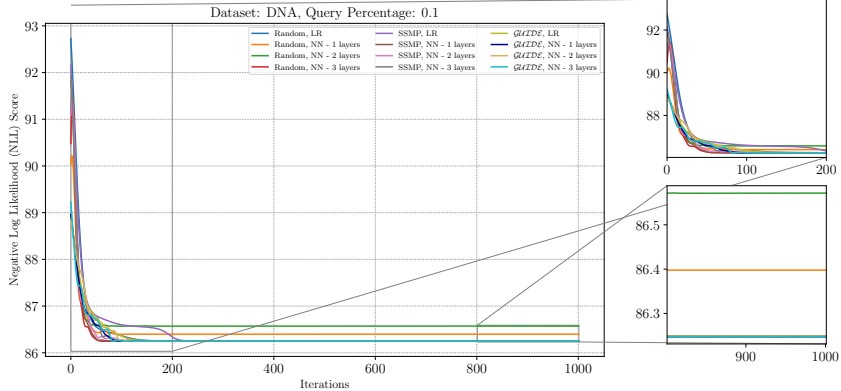

Figure 29: Analysis of ITSELF Loss Across Various Pre-Trained Models for PCs on the DNA Dataset at a Query Ratio of 0.1.

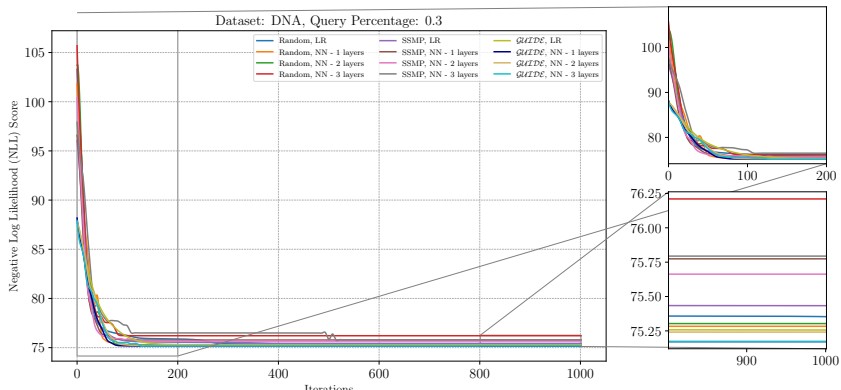

Figure 30: Analysis of ITSELF Loss Across Various Pre-Trained Models for PCs on the DNA Dataset at a Query Ratio of 0.3.

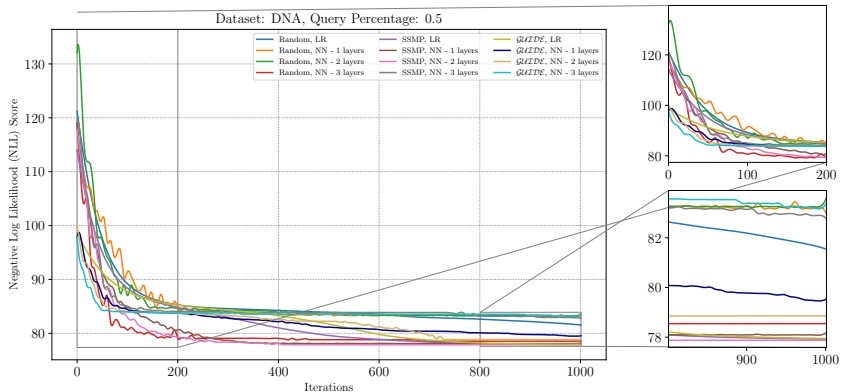

Figure 31: Analysis of ITSELF Loss Across Various Pre-Trained Models for PCs on the DNA Dataset at a Query Ratio of 0.5.

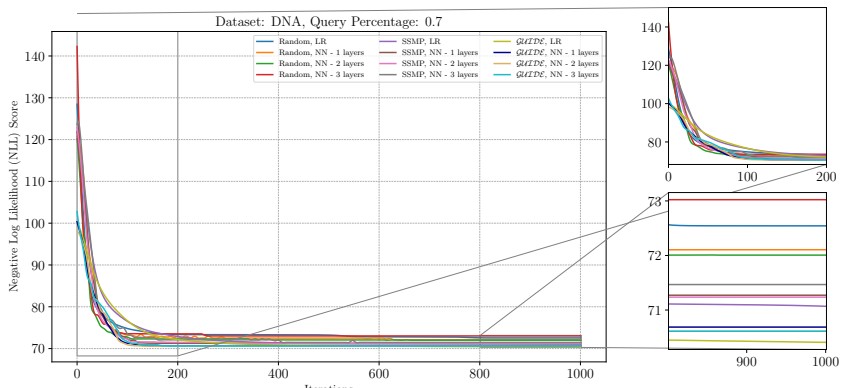

Figure 32: Analysis of ITSELF Loss Across Various Pre-Trained Models for PCs on the DNA Dataset at a Query Ratio of 0.7.

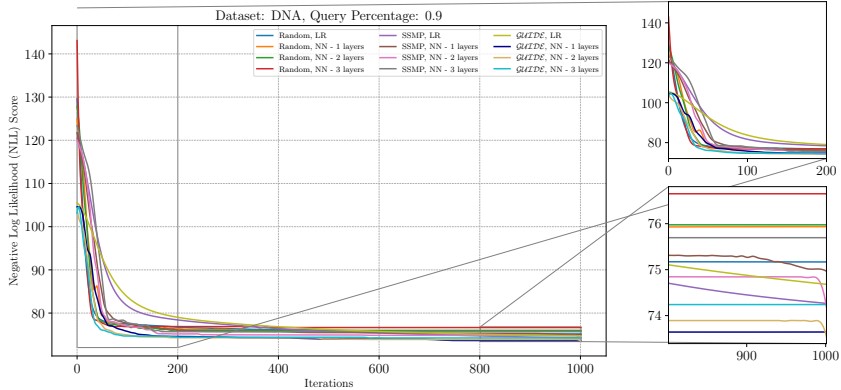

Figure 33: Analysis of ITSELF Loss Across Various Pre-Trained Models for PCs on the DNA Dataset at a Query Ratio of 0.9.

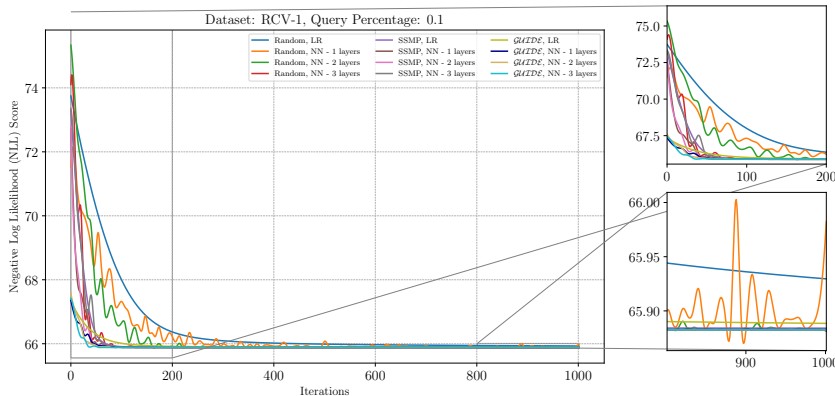

Figure 34: Analysis of ITSELF Loss Across Various Pre-Trained Models for PCs on the RCV-1 Dataset at a Query Ratio of 0.1.

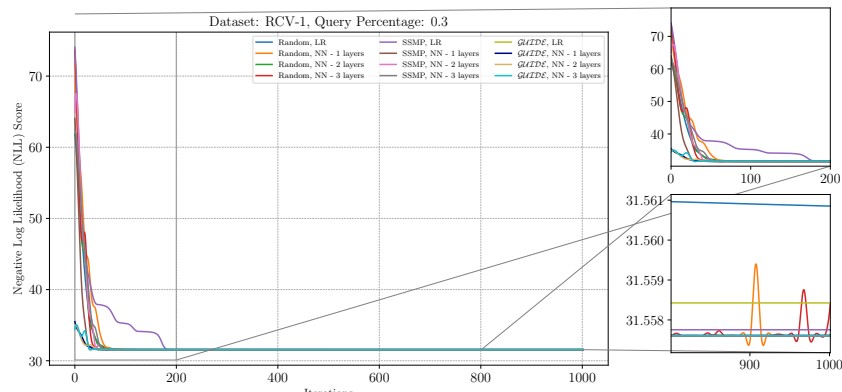

Figure 35: Analysis of ITSELF Loss Across Various Pre-Trained Models for PCs on the RCV-1 Dataset at a Query Ratio of 0.3.

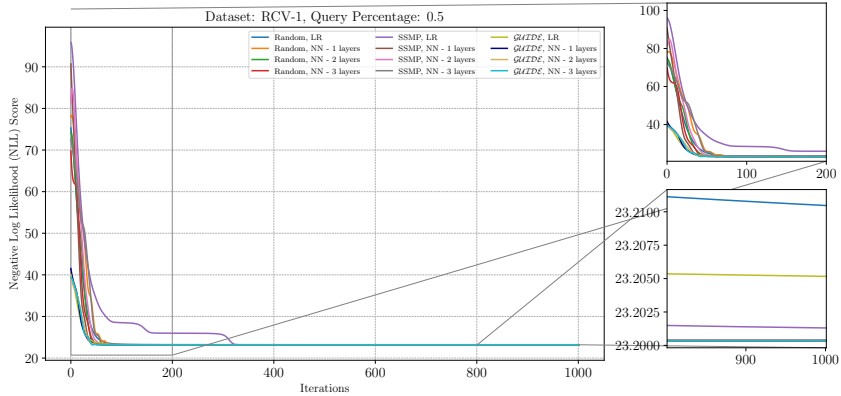

Figure 36: Analysis of ITSELF Loss Across Various Pre-Trained Models for PCs on the RCV-1 Dataset at a Query Ratio of 0.5.

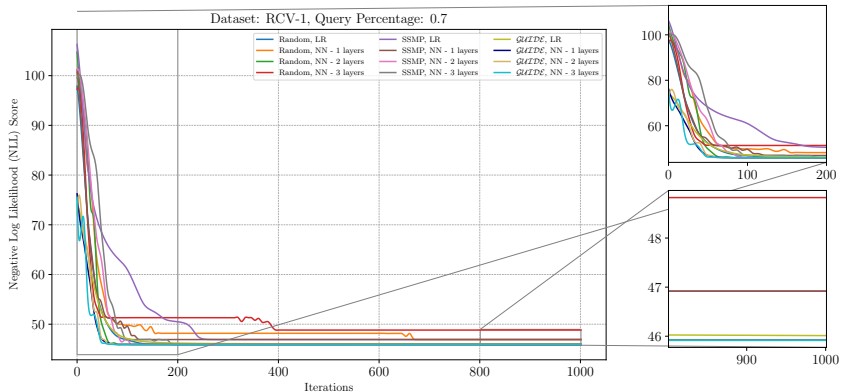

Figure 37: Analysis of ITSELF Loss Across Various Pre-Trained Models for PCs on the RCV-1 Dataset at a Query Ratio of 0.7.

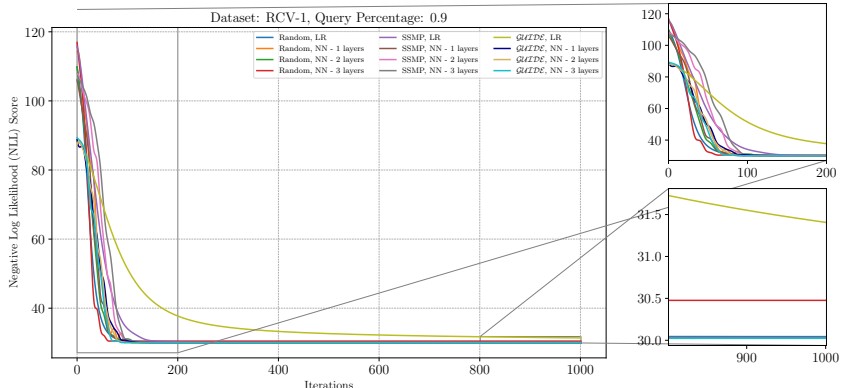

Figure 38: Analysis of ITSELF Loss Across Various Pre-Trained Models for PCs on the RCV-1 Dataset at a Query Ratio of 0.9.

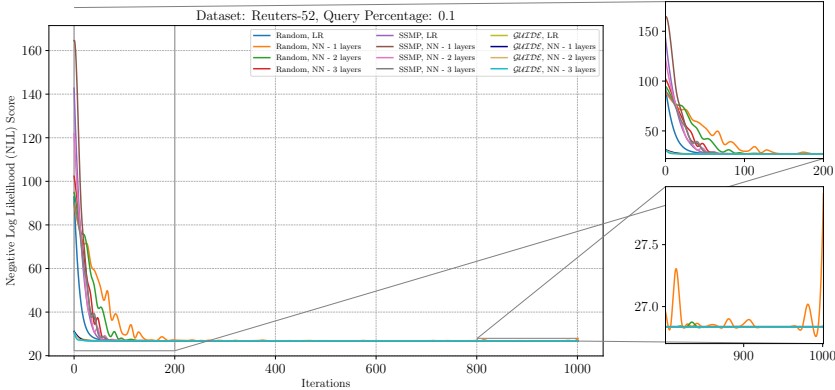

Figure 39: Analysis of ITSELF Loss Across Various Pre-Trained Models for PCs on the Reuters-52 Dataset at a Query Ratio of 0.1.

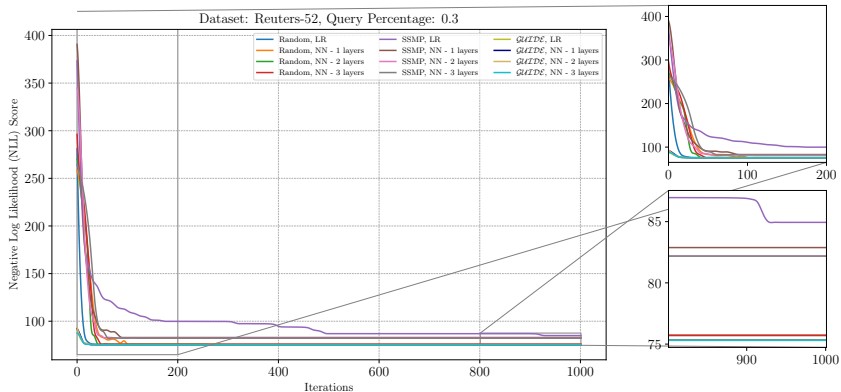

Figure 40: Analysis of ITSELF Loss Across Various Pre-Trained Models for PCs on the Reuters-52 Dataset at a Query Ratio of 0.3.

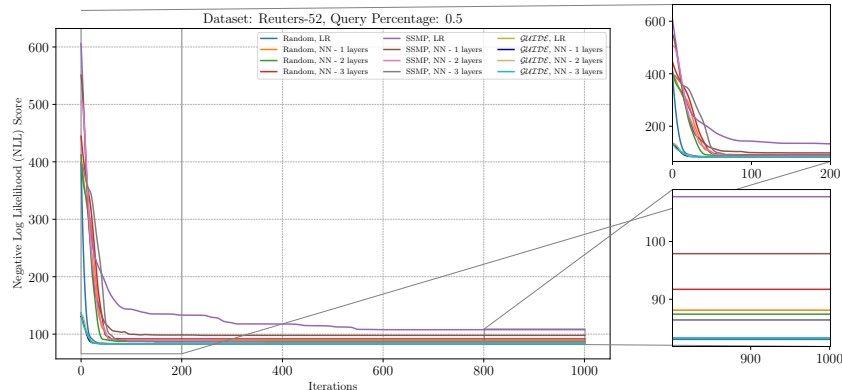

Figure 41: Analysis of ITSELF Loss Across Various Pre-Trained Models for PCs on the Reuters-52 Dataset at a Query Ratio of 0.5.

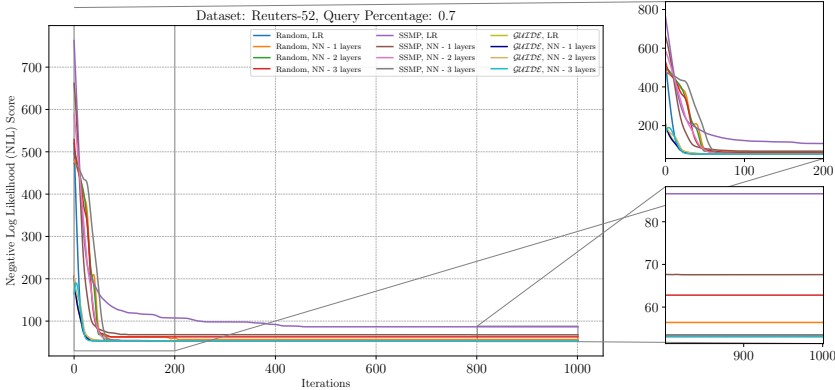

Figure 42: Analysis of ITSELF Loss Across Various Pre-Trained Models for PCs on the Reuters-52 Dataset at a Query Ratio of 0.7.

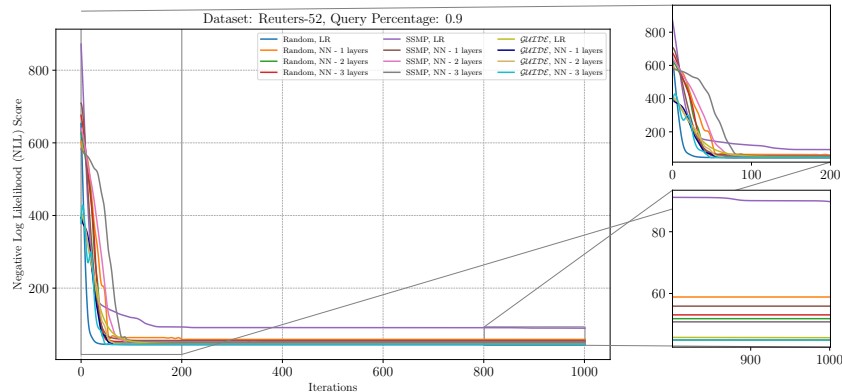

Figure 43: Analysis of ITSELF Loss Across Various Pre-Trained Models for PCs on the Reuters-52 Dataset at a Query Ratio of 0.9.

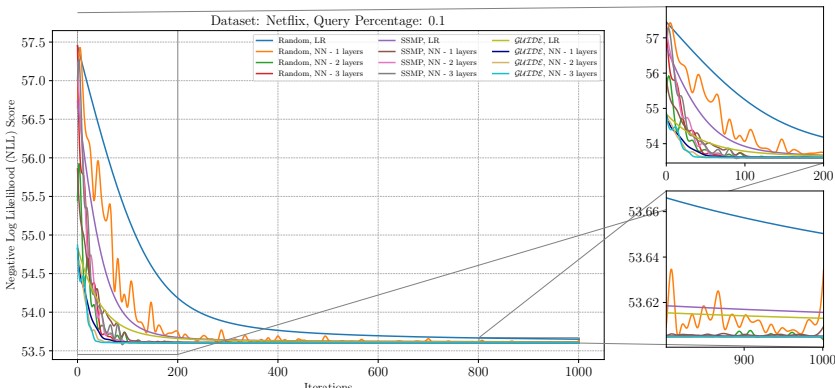

Figure 44: Analysis of ITSELF Loss Across Various Pre-Trained Models for PCs on the Netflix Dataset at a Query Ratio of 0.1.

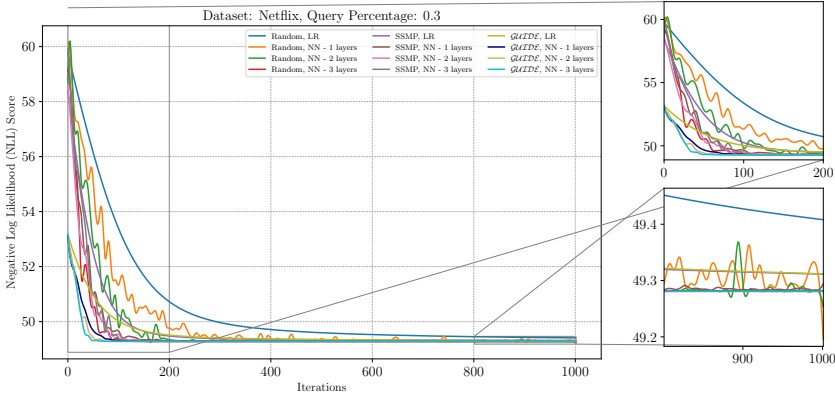

Figure 45: Analysis of ITSELF Loss Across Various Pre-Trained Models for PCs on the Netflix Dataset at a Query Ratio of 0.3.

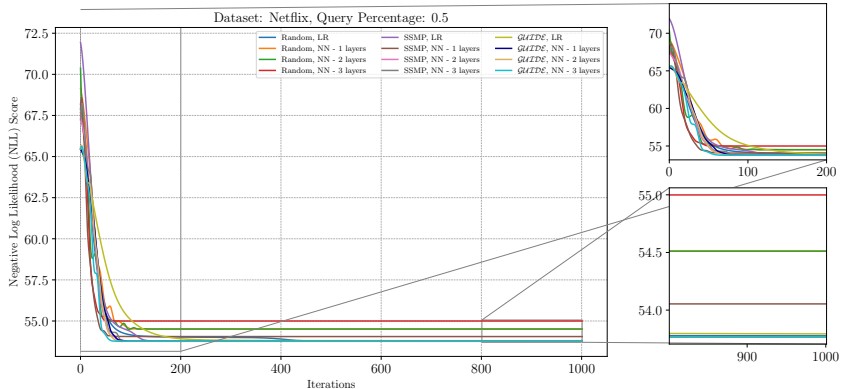

Figure 46: Analysis of ITSELF Loss Across Various Pre-Trained Models for PCs on the Netflix Dataset at a Query Ratio of 0.5.

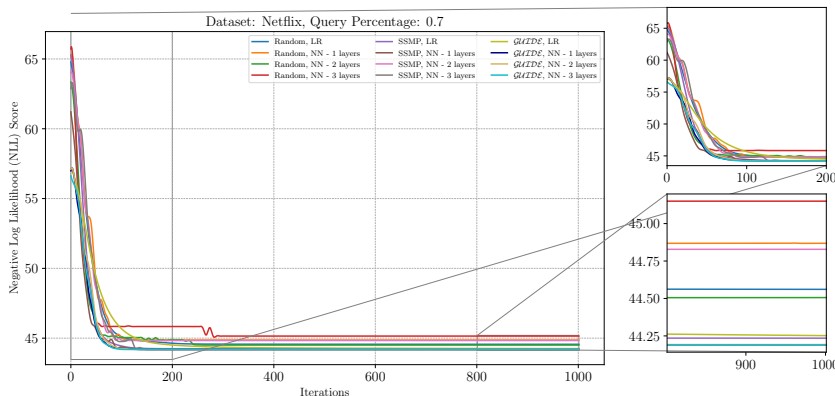

Figure 47: Analysis of ITSELF Loss Across Various Pre-Trained Models for PCs on the Netflix Dataset at a Query Ratio of 0.7.

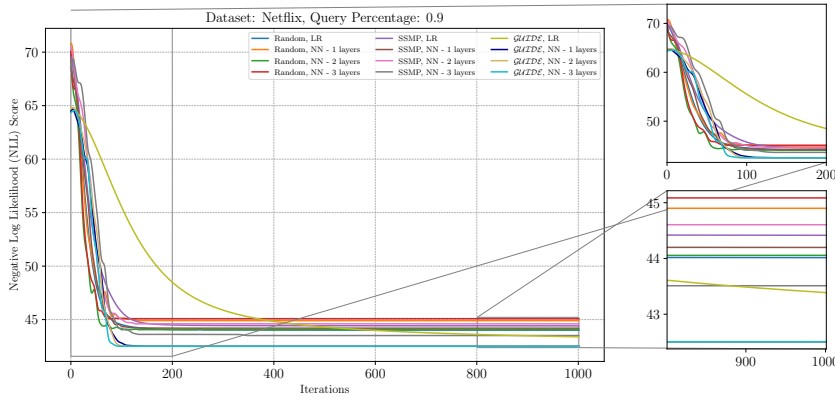

Figure 48: Analysis of ITSELF Loss Across Various Pre-Trained Models for PCs on the Netflix Dataset at a Query Ratio of 0.9.

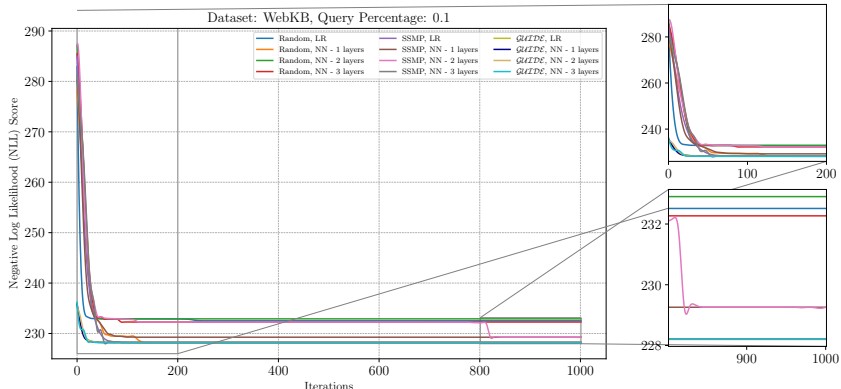

Figure 49: Analysis of ITSELF Loss Across Various Pre-Trained Models for PCs on the WebKB Dataset at a Query Ratio of 0.1.

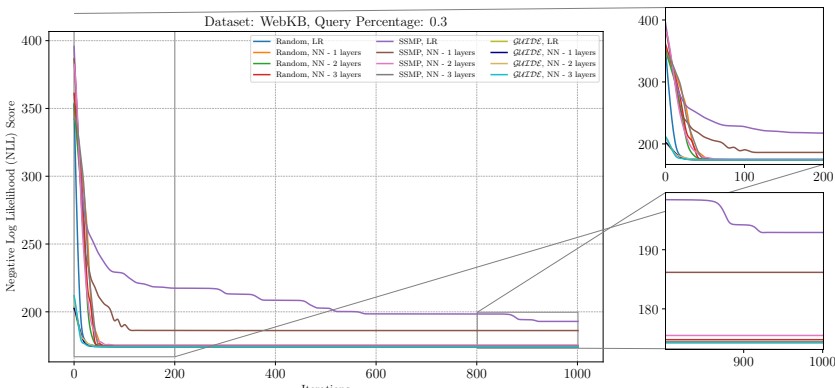

Figure 50: Analysis of ITSELF Loss Across Various Pre-Trained Models for PCs on the WebKB Dataset at a Query Ratio of 0.3.

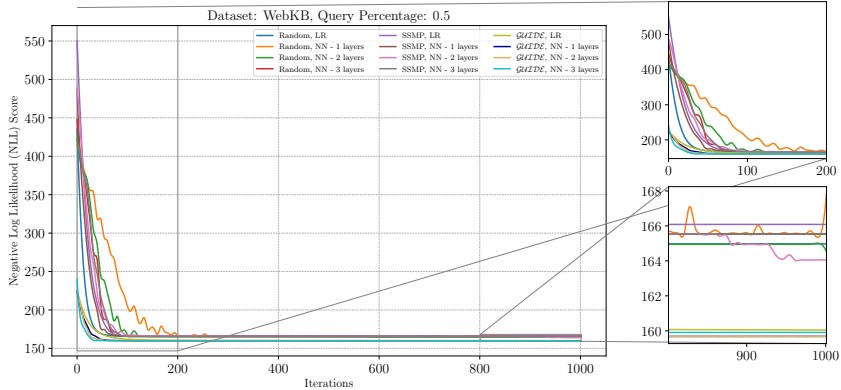

Figure 51: Analysis of ITSELF Loss Across Various Pre-Trained Models for PCs on the WebKB Dataset at a Query Ratio of 0.5.

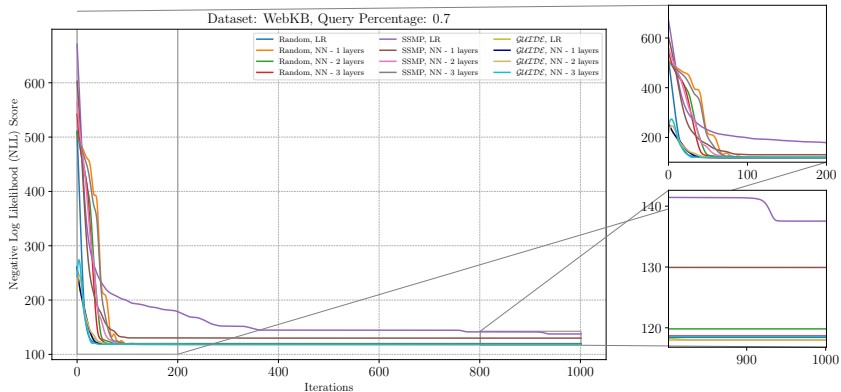

Figure 52: Analysis of ITSELF Loss Across Various Pre-Trained Models for PCs on the WebKB Dataset at a Query Ratio of 0.7.

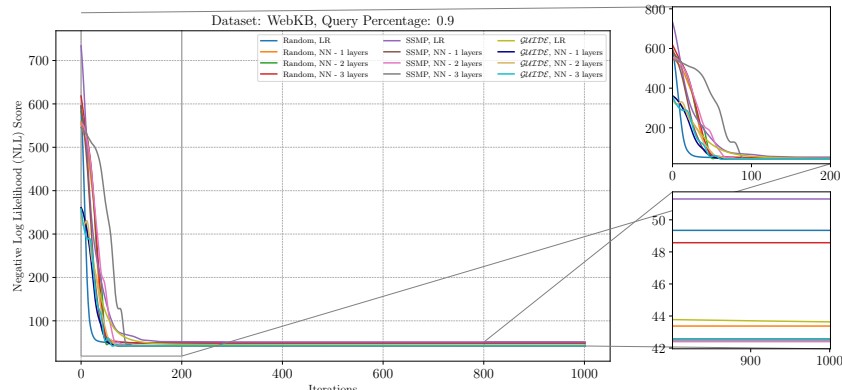

Figure 53: Analysis of ITSELF Loss Across Various Pre-Trained Models for PCs on the WebKB Dataset at a Query Ratio of 0.9.

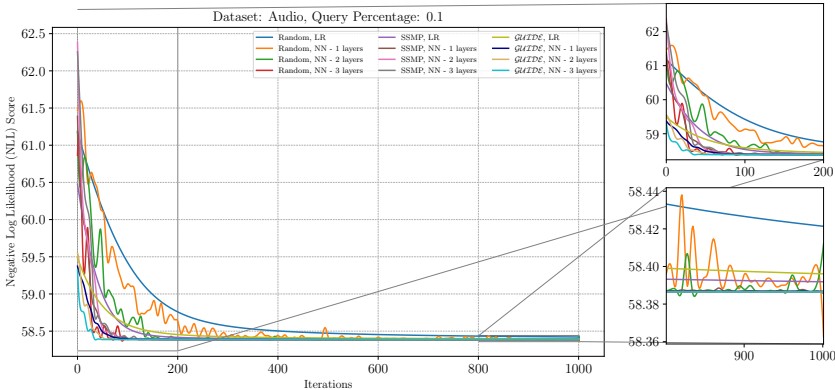

Figure 54: Analysis of ITSELF Loss Across Various Pre-Trained Models for PCs on the Audio Dataset at a Query Ratio of 0.1.

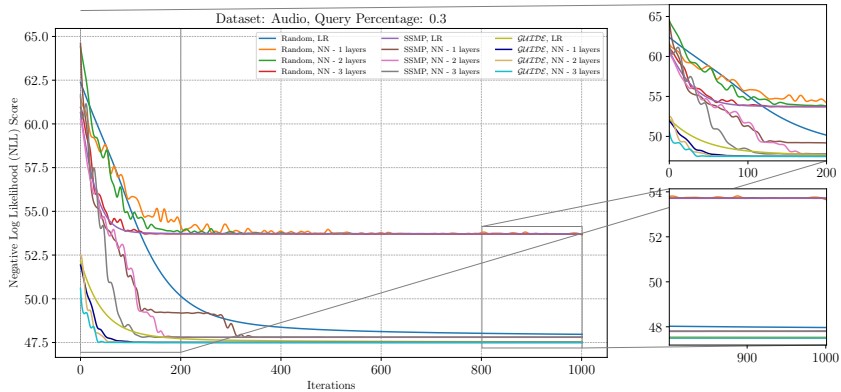

Figure 55: Analysis of ITSELF Loss Across Various Pre-Trained Models for PCs on the Audio Dataset at a Query Ratio of 0.3.

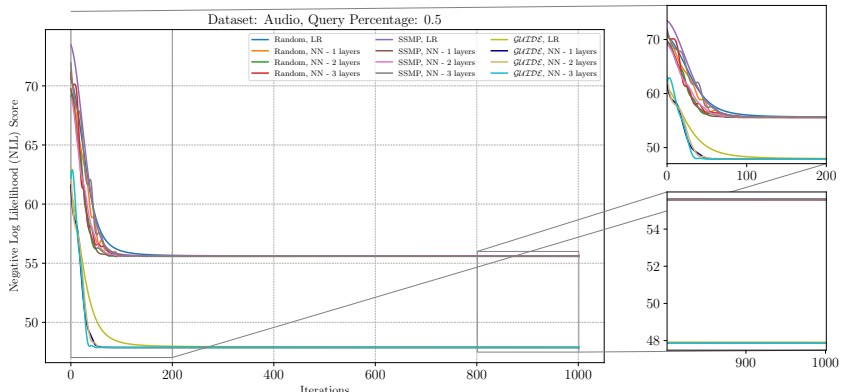

Figure 56: Analysis of ITSELF Loss Across Various Pre-Trained Models for PCs on the Audio Dataset at a Query Ratio of 0.5.

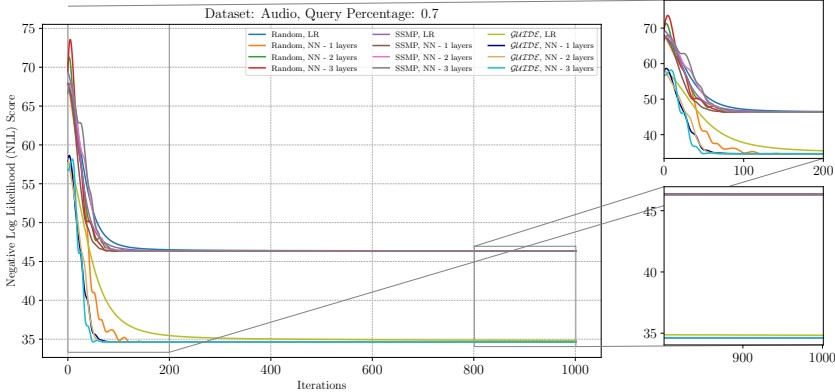

Figure 57: Analysis of ITSELF Loss Across Various Pre-Trained Models for PCs on the Audio Dataset at a Query Ratio of 0.7.

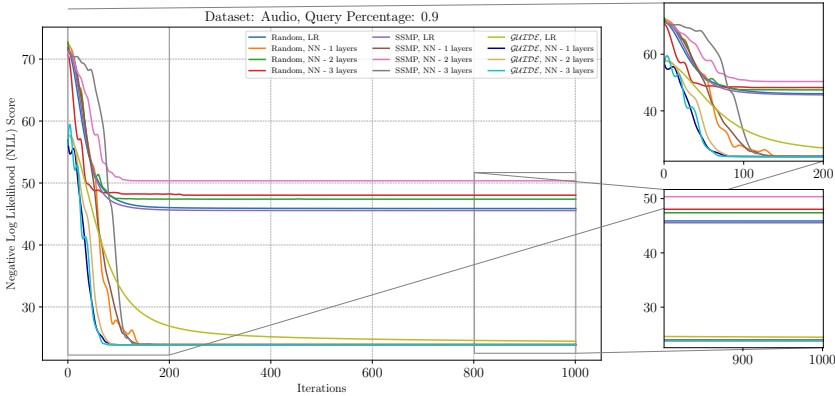

Figure 58: Analysis of ITSELF Loss Across Various Pre-Trained Models for PCs on the Audio Dataset at a Query Ratio of 0.9.

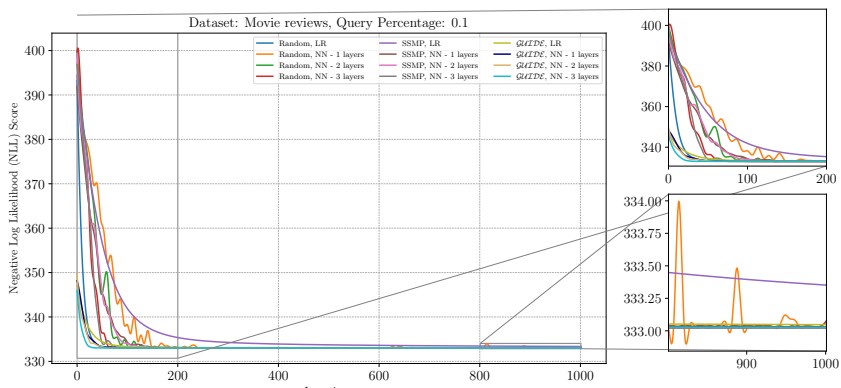

Figure 59: Analysis of ITSELF Loss Across Various Pre-Trained Models for PCs on the Movie reviews Dataset at a Query Ratio of 0.1.

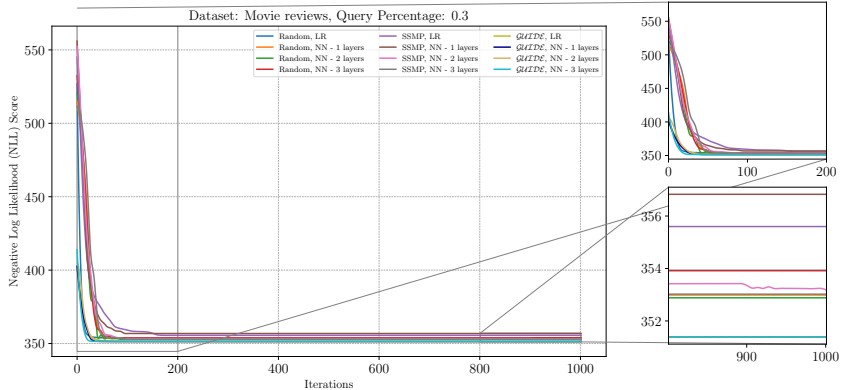

Figure 60: Analysis of ITSELF Loss Across Various Pre-Trained Models for PCs on the Movie reviews Dataset at a Query Ratio of 0.3.

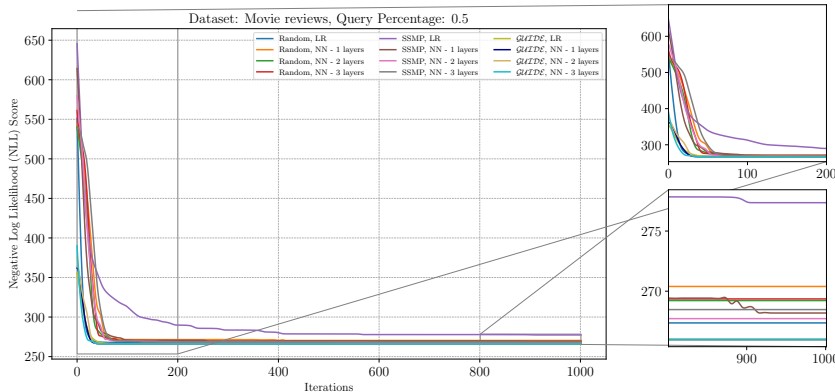

Figure 61: Analysis of ITSELF Loss Across Various Pre-Trained Models for PCs on the Movie reviews Dataset at a Query Ratio of 0.5.

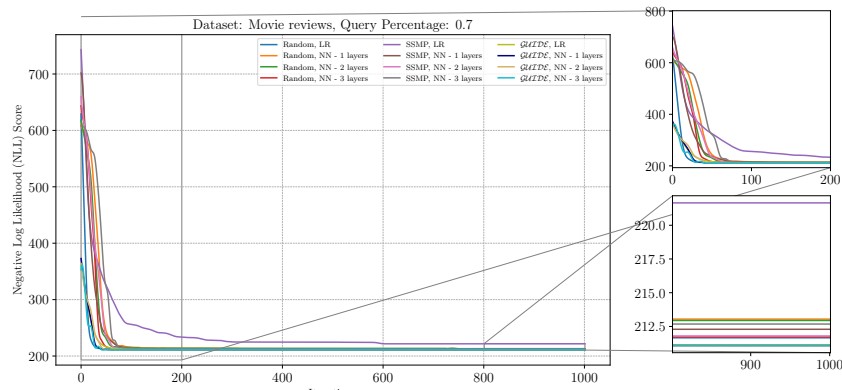

Figure 62: Analysis of ITSELF Loss Across Various Pre-Trained Models for PCs on the Movie reviews Dataset at a Query Ratio of 0.7.

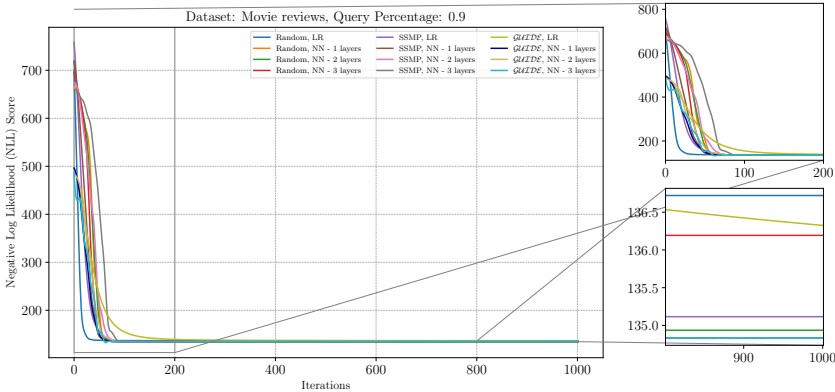

Figure 63: Analysis of ITSELF Loss Across Various Pre-Trained Models for PCs on the Movie reviews Dataset at a Query Ratio of 0.9.

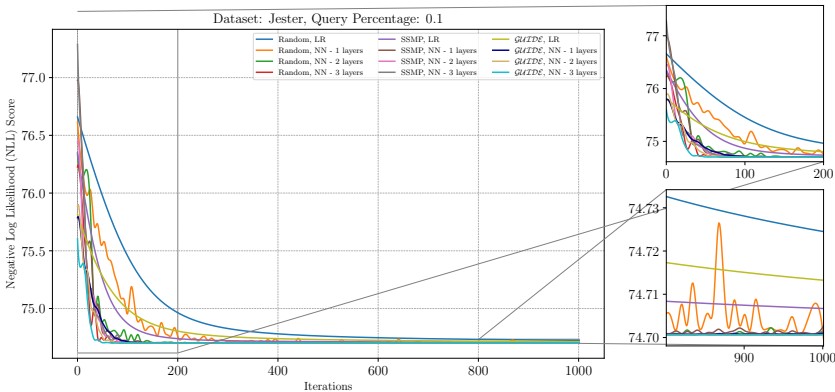

Figure 64: Analysis of ITSELF Loss Across Various Pre-Trained Models for PCs on the Jester Dataset at a Query Ratio of 0.1.

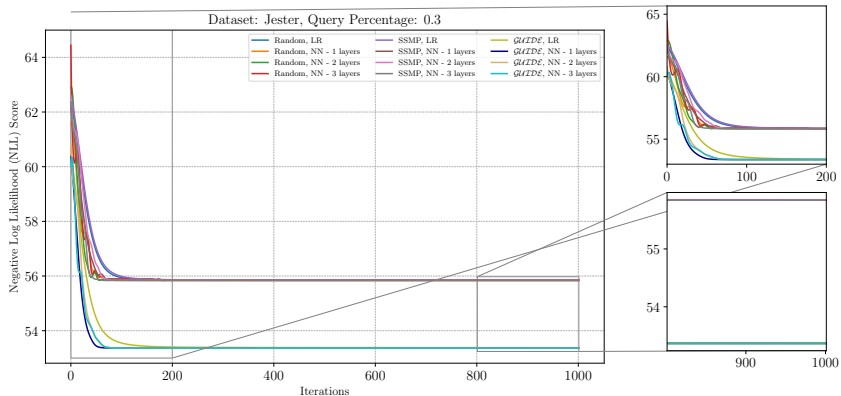

Figure 65: Analysis of ITSELF Loss Across Various Pre-Trained Models for PCs on the Jester Dataset at a Query Ratio of 0.3.

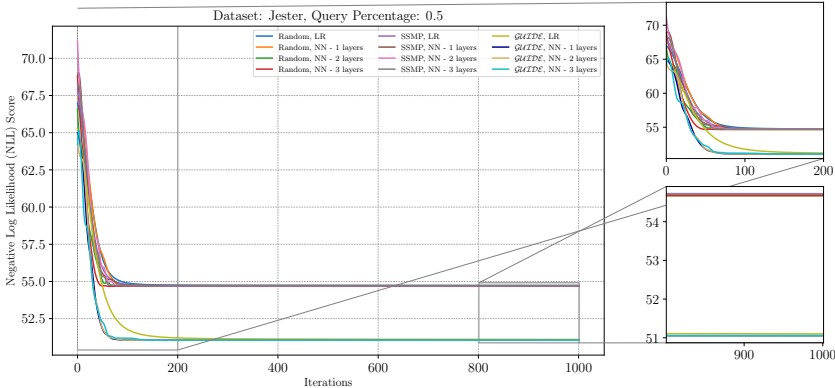

Figure 66: Analysis of ITSELF Loss Across Various Pre-Trained Models for PCs on the Jester Dataset at a Query Ratio of 0.5.

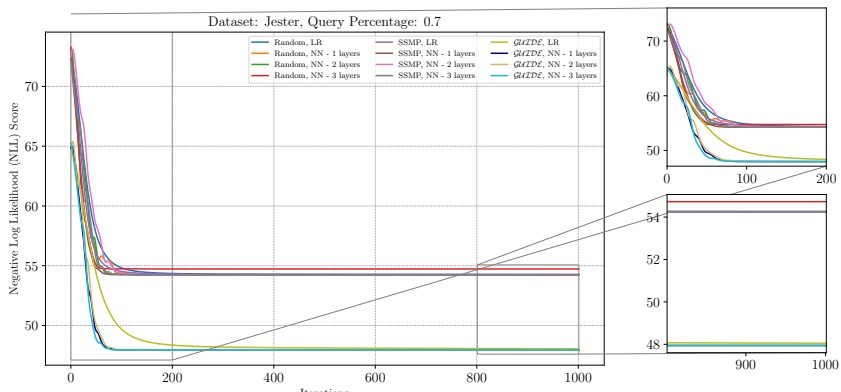

Figure 67: Analysis of ITSELF Loss Across Various Pre-Trained Models for PCs on the Jester Dataset at a Query Ratio of 0.7.

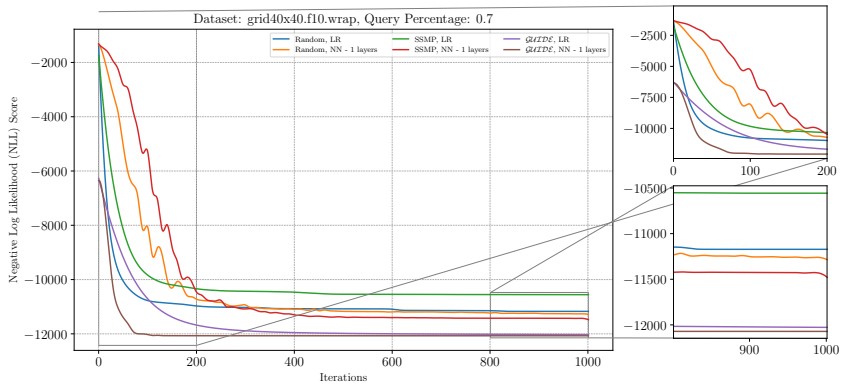

Figure 68: Analysis of ITSELF Loss Across Various Pre-Trained Models for PGMs on the grid40x40.f10.wrap Dataset at a Query Ratio of 0.7.

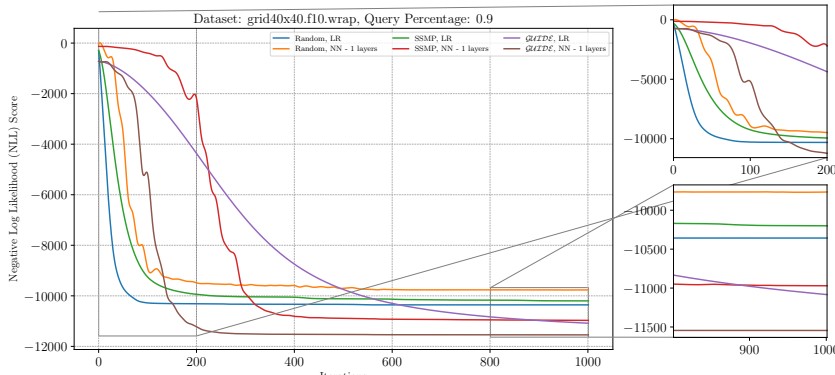

Figure 69: Analysis of ITSELF Loss Across Various Pre-Trained Models for PGMs on the grid40x40.f10.wrap Dataset at a Query Ratio of 0.9.

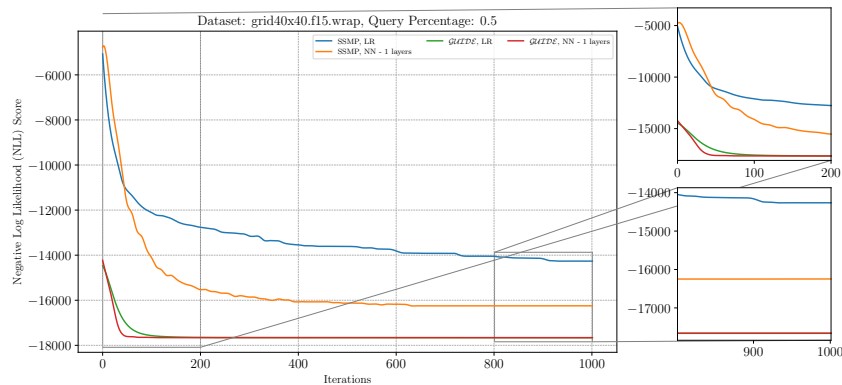

Figure 70: Analysis of ITSELF Loss Across Various Pre-Trained Models for PGMs on the grid40x40.f15.wrap Dataset at a Query Ratio of 0.5.

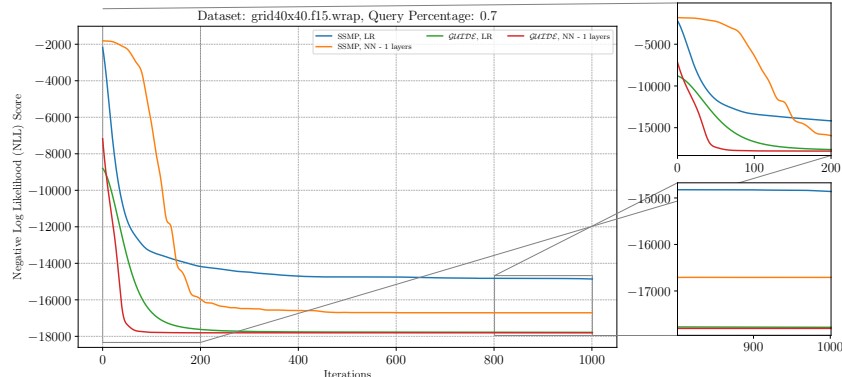

Figure 71: Analysis of ITSELF Loss Across Various Pre-Trained Models for PGMs on the grid40x40.f15.wrap Dataset at a Query Ratio of 0.7.

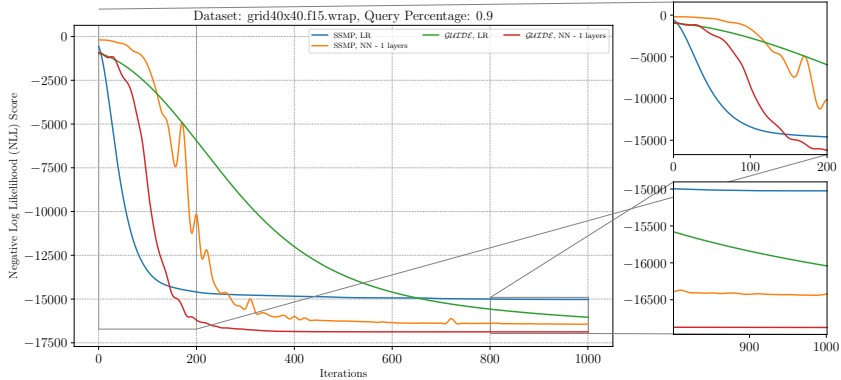

Figure 72: Analysis of ITSELF Loss Across Various Pre-Trained Models for PGMs on the grid40x40.f15.wrap Dataset at a Query Ratio of 0.9.

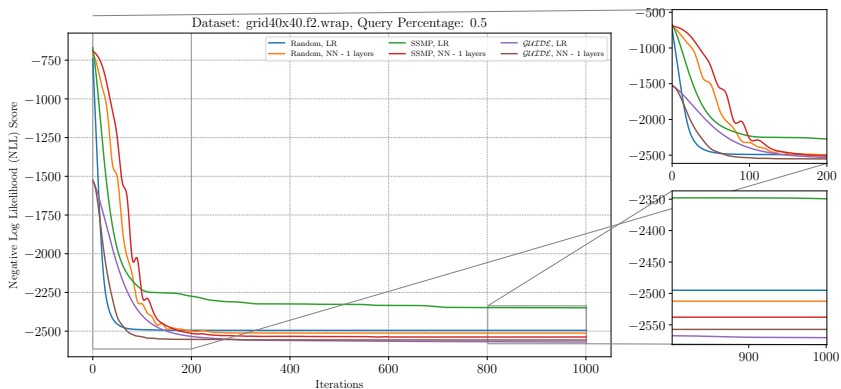

Figure 73: Analysis of ITSELF Loss Across Various Pre-Trained Models for PGMs on the grid40x40.f2.wrap Dataset at a Query Ratio of 0.5.

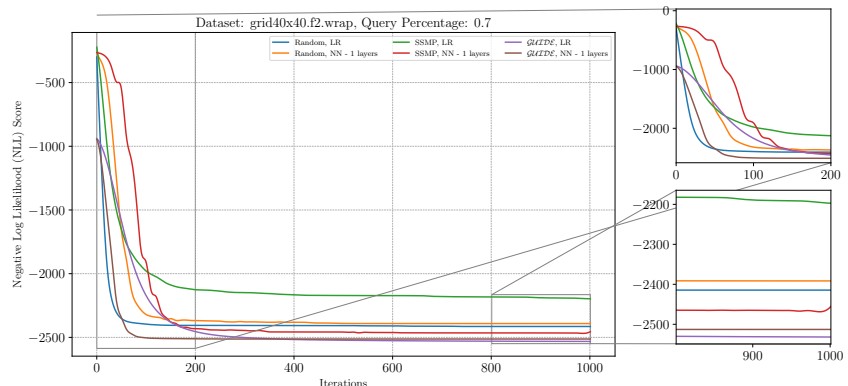

Figure 74: Analysis of ITSELF Loss Across Various Pre-Trained Models for PGMs on the grid40x40.f2.wrap Dataset at a Query Ratio of 0.7.

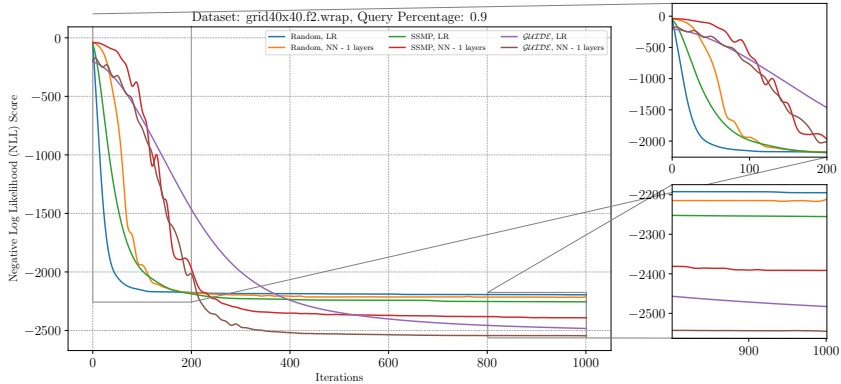

Figure 75: Analysis of ITSELF Loss Across Various Pre-Trained Models for PGMs on the grid40x40.f2.wrap Dataset at a Query Ratio of 0.9.

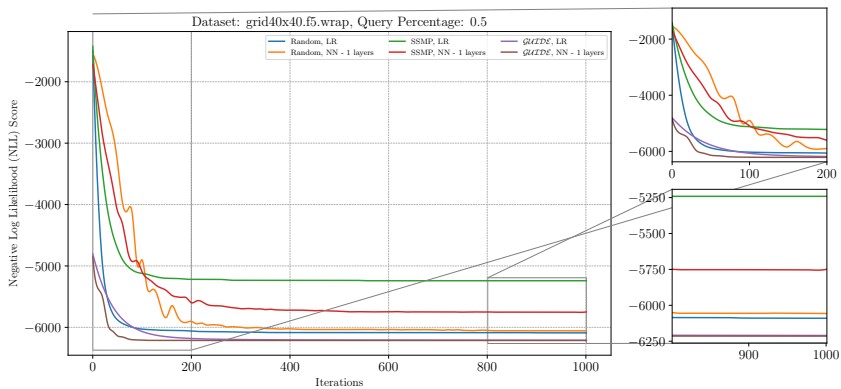

Figure 76: Analysis of ITSELF Loss Across Various Pre-Trained Models for PGMs on the grid40x40.f5.wrap Dataset at a Query Ratio of 0.5.

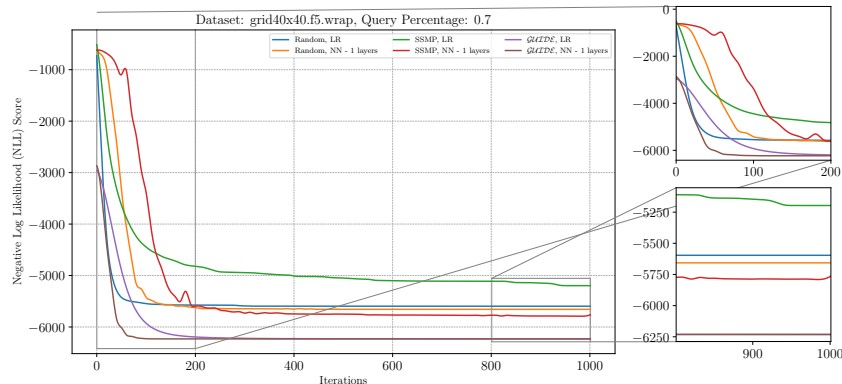

Figure 77: Analysis of ITSELF Loss Across Various Pre-Trained Models for PGMs on the grid40x40.f5.wrap Dataset at a Query Ratio of 0.7.

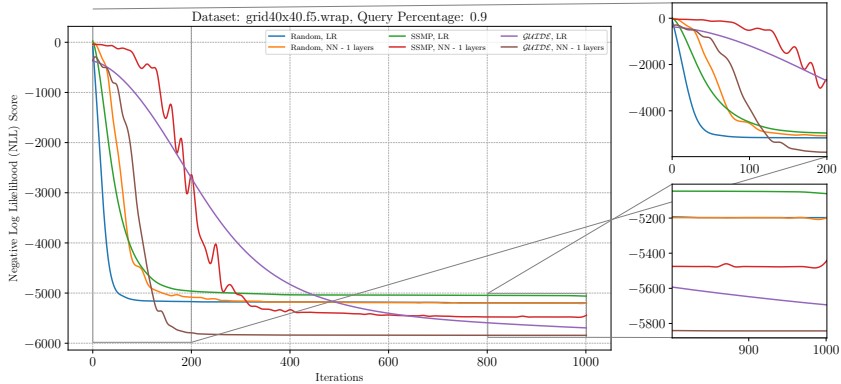

Figure 78: Analysis of ITSELF Loss Across Various Pre-Trained Models for PGMs on the grid40x40.f5.wrap Dataset at a Query Ratio of 0.9.

# D Inference Time Comparison

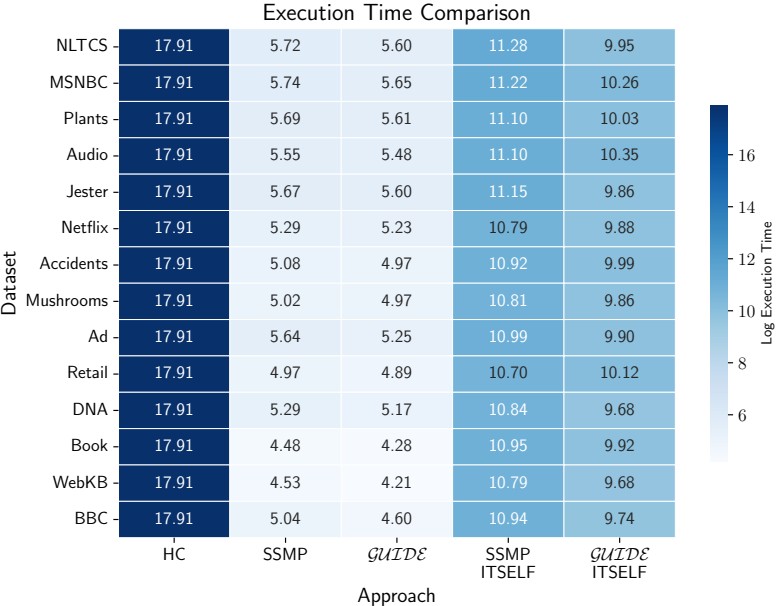

Figure 79: Heatmap depicting the inference time for MADE on a logarithmic microsecond scale, where a lighter color denotes shorter (more favorable) durations.

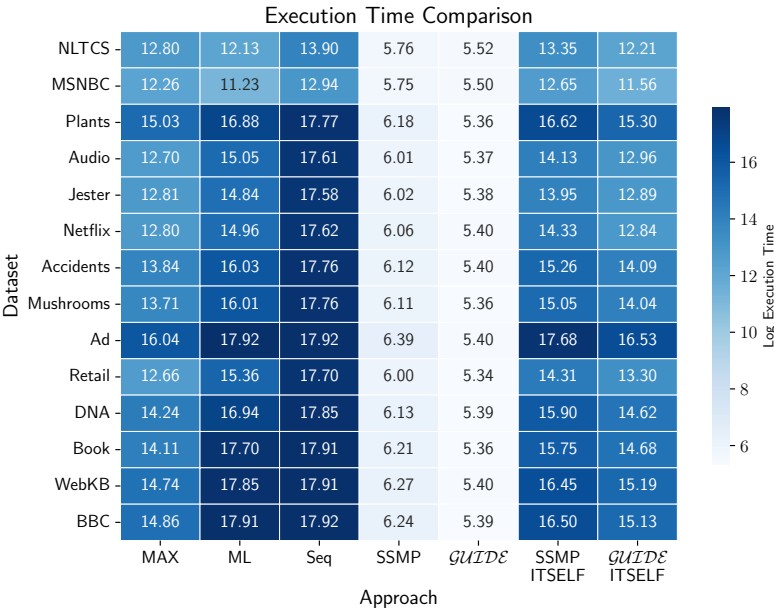

Figure 80: Heatmap depicting the inference time for PC on a logarithmic microsecond scale, where a lighter color denotes shorter (more favorable) durations.

We present the inference times for all baselines and proposed methods in Figures 79 to 81. Figure 79 details the inference times for MADE, while Figures 80 and 81 respectively illustrate the times for PCs and PGMs. This comparison facilitates a direct evaluation of the computational efficiency across different models.

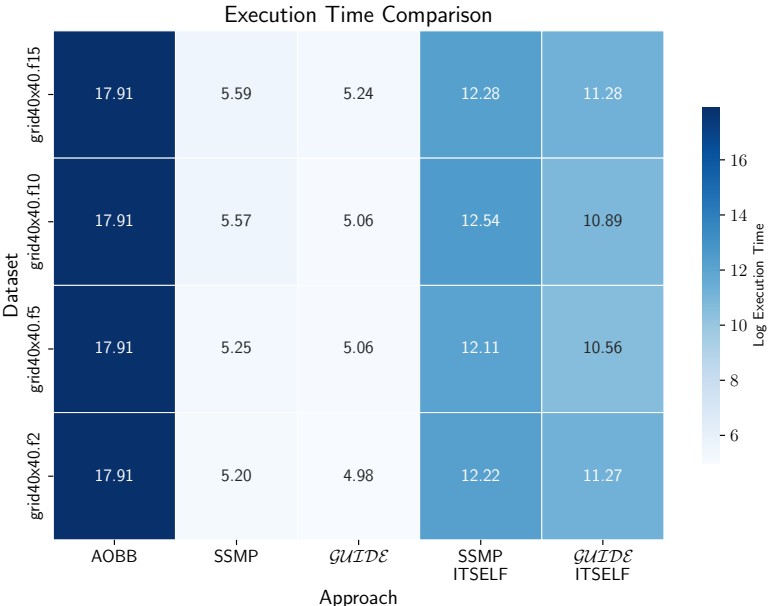

Figure 81: Heatmap depicting the inference time for PGM on a logarithmic microsecond scale, where a lighter color denotes shorter (more favorable) durations.

Each cell displays the natural logarithm of the time, measured in microseconds, for each method and dataset. Lighter colors indicate lower values. Notably, inferences using SSMP and $\mathcal{GUIDE}$ require the shortest time, as these methods necessitate only a single forward pass through the neural network to obtain the values for the query variables.

For MADE, the subsequent fastest method employs a model trained with $\mathcal{GUIDE}$ and conducts inference using ITSELF, outperforming the approach that uses SSMP for training. This advantage stems from the reduced number of ITSELF iterations required by $\mathcal{GUIDE}$, benefiting from a more effectively trained model. In PGMs, a similar pattern emerges with $\mathcal{GUIDE}$ + ITSELF as the next fastest method, followed by SSMP + ITSELF. For PCs, MAX ranks as the next fastest, closely followed by the $\mathcal{GUIDE}$ + ITSELF and SSMP + ITSELF methods. Finally, the ML and Seq methods display the highest inference times.

Thus, if you require a highly efficient method capable of performing inference in a fraction of a millisecond, $\mathcal{GUIDE}$ is the optimal choice. It outperforms the baseline for both MADE and PGMs. However, if higher log-likelihood scores are necessary, $\mathcal{GUIDE}$ + ITSELF would be suitable, as it generally surpasses the baselines in speed and performance across various scenarios.

## E    Gap Analysis For PGM

Table 2 presents the log-likelihood score gap between the neural network methods (SSMP, $\mathcal{GUIDE}$, SSMP + ITSELF, $\mathcal{GUIDE}$ + ITSELF) and exact solutions. These exact solutions are obtained using AOBB, which provides near-optimal results for smaller datasets. For each approach M, the gap is calculated as the relative difference between the score of the near-optimal solution (determined by AOBB) and the score achieved by M. This approach is feasible due to the use of small datasets, allowing identification of exact solutions.

The final column highlights the neural-based approach achieving the best performance for each dataset and query ratio combination. Notably, $\mathcal{GUIDE}$ and ITSELF consistently surpass other neural baselines across almost all dataset-query pairs. This analysis provides a comprehensive assessment of the proposed methods relative to exact solutions on small datasets, enabling a direct comparison of their effectiveness.

Table 2: Gap Between AOBB And Other Methods.

| Method | Query Ratio | SSMP | $\mathcal{GUIDE}$ | SSMP + ITSELF | $\mathcal{GUIDE}$ + ITSELF | Best Method |
|---|---|---|---|---|---|---|
| **Grids-17** | 0.900 | 0.082 | 0.060 | 0.069 | 0.075 | $\mathcal{GUIDE}$ |
| **Grids-17** | 0.800 | 0.051 | 0.038 | 0.040 | 0.035 | $\mathcal{GUIDE}$ + ITSELF |
| **Grids-17** | 0.700 | 0.042 | 0.030 | 0.034 | 0.016 | $\mathcal{GUIDE}$ + ITSELF |
| **Grids-17** | 0.500 | 0.026 | 0.024 | 0.024 | 0.007 | $\mathcal{GUIDE}$ + ITSELF |
| **Grids-18** | 0.900 | 0.081 | 0.062 | 0.071 | 0.102 | $\mathcal{GUIDE}$ |
| **Grids-18** | 0.700 | 0.033 | 0.027 | 0.024 | 0.015 | $\mathcal{GUIDE}$ + ITSELF |
| **Grids-18** | 0.500 | 0.020 | 0.018 | 0.018 | 0.006 | $\mathcal{GUIDE}$ + ITSELF |
| **Grids-18** | 0.800 | 0.054 | 0.035 | 0.045 | 0.037 | $\mathcal{GUIDE}$ |
| **Segmentation-14** | 0.500 | 0.032 | 0.032 | 0.032 | 0.004 | $\mathcal{GUIDE}$ + ITSELF |
| **Segmentation-14** | 0.900 | 0.045 | 0.014 | 0.014 | 0.005 | $\mathcal{GUIDE}$ + ITSELF |
| **Segmentation-14** | 0.800 | 0.051 | 0.024 | 0.024 | 0.006 | $\mathcal{GUIDE}$ + ITSELF |
| **Segmentation-14** | 0.700 | 0.029 | 0.029 | 0.029 | 0.005 | $\mathcal{GUIDE}$ + ITSELF |
| **Segmentation-15** | 0.800 | 0.046 | 0.002 | 0.002 | 0.002 | $\mathcal{GUIDE}$ + ITSELF |
| **Segmentation-15** | 0.500 | 0.003 | 0.003 | 0.003 | 0.000 | $\mathcal{GUIDE}$ + ITSELF |
| **Segmentation-15** | 0.900 | 0.675 | 0.255 | 0.433 | 0.305 | $\mathcal{GUIDE}$ |
| **Segmentation-15** | 0.700 | 0.003 | 0.003 | 0.003 | 0.002 | $\mathcal{GUIDE}$ + ITSELF |

# F  Log Likelihood Scores Comparison

This section compares log-likelihood scores across baselines, SSMP, SSMP + ITSELF, $\mathcal{GUIDE}$ and $\mathcal{GUIDE}$ + ITSELF for all datasets and PMs. The log likelihood plots for NAMs are depicted in Figures 82 to 101, while those for PCs are illustrated in Figures 102 to 121. Each bar represents the mean log likelihood score of the corresponding method, with tick marks indicating the mean ± standard deviation. Higher values in these scores signify better performance by the method, considering they represent log likelihood scores.

## F.1  Scores for NAM

Figures 82 to 101 present the log likelihood scores for NAMs, illustrating the performance of ITSELF inference and $\mathcal{GUIDE}$ training relative to other baselines. The heatmaps and contingency tables discussed in the main paper corroborate the superior performance of $\mathcal{GUIDE}$ + ITSELF. These visual representations allow for a comprehensive understanding of the performance of our methods and baseline approaches, including HC and SSMP, across various datasets and query ratios.

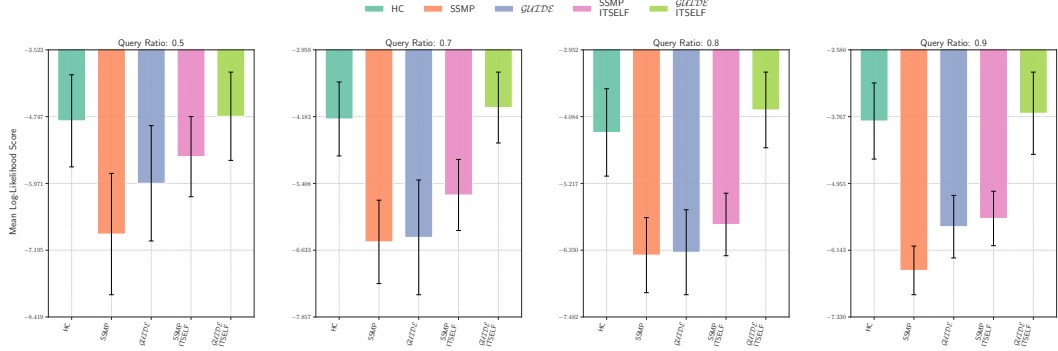

Figure 82: Log-Likelihood Scores on NLTCS for NAM. Higher Scores Indicate Better Performance.

## F.2  Scores for PCs

Analyzing Figures 102 to 121, which focuses on PCs, reveals similar patterns. The neural-based methods significantly outperform the MAX baseline. Among these, $\mathcal{GUIDE}$ + ITSELF surpasses all other polynomial-time baselines and neural methods in over 80 percent of the experiments. This demonstrates that ITSELF substantially enhances the chances of approaching optimal solutions by performing test-time optimization. When comparing traditional inference with ITSELF, ITSELF consistently proves superior. Moreover, $\mathcal{GUIDE}$ outperforms the other neural-based training methods (SSMP).

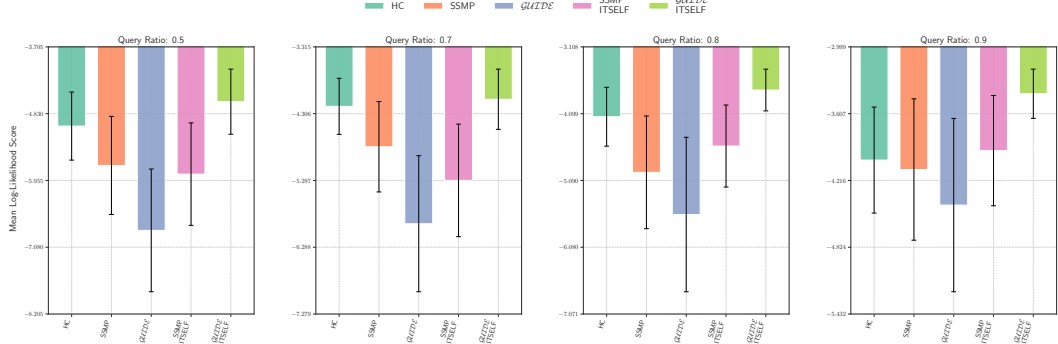

Figure 83: Log-Likelihood Scores on   for NAM. Higher Scores Indicate Better Performance.

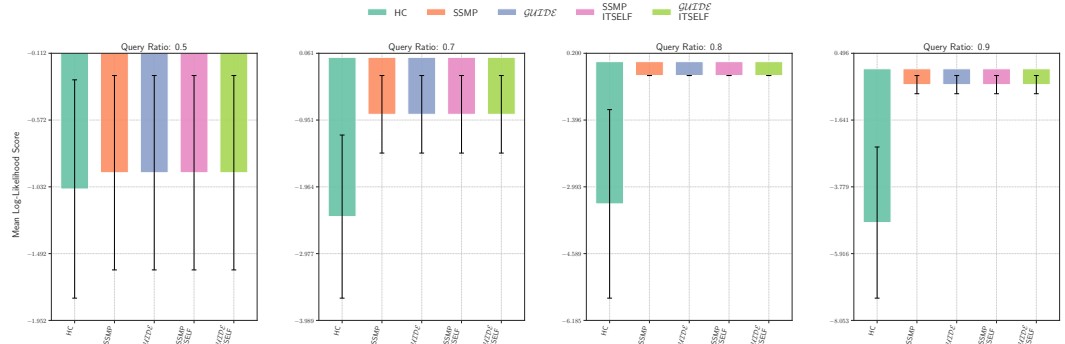

Figure 84: Log-Likelihood Scores on KDDCup2k  for NAM. Higher Scores Indicate Better Performance.

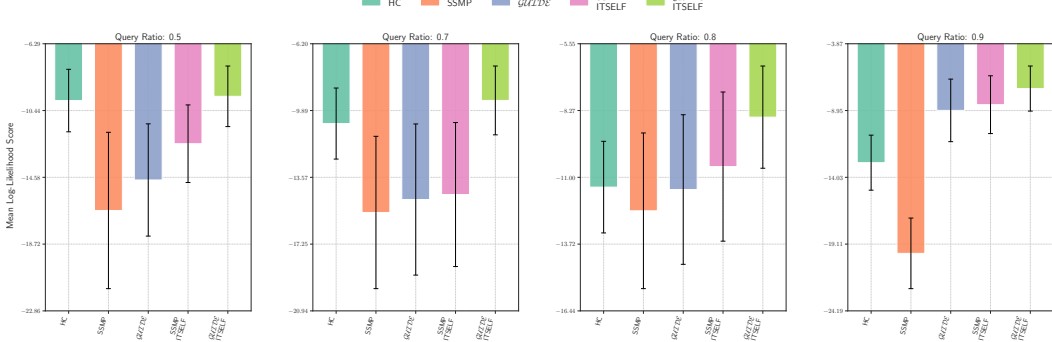

Figure 85: Log-Likelihood Scores on Plants  for NAM. Higher Scores Indicate Better Performance.

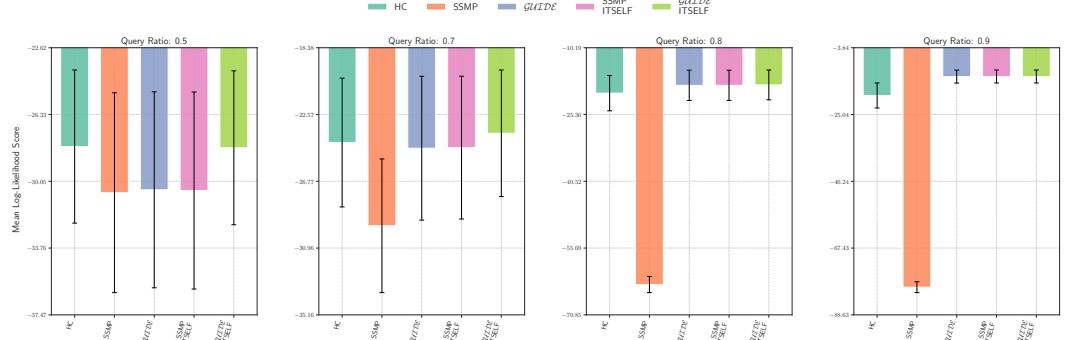

Figure 86: Log-Likelihood Scores on Audio for NAM. Higher Scores Indicate Better Performance.

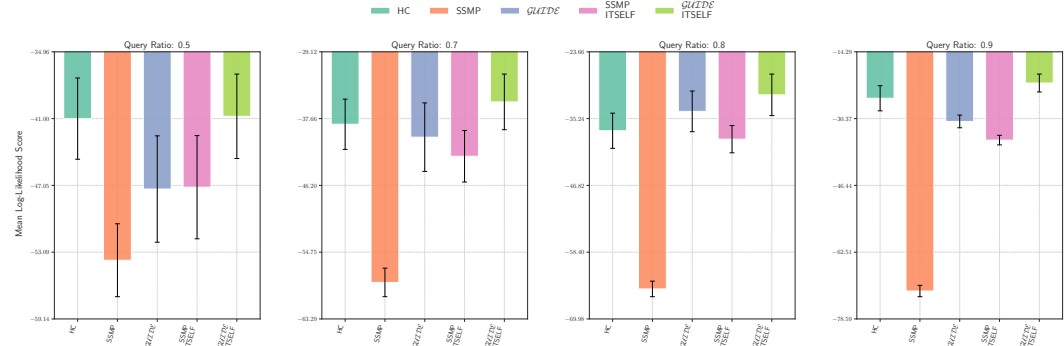

Figure 87: Log-Likelihood Scores on Jester for NAM. Higher Scores Indicate Better Performance.

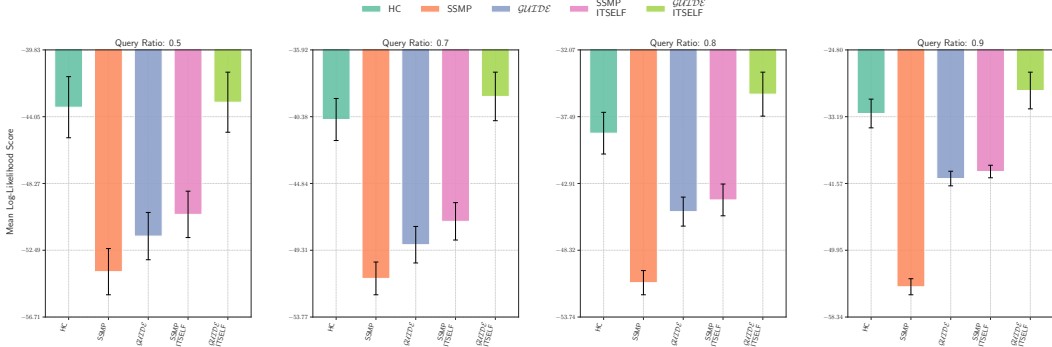

Figure 88: Log-Likelihood Scores on Netflix for NAM. Higher Scores Indicate Better Performance.

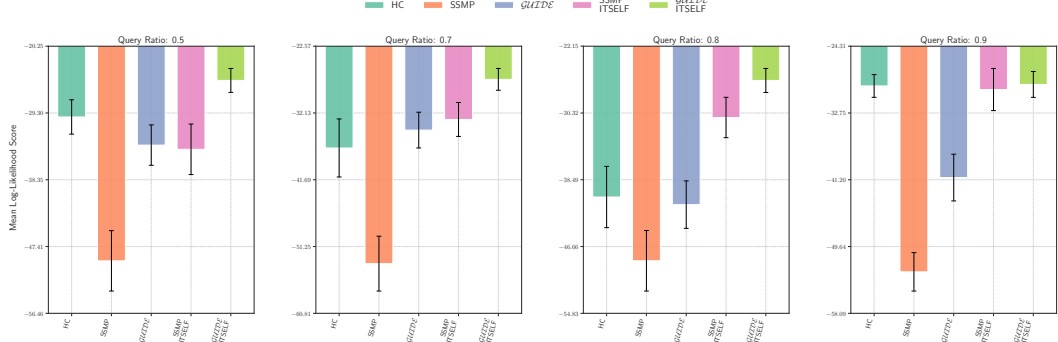

Figure 89: Log-Likelihood Scores on Accidents for NAM. Higher Scores Indicate Better Performance.

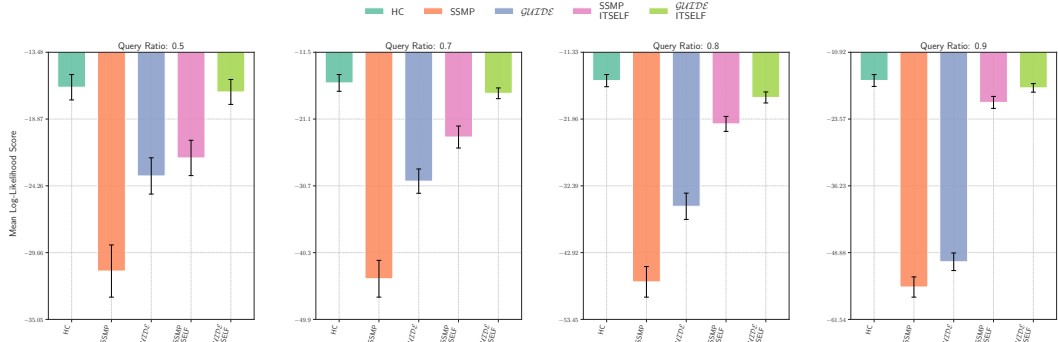

Figure 90: Log-Likelihood Scores on Mushrooms for NAM. Higher Scores Indicate Better Performance.

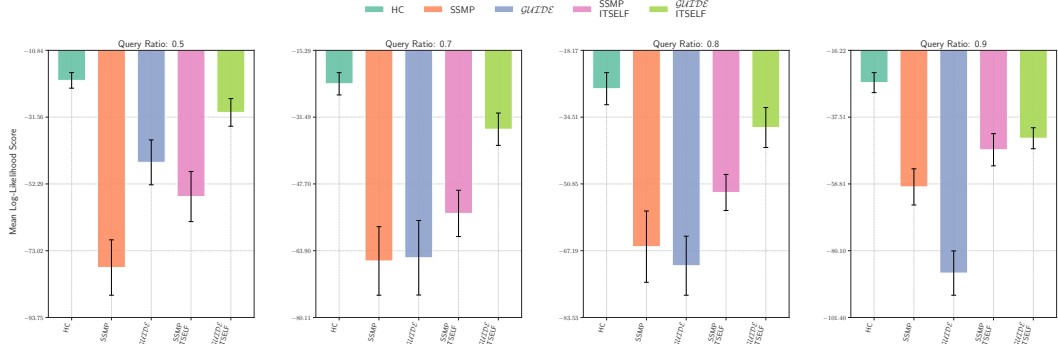

Figure 91: Log-Likelihood Scores on Connect 4 for NAM. Higher Scores Indicate Better Performance.

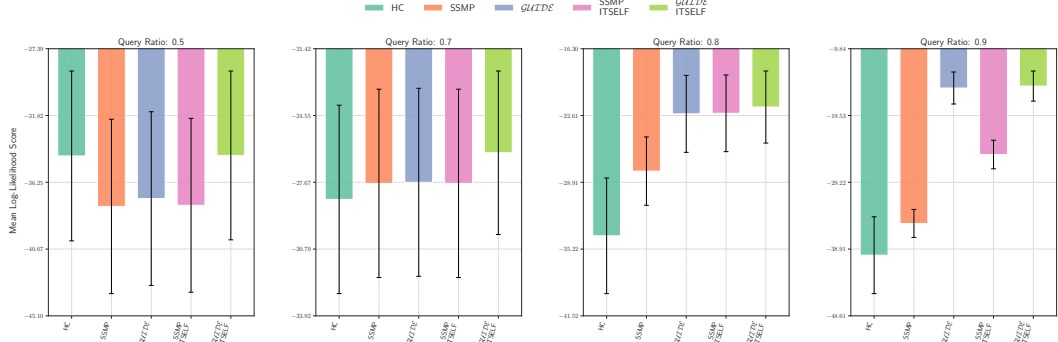

Figure 92: Log-Likelihood Scores on RCV-1 for NAM. Higher Scores Indicate Better Performance.

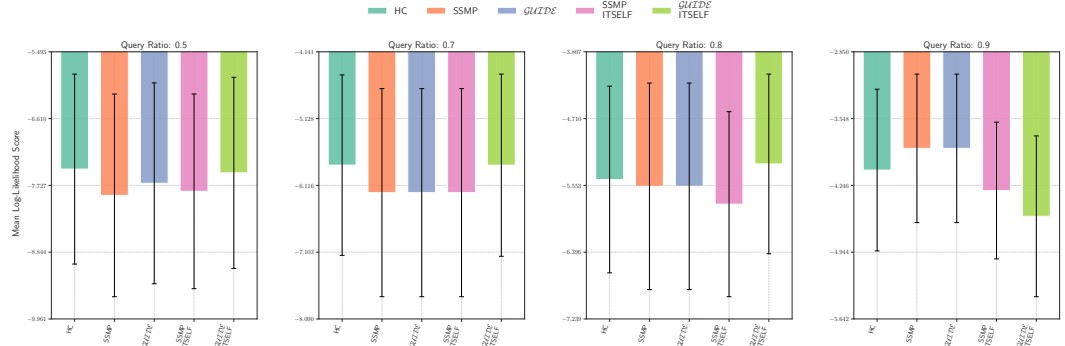

Figure 93: Log-Likelihood Scores on Retail for NAM. Higher Scores Indicate Better Performance.

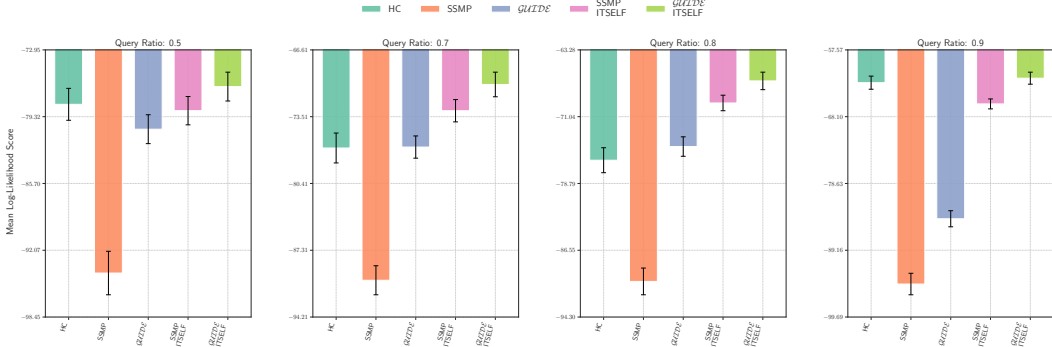

Figure 94: Log-Likelihood Scores on DNA for NAM. Higher Scores Indicate Better Performance.

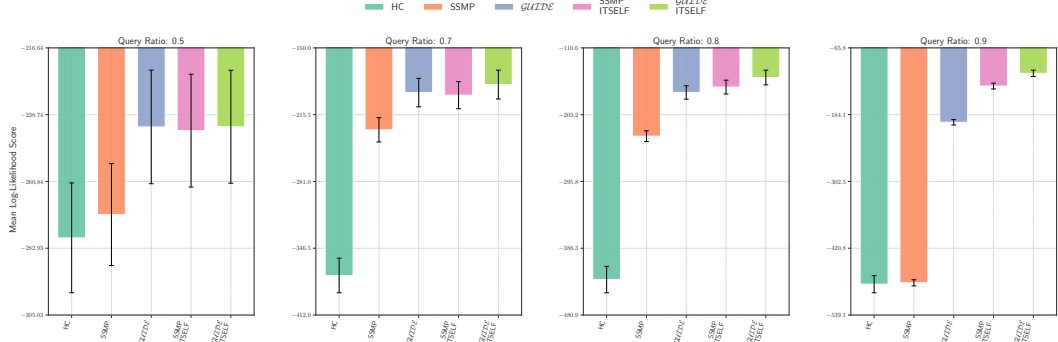

Figure 95: Log-Likelihood Scores on Movie reviews for NAM. Higher Scores Indicate Better Performance.

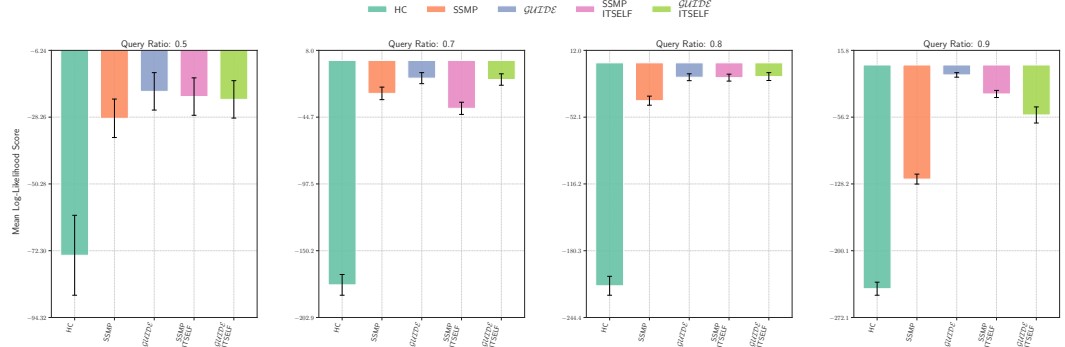

Figure 96: Log-Likelihood Scores on Book for NAM. Higher Scores Indicate Better Performance.

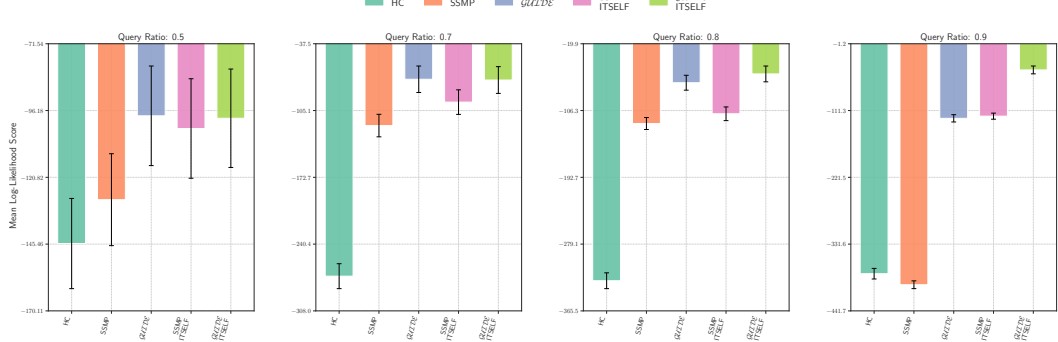

Figure 97: Log-Likelihood Scores on WebKB for NAM. Higher Scores Indicate Better Performance.

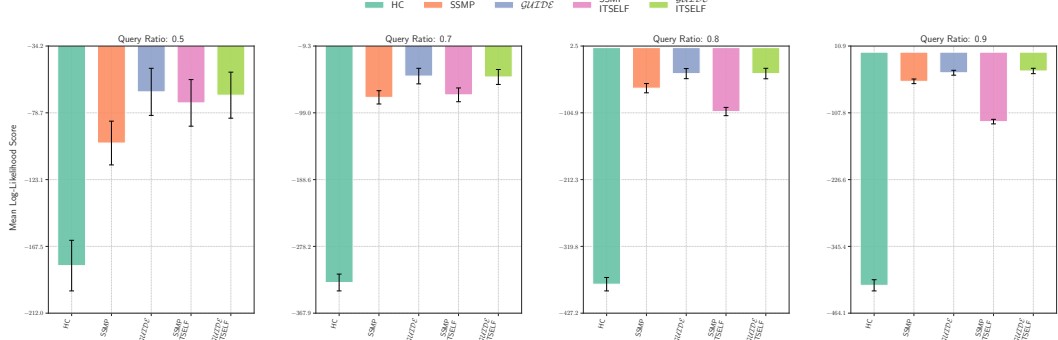

Figure 98: Log-Likelihood Scores on Reuters-52 for NAM. Higher Scores Indicate Better Performance.

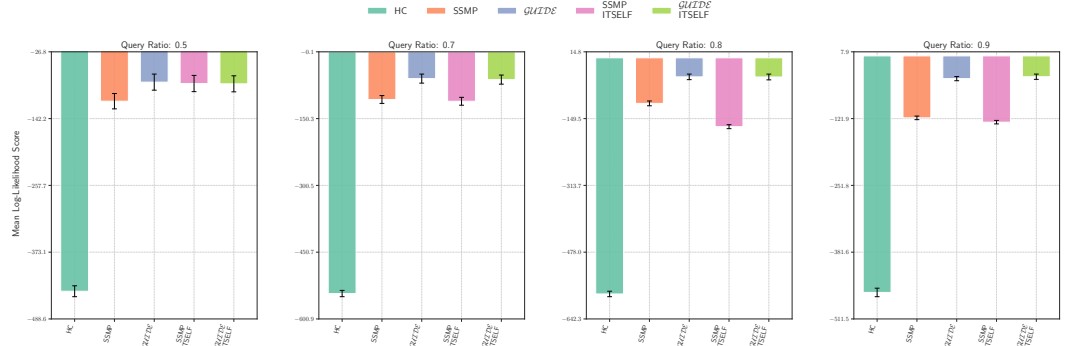

Figure 99: Log-Likelihood Scores on 20 NewsGroup for NAM. Higher Scores Indicate Better Performance.

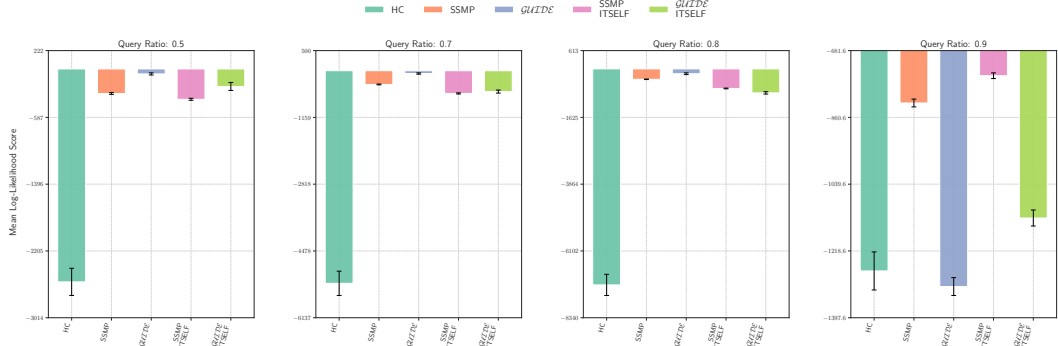

Figure 100: Log-Likelihood Scores on Ad for NAM. Higher Scores Indicate Better Performance.

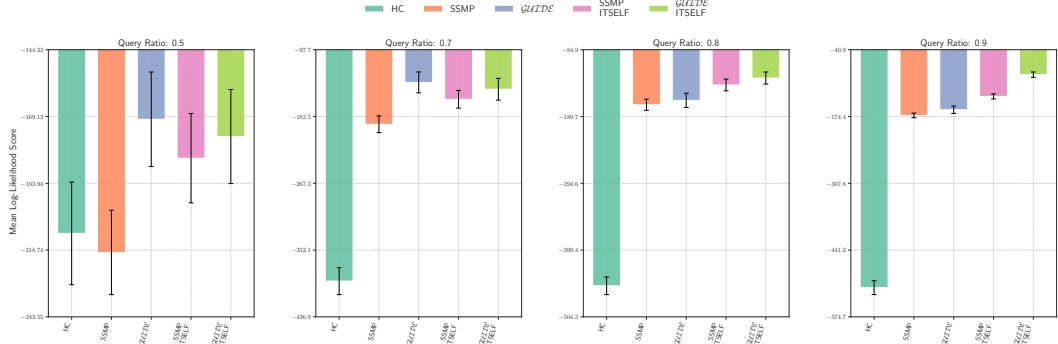

Figure 101: Log-Likelihood Scores on BBC for NAM. Higher Scores Indicate Better Performance.

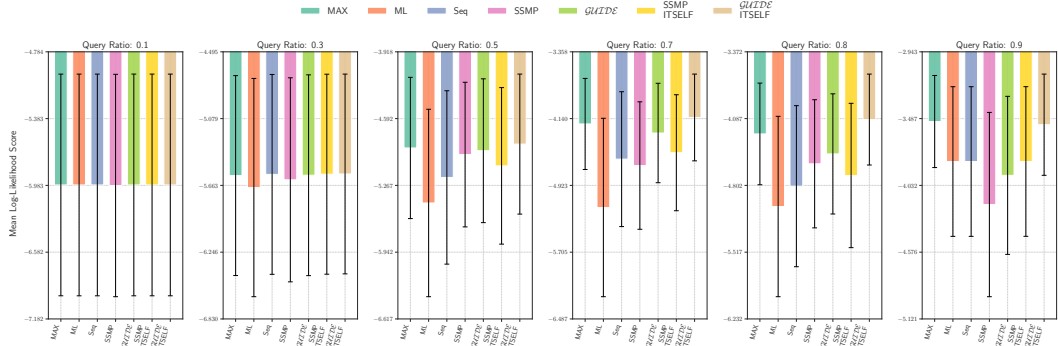

Figure 102: Log-Likelihood Scores on NLTCS for PCs. Higher Scores Indicate Better Performance.

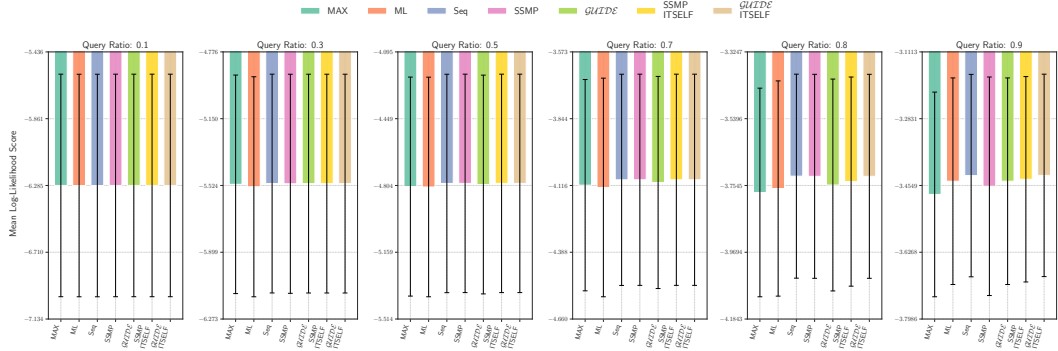

Figure 103: Log-Likelihood Scores on for PCs. Higher Scores Indicate Better Performance.

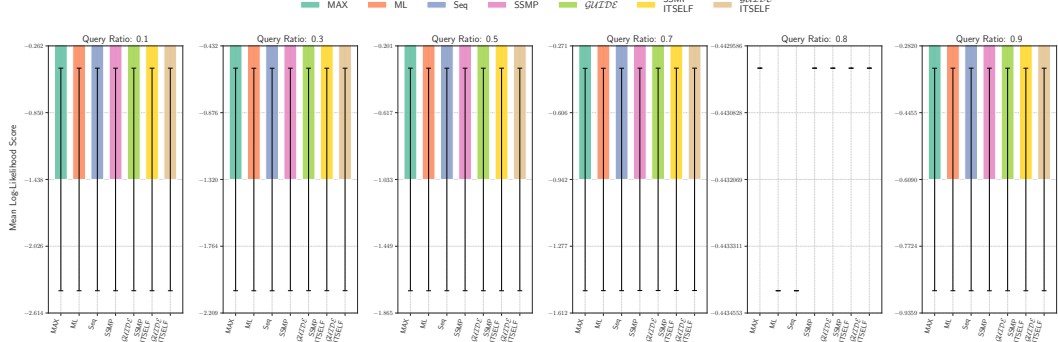

Figure 104: Log-Likelihood Scores on KDDCup2k for PCs. Higher Scores Indicate Better Performance.

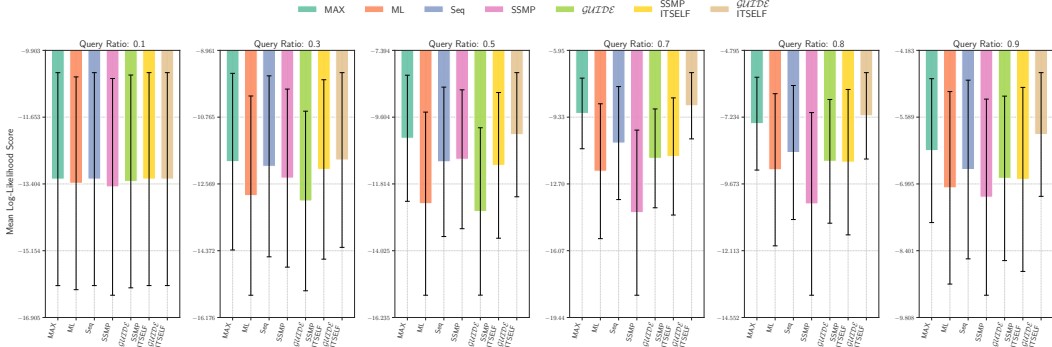

Figure 105: Log-Likelihood Scores on Plants for PCs. Higher Scores Indicate Better Performance.

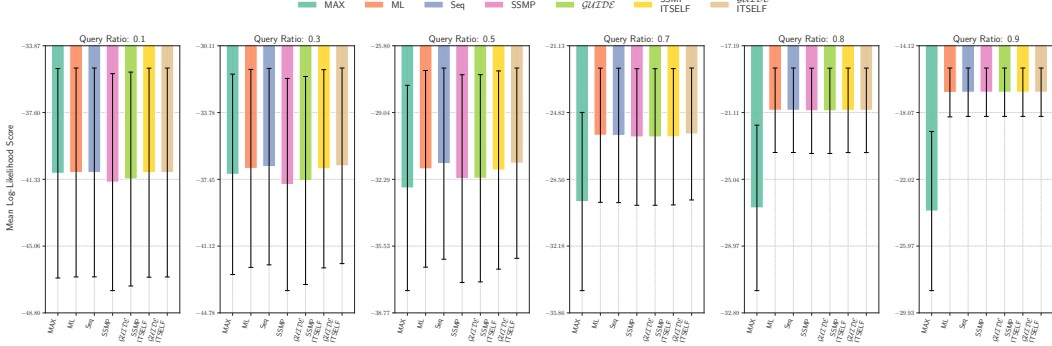

Figure 106: Log-Likelihood Scores on Audio for PCs. Higher Scores Indicate Better Performance.

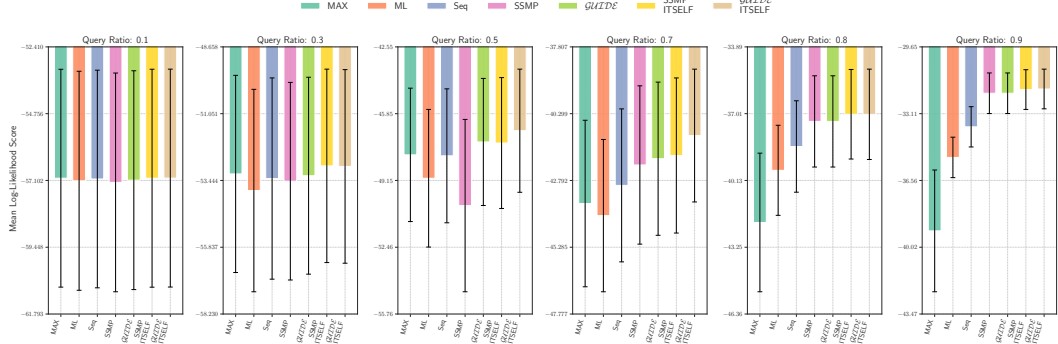

Figure 107: Log-Likelihood Scores on Jester  for PCs. Higher Scores Indicate Better Performance.

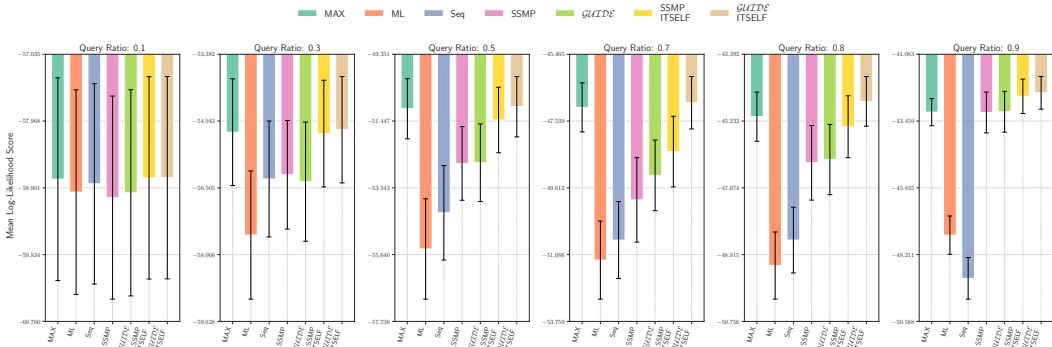

Figure 108: Log-Likelihood Scores on Netflix  for PCs. Higher Scores Indicate Better Performance.

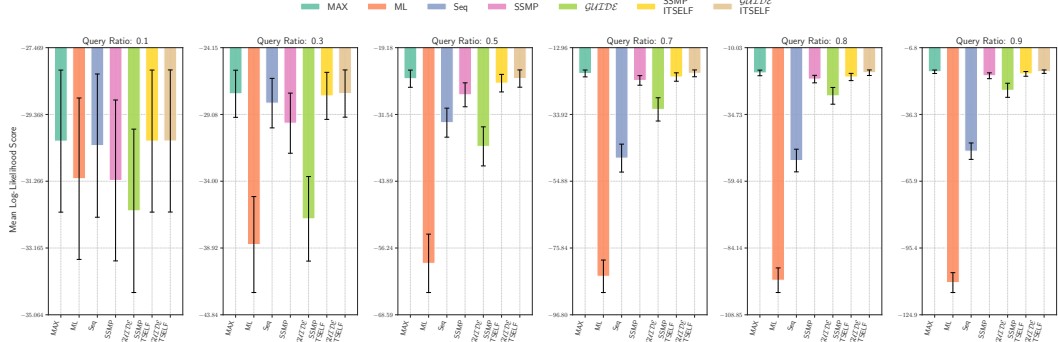

Figure 109: Log-Likelihood Scores on Accidents  for PCs. Higher Scores Indicate Better Performance.

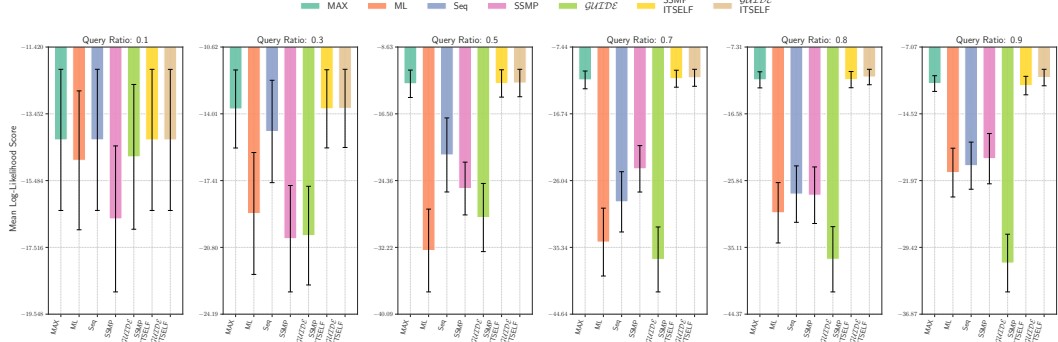

Figure 110: Log-Likelihood Scores on Mushrooms for PCs. Higher Scores Indicate Better Performance.

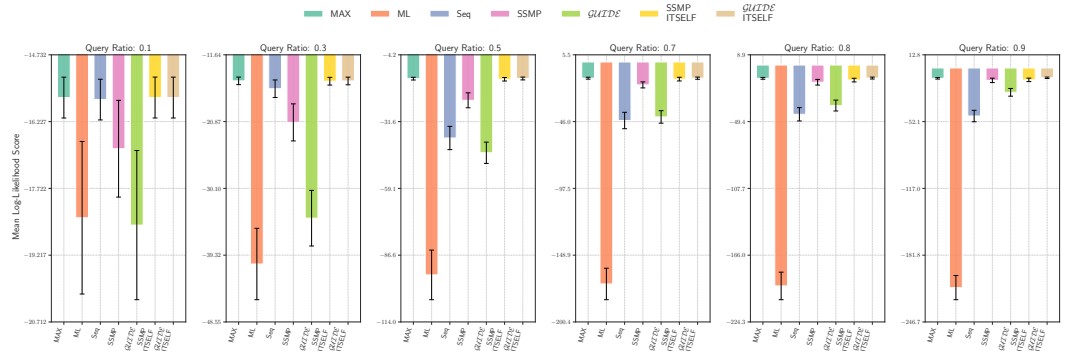

Figure 111: Log-Likelihood Scores on Connect 4 for PCs. Higher Scores Indicate Better Performance.

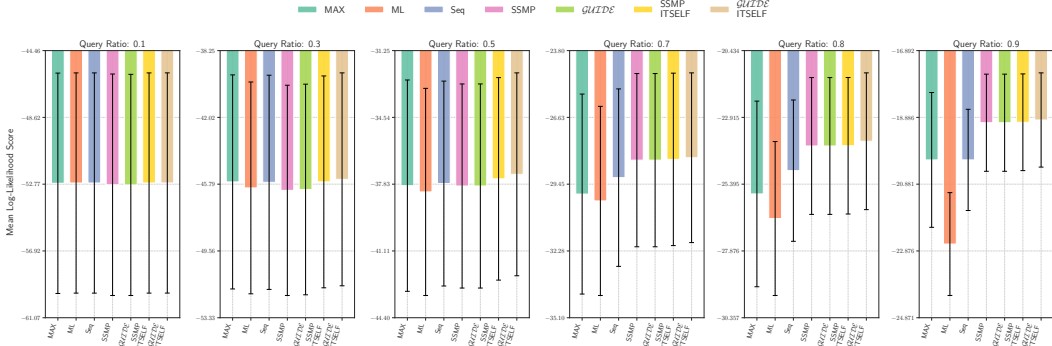

Figure 112: Log-Likelihood Scores on RCV-1 for PCs. Higher Scores Indicate Better Performance.

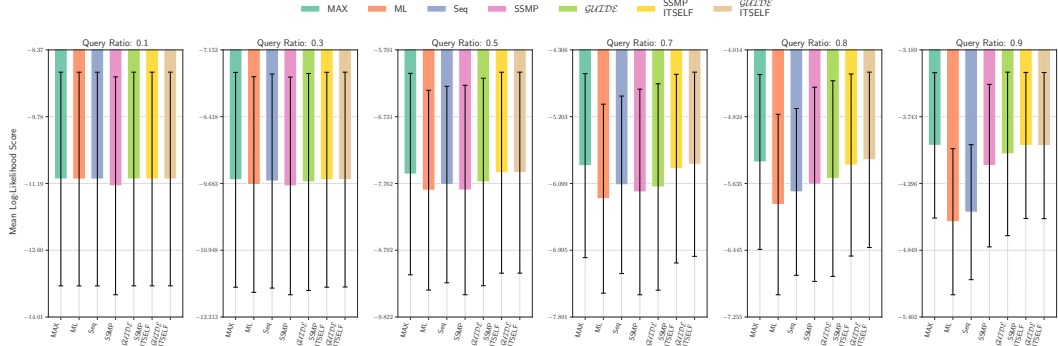

Figure 113: Log-Likelihood Scores on Retail for PCs. Higher Scores Indicate Better Performance.

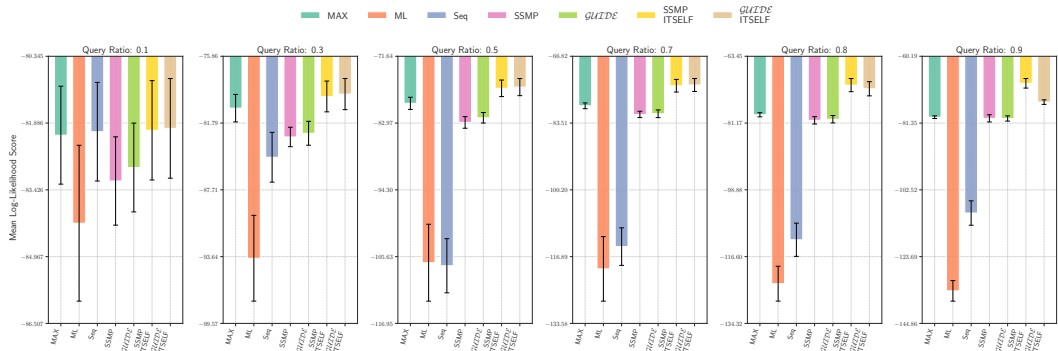

Figure 114: Log-Likelihood Scores on DNA for PCs. Higher Scores Indicate Better Performance.

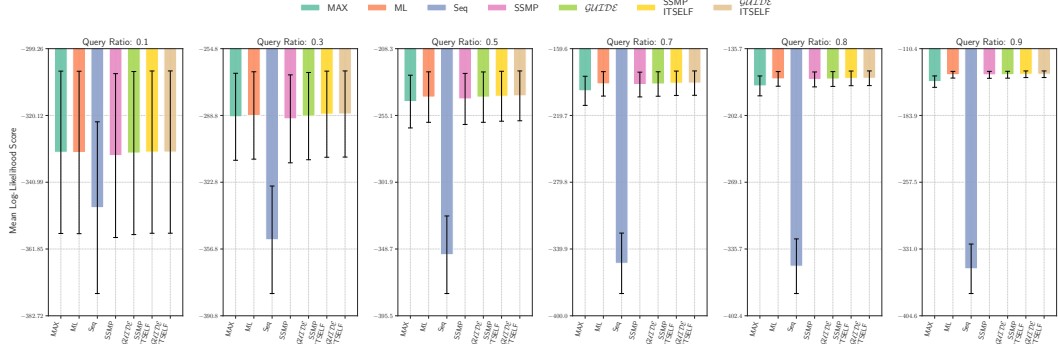

Figure 115: Log-Likelihood Scores on Movie reviews for PCs. Higher Scores Indicate Better Performance.

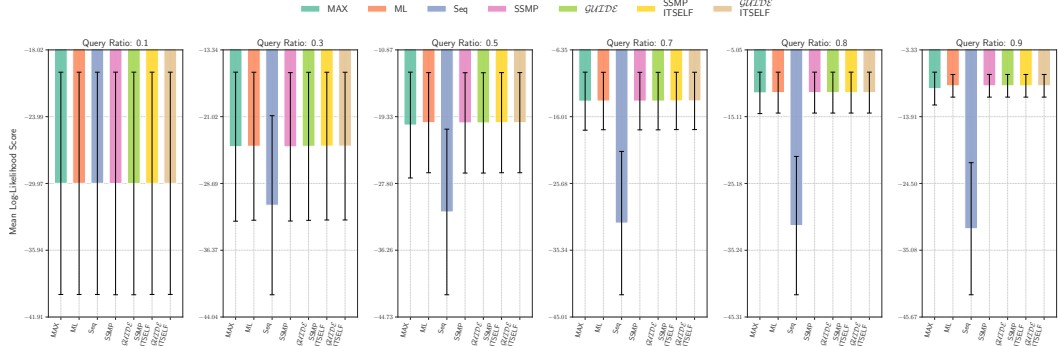

Figure 116: Log-Likelihood Scores on Book for PCs. Higher Scores Indicate Better Performance.

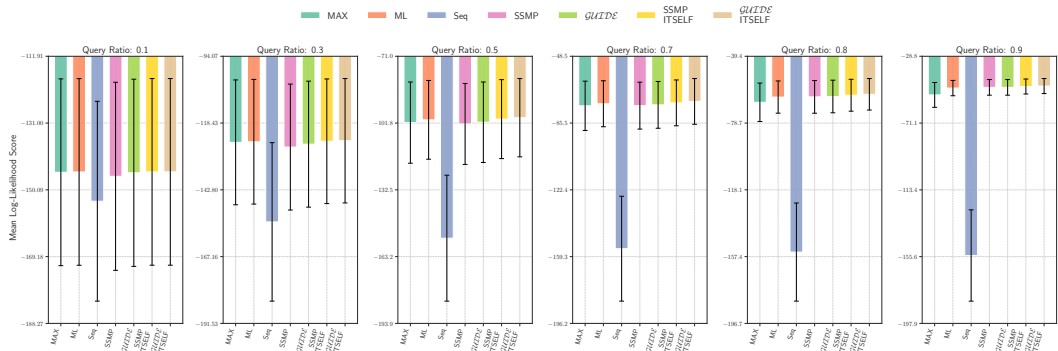

Figure 117: Log-Likelihood Scores on WebKB for PCs. Higher Scores Indicate Better Performance.

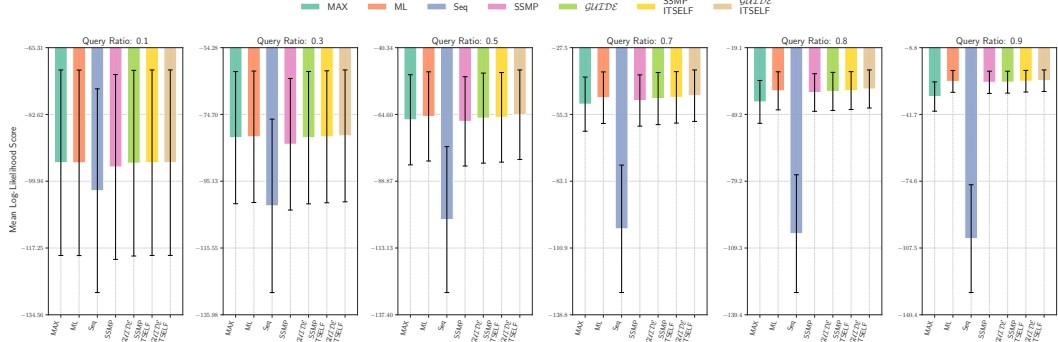

Figure 118: Log-Likelihood Scores on Reuters-52 for PCs. Higher Scores Indicate Better Performance.

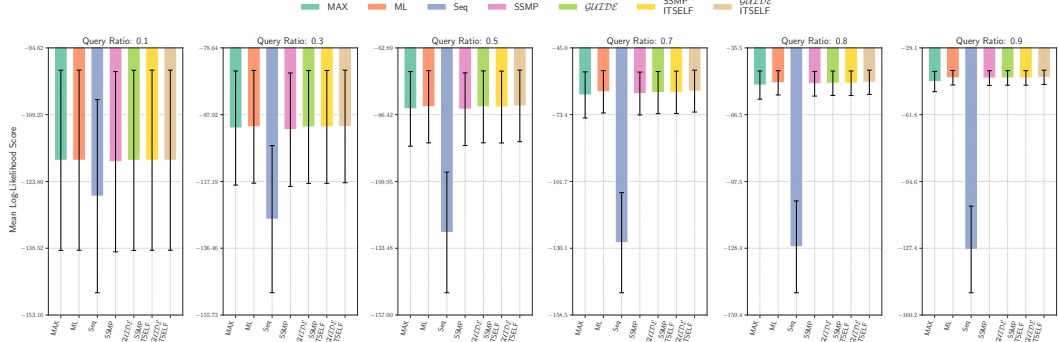

Figure 119: Log-Likelihood Scores on 20 NewsGroup for PCs. Higher Scores Indicate Better Performance.

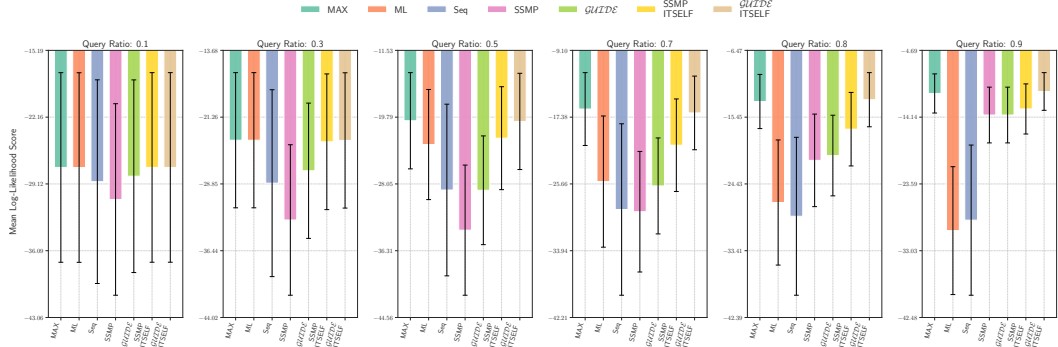

Figure 120: Log-Likelihood Scores on Ad for PCs. Higher Scores Indicate Better Performance.

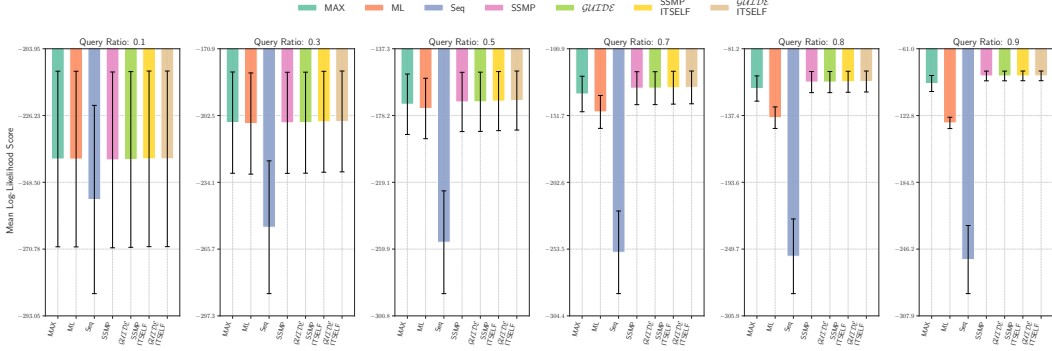

Figure 121: Log-Likelihood Scores on BBC for PCs. Higher Scores Indicate Better Performance.

