# OpenReview forum: "A Neural Network Approach for Efficiently Answering Most Probable Explanation Queries in Probabilistic Models"
_NeurIPS.cc/2024/Conference — NeurIPS 2024 spotlight_

### Official Review · Reviewer_jEuh · 2024-07-10

**Soundness:** 3
**Presentation:** 2
**Contribution:** 3
**Rating:** 7
**Confidence:** 2

**Summary:**

This paper proposes an approximate MPE inference scheme for multiple ***discrete*** probabilistic models that guarantee efficient log-likelihood computations. The approach is based on training a carefully-designed neural network to answer these queries, building upon three key ideas:
1) self-supervised learning, in order to avoid labelling via expensive MPE oracles
2) ITSELF: iterative optimization of the NN parameters at prediction time, resulting in both anytime inference and continual learning
3) GUIDE: a teacher-student training scheme for mitigating overfitting

The approach is evaluated on multiple binary benchmarks from the TPM and PGM literature.

**update after rebuttal**

The authors effectively addressed the concerns on the paper. Overall, I think that this is solid work and I hope that the authors consider adding their explanation on generalizing the approach to continuous variables and the runtime results in the main text.

**Strengths:**

- The paper is overall well-written
- The key contributions are sensible and (to the best of my knowledge) novel
- Extensive empirical evaluation

**Weaknesses:**

- The limitations of the approach are not clearly stated.
- Some experimental details are unclear
- Runtime evaluation, a crucial aspect of any approximate algorithm, is deferred to the supplementary work

**Questions:**

1) "Without loss of generality, we use binary variables" How is this WLOG? Does the approach seems limited to discrete models. If that's the case, I would be more upfront about that.
2) Figure 1: what are the numbers in the contingency tables?
3) How often is the approach recovering the true MPE assignment? Why not devising an experiment where ground-truth solutions can be obtained?
4) What is the training time of the proposed approach?

---

### Minors:

Lines 139-143: is there a difference between the NN relaxation and the PGM/PC? It looks identical on a first glance.

Figure 2: Shouldn't the caption be "Top - PC, Bottom - ***NAM***"? Also in b), shouldn't it be "GUIDE + ITSELF ***vs.*** HC"?

**Limitations:**

The limitations of the approach are not clearly stated in the text.

---

> ### Author Rebuttal · Authors · 2024-08-07
>
> ## Response to Reviewer jEuh
> ### Comment 1
> > The limitations of the approach are not clearly stated.
>
> **Response:** We have discussed our approach's limitations in the Conclusion and Future Work section. We summarize the limitations below:
>
> The primary limitations are the inability to answer complex queries with constraints and the current lack of support for training a single neural network with losses from multiple PMs to embed their inference mechanisms. Employing a more advanced encoding scheme could significantly improve the proposed method's performance. Addressing these limitations will be the focus of our future research.
>
> ### Comment 2
> > Runtime evaluation is deferred to the supplementary work
>
> **Response:** Due to space constraints, we were unable to include the figure for runtime evaluation in the main paper. However, we have provided a detailed comparison of the methods' inference times in lines 374-378 of the manuscript. Upon acceptance, we will incorporate the relevant figures into the final version of the paper as well.
>
> ### Comment 3
> > "Without loss of generality, we use binary variables" How is this WLOG? Does the approach seems limited to discrete models.
>
> **Response:** The current approach can be extended to handle both multi-valued discrete and continuous variables.
>
> For multi-valued discrete variables:
>
> - We can adapt the method by employing a multi-class, multi-output classification head.
> - The neural network will have a softmax output node for each query variable, with the softmax acting as soft evidence.
>
> For continuous variables:
>
> - We will use a linear activation function at the output layer.
> - The loss function (multi-linear representation of the Probabilistic Model) will be modified to accommodate continuous neural network outputs.
> - For example, in Probabilistic Circuits over continuous variables using Gaussian distributions, continuous values can be directly plugged into the loss function for gradient backpropagation.
>
> This extension primarily requires adjusting the network's output layer and the self-supervised loss function (defined based on the PM). All other aspects of our method, including the ITSELF and GUIDE procedures, remain unchanged.
>
> ### Comment 4
> > Figure 1: what are the numbers in the contingency tables?
>
> **Response:** We kindly refer the reviewer to lines 301-305 in Section 5 for the interpretation of the numbers in the contingency tables. For clarity, we reiterate the explanation here:
>
> > "Each cell $(i,j)$ in the table represents the frequency with which the method in row $i$ outperformed the method in column $j$ based on average log-likelihood scores. Any discrepancy between the total number of experiments and the combined frequencies of cells $(i,j)$ and $(j,i)$ indicates instances where the compared methods achieved similar scores."
>
> For example, in Figure 1a, the value 108 (bottom left corner) indicates that GUIDE+ITSELF (row) outperforms the MAX approximation (column) in 108 out of 120 instances for Probabilistic Circuits.
>
> ### Comment 5
> > How often is the approach recovering the true MPE assignment? Why not devising an experiment where ground-truth solutions can be obtained?
>
> **Response:** We present the experiments on smaller models in Section D of the supplementary material and compare the solutions provided by the proposed method with the exact solutions. We specifically examine the gap between the scores of exact solutions and all neural-based methods. The graphical models used in this comparison have low treewidth, allowing the computation of exact solutions for these datasets.
>
> Answering MPE queries over probabilistic models is an NP-Hard task, limiting the availability of ground truth solutions to models with few variables and simple structures (small treewidth). Consequently, our main evaluations on larger, more complex datasets do not include direct comparisons to ground truth. But recall that higher the likelihood of the solution, the better the solution, and therefore, we can always compare solutions output by multiple methods even though the ground truth is not available.
>
>
>
> ### Comment 6
> > What is the training time of the proposed approach?
>
> **Response:** The training duration for our models varies based on model complexity and dataset size, typically ranging from 10 minutes to 2.5 hours. Notably, our ITSELF algorithm offers a significant advantage in this regard. By initializing the network with random weights (no training time), ITSELF can achieve near-optimal solutions, thus requiring zero training time.
>
> ### Comment 7
> > Lines 139-143: is there a difference between the NN relaxation and the PGM/PC?
>
> **Response:** The neural network (NN) relaxation serves as a continuous approximation of the discrete Probabilistic Model (PM). This approximation is essential because:
>
> 1. NN outputs are continuous values in $[0,1]$, which cannot be directly plugged into the discrete PM function.
> 2. Converting continuous values to binary through thresholding or similar operations may not be differentiable, hindering gradient flow.
>
> To address this, we define a continuous loss function $\ell^c(\mathbf{q}^c,\mathbf{e}): [0,1]^n \rightarrow \mathbb{R}$ that coincides with the discrete loss $\ell$ on $\\{0,1\\}^n$. While we explored alternatives like the Straight Through Estimator, our proposed loss with penalty demonstrated superior performance.
>
> Propositions 1 and 2 provide further details and theoretical results regarding this approximation.
>
> ### Comment 8
> > Figure 2: Shouldn't the caption be "Top - PC, Bottom - NAM"? Also in b), shouldn't it be "GUIDE + ITSELF vs. HC"?
>
> **Response:** MADE is the NAM model selected for our experiments, which explains why we used the term "Bottom - MADE" instead of "Bottom - NAM." You are correct in noting that the caption for Figure 2b should be "GUIDE + ITSELF vs. HC." We will update the caption accordingly in the revised manuscript.

---

> > ### Comment · Reviewer_jEuh · 2024-08-11
> > **Response to the authors**
> >
> > Thank you for clarifying some aspects of your work. I don't have any further questions at the present time.

---

> > > ### Comment · Area_Chair_gopV · 2024-08-12
> > >
> > > Dear reviewer,
> > >
> > > Your initial review was short and shallow. For instance, you wrote that some experimental details are unclear without giving examples or pointers. Now, after the authors have provided a rebuttal, answering all of the weaknesses you mentioned and your questions, you simply answer that you don't have any more questions, but without raising your score. Unfortunately, I will not be able to consider your review in my decision making unless you provide more substance and concrete reasons why the paper should not be accepted.

---

> > > > ### Comment · Reviewer_jEuh · 2024-08-13
> > > > **Response to AC**
> > > >
> > > > Dear AC,
> > > >
> > > > Sorry to hear that you found my review shallow. To clarify, Q1 refers to Weakness 1, Q2 and Q3 refer to Weaknesses 2 and 3, and the authors adeguately addressed those concerns in their response. I should have mentioned that.
> > > >
> > > > Since the discussion time was (is) not over, I was simply taking my time to formulate further questions.
> > > > I don't see the hurry in immediately raising the score, we have a 9 days long AC/reviewers discussion phase ahead.

---

### Official Review · Reviewer_1g2T · 2024-07-13

**Soundness:** 3
**Presentation:** 3
**Contribution:** 3
**Rating:** 7
**Confidence:** 4

**Summary:**

This paper proposes a neural network-based approach to approximate the most probable explanations (MPE) on probabilistic models. The basic idea is to train a neural network that takes as input an encoding of an assignment to an arbitrary subset of evidence variables and outputs an assignment to the remaining query variables, with the objective to maximize the likelihood given the predicted assignment. This is feasible on probabilistic models that support tractable likelihood and gradient computations such as PGMs, PCs, and NAMs. The authors present two key strategies to enhance the basic architecture. First, they employ a teacher-student architecture (GUIDE) that trains a student network by supervised learning with the teacher network’s MPE solutions as labels. In addition, they introduce inference time optimization to further refine the pre-trained models given specific MPE queries. The proposed methods are evaluated empirically on benchmark datasets and compared against baseline approximation methods.

**Strengths:**

+ The proposed approach can answer any-MPE without a priori fixing the query/evidence variables, addressing a major limitation of prior work using neural networks to compute MPE / marginal MAP queries. The improvements using teacher-student architecture and inference tim optimization are also interesting and effective as demonstrated by experiments.

+ The paper is overall well-written and easy to follow (minus some minor comments below).

+ The experiments are thorough, and the results convincingly show that GUIDE+ITSELF outperforms baseline approximation methods across datasets and types of probabilistic models.

**Weaknesses:**

- It appears that the inference time optimization is doing a lot of the heavy lifting compared to pretraining. On experiments on PCs, the simple baseline MAX, which is a popular method for PCs, seems to outperform both SSMP and GUIDE without inference time training. The same is true, albeit to a lesser degree, for MADE.

- Evaluation based on average joint likelihood p(q,e) may be biased towards the neural-network based approaches, compared to evaluating using average conditional likelihood p(q|e). Because the loss function is based on joint likelihood, it will bias the network training to perform better on evidence e with larger probability p(e) than those with smaller probability. Average joint log-likelihood could easily hide poor results on instances with low probability evidence.


- The significance of Propositions 1 & 2 is unclear. More useful in motivating the training objective might be showing that the chosen loss function upper-bounds the target loss (negative log-likelihood) that we want to minimize.

**Questions:**

1. Can the proposed approach be extended to continuous variables?

2. In each iteration of GUIDE, the teacher network is reinitialized with the student network. Could this result in the network being stuck at a local minimum?

3. How do the results compare with exact solutions (on model and problem instances that can be solved exactly)?

4. How are the results if evaluating using conditional likelihood p(q|e) instead of p(q,e)?

5. How many unique query instances (evidence variables and assignments) were used in the training data? (ie what proportion of all possible instances)

Other comments:
- I would suggest adding a diagram showing the components of the proposed method to improve clarity.
- Figure 2: typos in the percentage difference labels (e.g., >-10 and <10 should be >10 and <-10).
- Why was this particular input encoding chosen? For example, a natural alternative using an input node to indicate whether a variable $X_i \in \mathbf{E}$ and another input node to denote its value $x_i$.

**Limitations:**

The current approach is currently limited to binary variables.

---

> ### Author Rebuttal · Authors · 2024-08-07
>
> ## Response to Reviewer 1g2T
> ### Comment 1
> > It appears that the inference time optimization is doing a lot of the heavy lifting compared to pretraining.
>
> **Response:**
> The any-MPE task is challenging due to exponential input configurations and variable divisions. Training a single network for all queries is difficult due to distribution shifts. Traditionally, this would require large networks to accommodate all possibilities. ITSELF addresses these issues through novel inference time optimization, enabling superior results in any-MPE task:
>
> 1. It directly optimizes neural network parameters on test examples by using a self-supervised loss function.
> 2. It enables efficient adaptation to diverse query distributions without extensive pre-training.
>
> Despite the dependence on ITSELF, GUIDE provides a superior initialization for ITSELF among various pre-training methods, leading to faster convergence, improved output quality, and reduced inference time (Sections B and C, supplementary material).
>
> In Section 4 (lines 187-200), we also provide detailed information and prior studies supporting this approach, highlighting its effectiveness and theoretical foundation.
>
> This approach offers several key advantages:
>
> 1. GUIDE enables the use of more compact network architectures.
> 2. The number of iterations required at test time decreases.
>
> By leveraging GUIDE's initialization capabilities and ITSELF's adaptive inference, we achieve superior performance among compared methods.
> ### Comment 2
> > How are the results if evaluating using conditional likelihood $p(q|e)$ instead of $p(q,e)$?
>
> **Response:**
> The log-conditional likelihood evaluation will not alter the ranking of query variable Q assignments because $$\max_q \log P(q|e) = \max_q \log P(q,e) - \log P(e)$$ The term $\log P(e)$ remains constant for all solutions for a given query instance. Furthermore, using the conditional likelihood $P(q|e)$ as the loss function would be identical to using the likelihood since $\log P(e)$ does not contribute gradients to the loss.
> ### Comment 3
> > The significance of Propositions 1 and 2 is unclear.
>
> **Response:** Thank you for the comment. We will rephrase the propositions to highlight their bounding properties, emphasizing that our chosen loss function is a valid surrogate for the target negative log-likelihood (NLL).
> ### Comment 4
> > Can the proposed approach be extended to continuous variables?
>
> **Response:**
> Yes, the approach can be easily adapted to continuous variables by using a linear output activation function. The loss function (PM) also needs to be modified to allow the use of continuous neural network outputs. For instance, PCs over continuous variables use Gaussian distributions for each variable. The continuous NN output can be inserted into the modified loss function, allowing gradient backpropagation.
> ### Comment 5
> > In each iteration of GUIDE, the teacher network is reinitialized with the student network. Could this result in the network being stuck at local minimum?
>
> **Response:** Theoretically and empirically, a sufficiently large teacher network should approach the global optimum, as discussed in Section 4, lines 194-197. Updating the teacher network with multiple MPE query instances per iteration mitigates the risk of local minima, even when initialized from the student network.
>
> The student-based teacher initialization aims to improve ITSELF's starting point for updating the teacher network. This approach accelerates convergence to near-optimal batch solutions.
>
> ### Comment 6
> > How do the results compare with exact solutions?
>
> **Response:** Due to space limitations in the main paper, we have included a detailed evaluation in Section D of the supplementary material. This section examines the performance gap between solutions provided by our neural-based method and exact solutions. We specifically selected GMs with low treewidth for this comparison, enabling the computation of exact solutions.
> ### Comment 7
> > How many unique query instances (evidence variables and assignments) were used in the training data? (ie what proportion of all possible instances)
>
> **Response:**
> For training the model, we extract the evidence values from the instances in the training dataset by randomly partitioning each example into query and evidence subsets. Consequently, as long as the model is trained, it will encounter new randomly generated query instances. Therefore, more epochs imply exposure to more unique instances.
> ### Comment 8
> > I would suggest adding a diagram showing the components of the proposed method to improve clarity.
>
> **Response:** We appreciate your suggestion and will include a diagram illustrating the key components of our proposed method.
> ### Comment 9
> > Figure 2: typos in the percentage difference labels (e.g., $>-10$ and $<10$ should be $>10$ and $<-10$).
>
> **Response:** We appreciate you bringing up the inconsistency in the percentage difference labels in Figure 2. We will update these labels in the revised manuscript.
> ### Comment 10
> > Why was this particular input encoding chosen?
> **Response:** We can employ any injective mapping from the set of all possible MPE queries over the given PM to the set $\\{0,1\\}^{2n}$. Your suggested mapping is valid. However, for variables in the query set, we must assign a value to the second node representing the variable's unknown value.
>
> We chose the specific encoding because it allows for an extension to an encoding suitable for training the neural network for the marginal MAP task, where the assignment (1,1) can be used for unobserved variables.
>
> ### Comment 11
> > The current approach is currently limited to binary variables.
>
> **Response:** Our approach extends to both multi-valued discrete and continuous variables. For multi-valued discrete variables, we can employ multi-class multi-output classification, using softmax output nodes for each query variable as soft evidence. The extension to continuous variables is in response to comment 4.

---

> ### Comment · Reviewer_1g2T · 2024-08-12
>
> Thank you for the detailed response. Most of my comments have been address, but I have a follow up regarding comment 2.
>
> >Furthermore, using the conditional likelihood $P(q|e)$ as the loss function would be identical to using the likelihood since $\log P(e)$ does not contribute gradients to the loss.
>
> While this is true when $e$ is a single fixed evidence, the data consists of multiple instances with different evidence assignments. In average joint log-likelihood, $\log P(q,e)$ for some evidence $e$ with high probability $P(e)$ could dominate $\log P(q',e')$ for another evidence $e'$ with much lower probability $P(e')$. Average conditional likelihood on the other hand weighs instances equally regardless of the probability of evidence of each instance. The dataset is constructed by sampling from the PM, which is likely to sample evidence assignments with higher probability more often, and average joint likelihood would weigh those instances even more. Thus, comparison by average joint log-likelihood could theoretically hide poor performance on low likely instances.
>
> While the discussion period is closing soon, I would highly encourage the authors to consider also evaluating with average conditional likelihood in the revision.

---

> > ### Author Response · Authors · 2024-08-13
> >
> > We believe what you are suggesting is to evaluate the impact of instances with relatively low probability of evidence on the performance of our method. Your idea of using average conditional likelihood instead of just average log likelihood is an excellent one and will help address the influence of low-probability instances on our method’s performance. While this is definitely a step in the right direction, it might not fully capture the entire picture.
> >
> > To build on this, and because your suggestion has inspired further thinking, we plan to use techniques like slice sampling or SampleSearch to specifically generate and evaluate these low-probability instances. This approach will allow us to thoroughly assess our model’s performance, particularly in handling rare events. We believe ITSELF and GUIDE will perform better on such instances because they use test time training.

---

### Official Review · Reviewer_q7PK · 2024-07-16

**Soundness:** 3
**Presentation:** 3
**Contribution:** 3
**Rating:** 6
**Confidence:** 2

**Summary:**

This paper proposes a novel neural network-based method to solve the computationally challenging task of finding the Most Probable Explanation (MPE), which is known to be NP-hard. The proposed method involves an inference-time optimization process with a self-supervised loss function to iteratively improve the solutions. It also employs a teacher-student framework that provides a better initial network, which in turn helps reduce the number of inference-time optimization steps. Experiments demonstrate the efficacy and scalability of the proposed method compared to various baselines across different datasets.

**Strengths:**

The paper presents a series of innovative methods, including inference-time optimization, a teacher-student framework, and self-supervised learning, to effectively address the MPE task.
The experimental results comprehensively evaluate the performance of the proposed approach and provide strong support for the claims made in the paper.
The paper is well-organized and clearly explains the details of the different modules and techniques used in the proposed solution.

**Weaknesses:**

The presentation of the main experimental results could be improved. The captions of Figure 1 and Figure 2 are quite simple and do not provide enough details about the information conveyed in the figures. It would be better to include more descriptive captions that explain the meaning of the rows, columns, and color schemes used in the visualizations.

**Questions:**

1. Could the authors provide more detailed explanations or visualizations to help the reader better understand the experimental setup and the interpretation of the results presented in Figures 1 and 2?
2. Are there any plans to further extend or generalize the proposed method to handle other types of probabilistic inference tasks beyond the MPE problem?

**Limitations:**

N.A

---

> ### Author Rebuttal · Authors · 2024-08-07
>
> ## Response to Reviewer q7PK
>
> ### Comment 1
> > The presentation of the main experimental results could be improved. The captions of Figure 1 and Figure 2 are quite simple and do not provide enough details about the information conveyed in the figures. It would be better to include more descriptive captions that explain the meaning of the rows, columns, and color schemes used in the visualizations.
> >
> > Could the authors provide more detailed explanations or visualizations to help the reader better understand the experimental setup and the interpretation of the results presented in Figures 1 and 2?
>
> **Response:**
> Thank you for your suggestion. We apologize for not providing detailed captions for the figures due to space limitations. However, we have included the necessary information in the main text (except the color scale description). Figure 1's heatmaps are explained in lines 301-305, while Figure 2's details are provided in lines 315-319, coinciding with their initial references.
>
> For the reviewer's convenience, we offer concise descriptions of the main text figures:
>
> The sub-figures in Figure 1 present heatmaps comparing the performance of different MPE solvers based on their average log-likelihood scores.
>
> > Each cell $(i,j)$ in the table represents how often the method in row $i$ outperformed the method in column $j$ based on average log-likelihood scores. Any difference between the total number of experiments and the combined frequencies of cells $(i,j)$ and $(j,i)$ indicates cases where the compared methods achieved similar scores.
>
> The color scale ranges from dark blue (higher values) to dark red (lower values), with lighter shades representing intermediate scores. To enhance clarity, we will provide a more detailed description of the color scale in the figure caption.
>
> Figure 2 displays heatmaps comparing the proposed method to the baseline using percentage differences in mean LL scores (numbers on the gradient axis showcase this value). We have provided the description of the comparison in the main text:
>
> > The y-axis presents datasets sorted by variable count and the x-axis by query ratio. Each cell displays the percentage difference in mean LL scores between the methods, calculated as:
>
>
> $$\text{Percentage Difference} = 100 \times (ll_{nn} - ll_{max}) / |ll_{max}|$$
>
> The color gradient spans from dark green (proposed method outperforms) to dark red (baseline surpasses), with lighter shades indicating intermediate performance differences.
>
> Furthermore, sections 5.1 and 5.2 describe the experimental setup in detail, including information about the datasets, graphical models used for MPE inference, baseline methods, and evaluation criteria. To ensure reproducibility, we have included detailed log-likelihood scores for all experiments in the supplementary material.
>
> We appreciate the reviewer's recommendation and concur that additional details about the color scales in the heatmaps will enhance the clarity and interpretability of Figures 1 and 2. Upon acceptance, we will incorporate these color scale descriptions into the main text. Other relevant details are already present in the main text or appendix, and we will emphasize these points as needed to improve overall comprehension.
>
> ### Comment 2
> > Are there any plans to further extend or generalize the proposed method to handle other types of probabilistic inference tasks beyond the MPE problem?
>
> **Response:**
> We plan to extend the proposed method to various inference tasks beyond MPE, contingent on the feasibility of loss function computation. Our future research directions include:
>
> 1. We will adapt our method to answer Marginal MAP queries over Probabilistic Circuits by modifying the loss function appropriately.
>
> 2. Constrained inference tasks: We aim to apply our method to problems such as the Constrained Most Probable Explanation Task.
>
> However, it is important to note that the scalability of the proposed method is dependent on the efficiency of evaluating the loss function for the given inference task. In cases where the evaluation of the loss function becomes computationally infeasible, the applicability of the proposed method (to train a neural network to answer queries over probabilistic models) may be limited. One such example is performing marginal MAP inference over Neural Autoregressive Models and Probabilistic Graphical Models. In these scenarios, the repeated computation of the loss function value and its gradient during the training phase, which is necessary for updating the parameters of the neural network, can become prohibitively expensive.

---

### Author Rebuttal · Authors · 2024-08-07

# Author Response to Reviews for Paper #12727
We appreciate the reviewers' constructive feedback, which will enhance our paper's quality. We have addressed all points raised and additionally provide the following responses to common questions and concerns below:

## Presentation of experimental results:
We acknowledge the need for improved clarity in presenting our experimental results. While the main paper contains most details about the figures, we will enhance the presentation by:

1. Adding color descriptions for the heatmaps in Figures 1 and 2.
2. Relocating descriptions from the main text to figure captions for easier comprehension.

These changes will facilitate a more thorough understanding of our empirical evaluation.

## Adaptation to other data types and inference tasks:

Although our experiments focus on binary variables, our proposed approach is adaptable to various scenarios. Continuous variables and multi-valued (discrete) variables can be accommodated by modifying the neural network's output layer and utilizing the corresponding PM as the loss function.

The method can also be extended to other inference tasks over PMs, such as marginal MAP and constrained most probable explanation problem.

## Comparison with exact solutions:
We recognize the importance of comparing our neural-based methods with exact solutions. However, due to the NP-hardness of MPE, computing exact solutions is infeasible for larger datasets. Nonetheless, as higher likelihoods indicate better solutions, we can still compare outputs from various methods, even in the absence of ground truth.

Our approach to this issue is twofold:

1. For main experiments using large datasets and probabilistic models, we compare our method against state-of-the-art approximate methods using likelihood scores.
2. In the supplementary material, we compare log-likelihood scores between our proposed method and exact solutions on smaller datasets.

---

### Comment · Area_Chair_gopV · 2024-08-07
**Please read and respond to the rebuttal**

Dear reviewers,

First of all, thank you for your service to the ML community. Writing high-quality reviews and engaging with authors' responses is essential to a healthy ML research community.

It is essential that you read the rebuttals and provide a response and/or follow-up questions within the next few days. This will allow the authors sufficient time to react. While a detailed response addressing all points is not necessary, at a minimum, you should indicate you have read and considered the review and whether you will maintain or revise your score. Please also take the time to read the other reviews. Understanding your fellow reviewers' key arguments can help converge towards more similar ratings in cases of diverging scores.

I want to thank you again, and I look forward to following the discussions here.

---

### Decision · Program_Chairs · 2024-09-25

**Decision:**

Accept (spotlight)

**Comment:**

There is agreement among the reviewers that the paper addresses an interesting and important problem and proposes an entirely new approach to answering MPE queries. Some concerns have been adequately addressed during the rebuttal.